# Applications of $p_T$-$x_R$ Variables in Describing Inclusive Cross Sections at the LHC

Frank E. Taylor

Department of Physics, Laboratory for Nuclear Science Massachusetts Institute of Technology Cambridge, Cambridge, MA 02139, USA; fet@mit.edu

**Abstract:** Invariant inclusive single-particle/jet cross sections in p–p collisions can be factorized in terms of two separable $p_T$ dependences, a $[p_T - \sqrt{s}]$ sector and an $[x_R - p_T - \sqrt{s}]$ sector. Here, we extend our earlier work by analyzing more extensive data to explore various s-dependent attributes and other systematics of inclusive jet, photon and single particle reactions. Approximate power laws in $\sqrt{s}$, $p_T$ and $x_R$ are found. Physical arguments are given which relate observations to the underlying physics of parton–parton hard scattering and the parton distribution functions in the proton. We show that the $A(\sqrt{s}, p_T)$ function, introduced in our earlier publication to describe the $p_T$ dependence of the inclusive cross section, is directly related to the underlying hard parton–parton scattering for jet production, with little influence from soft physics. In addition to the a function, we introduce another function, the $F(\sqrt{s}, x_R)$ function that obeys radial scaling for inclusive jets and offers another test of the underlying parton physics. An application to heavy ion physics is given, where we use our variables to determine the transparency of cold nuclear matter to penetrating heavy mesons through the lead nucleus.

**Keywords:** inclusive cross section 1; jets 2; LHC 3; radial scaling 4

## 1. Introduction

Inclusive jet, direct photon and heavy meson cross section measurements in p–p collisions at the multi-TeV energies, up to $\sqrt{s}$ = 13 TeV of the Large Hadron Collider (LHC), afford incisive tests of the standard model. The cross sections are frequently presented as functions of the transverse momentum $p_T$ and rapidity $y$ defined by $y = \ln((E + pz)/(E - pz))/2$, with $E$ being the particle/jet total energy and $pz$ being the component of the 3-momentum along the incoming proton direction in the p–p center of momentum (COM). Over the years, both the data and the agreement of data with Monte Carlo simulations (MC) have steadily improved as higher statistics are accumulated, better fits to the parton distributions and higher-order quantum chromodynamics (QCD) terms are considered. This theoretical–experimental interplay is an active area of research. A panoply of codes has been developed to simulate inclusive jet production, such as Pythia [1] 8.2 and Sherpa 2.1.1 [2]; and for direct photons, JETPHOX [3] and POWHWG [4]. The physics of heavy flavor production in p–p collisions is adequately described by the FONLL code [5], which is a fixed-order next-to-leading-order calculation. A good summary of simulation code can be found at [6]. Experimental papers compare data with the MC simulations by superimposing the simulation on the data points and/or by plotting the ratio of data to MC to generally good agreement.

For the curious student, it is worthwhile to attempt to 'touch the physics' by searching for the underlying power laws expected from hard parton scattering through the $p_T$ and y behaviors of the inclusive cross sections even though there is good agreement between data and simulations. We find in conventional practice that the underlying physics is frequently hidden in the details of how the experimental cross sections are presented and subsequently compared with highly developed computer simulations, when in fact there may be attributes of the measured cross sections that can be more directly related

to underlying process. The most egregious example is when only the data/MC ratio is presented, in which case, the student learns that the data and MC agree to a certain level of error, but gains no knowledge of the actual shape of the $p_T$ and y dependencies of the data.

We find that the current convention of presenting the inclusive cross sections in the form $d^2\sigma/dp_T dy$ followed in publications of LHC physics complicates direct comparisons of data with the underlying physics. The measured cross section in this form has the dimensions[1] of $1/(GeV/c)^3$, which is not naturally related to the primordial hard scattering of the colliding partons whose cross sections have a dimension of $1/(GeV/c)^4$. We will show that expressing the inclusive cross sections of heavy meson and baryon production in this 'un-natural form' confuses the mass dependence of the $p_T$ dependence and hides an underlying power law.

Furthermore, the measurements with higher statistics of the $d^2\sigma/dp_T dy$ cross sections are sometimes integrated over y and presented as a function of $p_T$, or sometimes integrated over $p_T$ expressed as a function of y, resulting in a great deal of detailed dynamics of the underlying scattering processes to be obscured. As higher statistics are accumulated, it is much more revealing to present inclusive cross sections in the double differential form so that both the $p_T$ and y dependences can be studied. It is important to present cross sections in differentials of the invariant phase space form, $Ed^3\sigma/dp^3$.

Inclusive cross sections using the Lorentz-invariant phase space form have the same dimensions as the underlying hard-scattering parton–parton cross sections and are given by:

$$\frac{Ed^3\sigma}{dp^3} = \int_{x_1}^1 dx_1 \int_{x_2}^1 dx_2 G(x_1, Q) G(x_2, Q) D(z, Q) \frac{1}{\pi z} \frac{d\hat{\sigma}}{d\hat{t}} \tag{1}$$

as in Equation (4.1) of Field and Feynman [7] and in similar expressions in Field, Feynman and Fox [8], where $x_{1,2}$ are the momentum fractions of partons 1 and 2, $G(x_{1,2}, Q)$ represent the number of colliding partons between $x$ and $x + dx$ at the momentum scale $Q$, $D(z, Q)$ is the parton-to-jet/particle fragmentation function of momentum fraction z of the jet/particle to the outgoing parton and $d\hat{\sigma}/d\hat{t}$ is the primordial parton–parton elastic differential scattering cross section in the Mandelstam variable, $\hat{t}$, defined as the square of the difference of the incoming parton 4-vector minus the outgoing parton 4-vector. The invariant cross section in this form has the dimension $1/(GeV/c)^4$, which is the dimension of the underlying hard-scattering elastic cross section $d\hat{\sigma}/d\hat{t}$. Equation (1) embodies well-understood physics since the late 1970s.

In addition, recent papers on inclusive processes involving the production and decay of heavy quark states do not attempt to explicitly measure the modified transverse momentum[2], $P_T \equiv \sqrt{p_T^2 + \Lambda_m^2}$, which enables the underlying power law $p_T$ dependence to be obvious and allows for an estimation of the mass of the heavy quark state itself, including the mother–daughter relation for indirect inclusive particle production through the "$\Lambda$ term". In principle, as an added benefit, the use of this phenomenology can probe transverse structure function effects. By comparing the $\Lambda$ value for prompt heavy meson production with the $\Lambda$ value for particles that are produced through 'mother–daughter' decay, the mass of the 'mother' particle can be probed.

Again, while a seemingly trivial point of kinematics, expressing the inclusive invariant cross section in the form $Ed^3\sigma/dp^3$ dimensionally connects the data to $d\hat{\sigma}/d\hat{t}$, the underlying parton–parton hard scattering in the parton–parton center of momentum frame and therefore more directly touches the underlying causal physics.

The intent of this paper is to describe the inclusive invariant cross sections in a physically obvious manner so that the underlying physics can be easily extracted and analyzed. We use the kinematic variables $p_T$ and the radial scaling variable $x_R = E/E_{max}$, where $E$ is the energy of the detected particle or jet in the p–p COM and $E_{max}$ is its maximum value, as well as rapidity, y, and the total COM energy, $\sqrt{s}$, in undertaking this study. In our previous publications [9–11], we found that single particle/jet inclusive invariant differential cross sections can be expressed as a product of a function that strictly depends

on $p_T$ and not on the rapidity, $y$, or $x_R$, and a function which is strongly dependent on $x_R$ that is characteristic of the underlying colliding parton distributions. The foundation of this phenomenology was developed in 1976 [10] during the early days of Fermilab. Others have contributed to this analysis framework [12,13]. In this paper, we refine our previous work to show that this factorization of these two sets of kinematic variables has a broad application to jets, particles and even heavy ion collisions.

In the following, we will discuss the $p_T$ distributions and the $x_R$ distributions of various inclusive cross section measurements and relate them in a straightforward manner to the nucleon parton distribution functions (PDFs) and the underlying hard-scattering cross sections. We examine inclusive cross section measurements of various inclusive processes in p–p scattering (jets, photons and mesons and baryons) at different values of $\sqrt{s}$ as measured by several collaborations [14–17] in terms of a $[p_T - x_R]$ factorized framework. In this study, we have developed a dimensional custodial that relates the s dependence of the magnitude parameter of the $p_T$ part of the invariant cross section to the power index of its $1/p_T$ dependence. The dimensional custodial holds for inclusive jets, photons, mesons and baryons and is therefore independent of process. In addition, we will show a particularly simple description of the $x_R$ dependence that is sensitive to the underlying parton–parton scattering. Finally, we demonstrate that the modified momentum factor, $\Lambda$, for meson/baryon production is directly related to the mass of the produced meson/baryon and that the underlying $p_T$ distribution is a power law in the modified transverse momentum, $P_T$.

## 2. The Formulation

Because the published inclusive data are given in the form $d^2\sigma/dp_Tdy$, we have to convert to the invariant cross section form by computing $d^2\sigma/2\pi p_Tdp_Tdy$, where we divide the cross section by ($2\pi p_T$), with $p_T$ taken as the central value of the published $p_T$ bin. This approximates the invariant cross section $d^2\sigma/\pi dp_T^2dy$ to a ~4% error, except for the lowest and highest $p_T$ bins where the approximation is ~10%. No correction of this binning definition was made.

In a previous publication [9], we have shown that the inclusive cross sections for single jets, direct photons and light and heavy quark states, up to and including b-quark states, have the factorized form:

$$\frac{d^2\sigma}{2\pi p_Tdp_Tdy} = C(\sqrt{s}, p_T, x_R(p_T, m, y, \sqrt{s})) = A(\sqrt{s}, p_T, \Lambda_m)f(\sqrt{s}, p_T, x_R), \qquad (2)$$

where the a function depends only on $p_T$, $\Lambda_m$ and $\sqrt{s}$ and the $f$—function depends primarily on the radial scaling variable $x_R$, with $\sqrt{s}$ and $p_T$-dependent corrections. We extend the formulation of our earlier publication to express the inclusive jet invariant cross section in p–p collisions for constant $p_T$ as a polynomial in logarithms of the form:

$$\ln\left(\frac{d^2\sigma}{2\pi p_Tdp_Tdy}\right)_{p_T} = \ln(A) + n_{xR}\ln(1 - x_R) + n_{xRQ}\ln^2(1 - x_R), \qquad (3)$$

where the left-hand side is the natural logarithm of the invariant cross section for constant $p_T$ and the right-hand side is a polynomial of powers of $\ln(1 - x_R)$. Therefore, the constant $p_T$ fits of Equation (2) determine three numbers: $A(p_T)$ and the power indices $n_{xR}$ and $n_{xRQ}$.

Since $\ln[A(p_T)]$ is determined by the $x_R = 0$ intercept of Equation (3), we expect that $A$ will be dependent on only $p_T$, $\sqrt{s}$ and $\Lambda_m$ but not on $y$. Note that for finite $p_T$ and $\Lambda_m$, the $x_R \to 0$ extrapolation limit corresponds to $\sqrt{s} \to \infty$. Therefore, we posit that $A(\sqrt{s}, p_T, \Lambda_m)$ will have a direct connection to the primordial parton–parton hard scattering and their parton distribution functions that is uncomplicated by subsequent soft physics of final-state parton fragmentation and hadronization. Furthermore, we will

show that the power indices $n_{xR}$ and $n_{xRQ}$ in Equation (3) have a close connection with the underlying colliding parton distributions.

Putting all these terms together, the invariant cross section has the factorized form:

$$\frac{d^2\sigma}{2\pi p_T dp_T dy} = A(\sqrt{s}, p_T, \Lambda_m)(1 - x_R)^{n_{xR}} \exp\left(n_{xRQ} \ln^2(1 - x_R)\right). \tag{4}$$

In our previous publications [9–11], we have shown (e.g., Figure 6 of reference [9]) that the transverse momentum function, $A(\sqrt{s}, p_T, \Lambda_m)$ (called the a function), is a power law to a good approximation of the form:

$$A(\sqrt{s}, p_T, \Lambda_m) = \frac{\kappa(s)}{\left(p_T^2 + \Lambda_m^2\right)^{\frac{n_{pT}}{2}}} = \frac{\kappa(s)}{P_T^{n_{pT}}}. \tag{5}$$

The term, $\Lambda_m$ in the modified transverse momentum, $P_T \equiv \sqrt{p_T^2 + \Lambda_m^2}$, is crucial in describing low $p_T$ heavy quark production, but for inclusive jets and isolated photons the modified transverse momentum is computed with $\Lambda_m = 0$. In these cases, we use the simple form:

$$A(\sqrt{s}, p_T, \Lambda_m = 0) = \frac{\kappa(s)}{p_T^{n_{pT}}}, \tag{6}$$

where $n_{pT}$ is the $p_T$ power law index and $\kappa(s)$ is the overall magnitude of the cross section which depends on $\sqrt{s}$. Notice that $A(\sqrt{s}, p_T, \Lambda_m)$ has the dimensions of the invariant cross section [cm$^2$/(GeV/c)$^2$] or [1/(GeV/c)$^4$], thus $\kappa(s)$ has the dimensions [cm$^2$/(GeV/c)$^2$] × [(GeV/c)$^{n_{pT}}$] or [1/(GeV/c)$^4$] × [(GeV/c)$^{n_{pT}}$]. The $A(\sqrt{s}, p_T, \Lambda_m)$ parameters $\kappa$ and $n_{pT}$ in Equations (5) and (6) are positively correlated[3].

The radial scaling variable $x_R$ is defined in terms of $p_T$, $y$ and $m$ (the detected jet/particle rest mass) by:

$$x_R \equiv \frac{E}{E_{\max}} = \frac{2\sqrt{p_T^2 + m^2}}{\sqrt{s}}\cosh(y) \approx \frac{2p_T}{\sqrt{s}}\cosh(\eta), \tag{7}$$

where, in the second equation, we have expressed $x_R$ in the limit that the jet/particle mass can be neglected ($m = 0$) in terms of the pseudo-rapidity $\eta = -\ln(\tan(\theta/2))/2$, where $\theta$ is the polar angle of the jet/particle with respect to the incoming beams direction and ranges between $2m/\sqrt{s} \leq x_R \leq 1$. We will show that for heavy meson and baryon production, $\Lambda_m \sim m$. The experimental radial scaling variable is constrained $2m/\sqrt{s} \leq x_R \leq 1$, where the lower limit corresponds to $p_T = 0$ at finite $\sqrt{s}$ and the high limit of $x_R$ corresponds to the exclusive process scattering kinematic boundary that preserves quantum numbers when $E$(jet) or $E$(meson or baryon) $\sim \sqrt{s}/2$. Notice that the rapidity distinguishes between forward and backward hemispheres, whereas the $x_R$ variable is only a measure of the radial distance of the kinematic point in the COM momentum space $(p_T - p_Z)$ scaled to its maximum value corresponding to $x_R = 1$. Therefore, $x_R$ does not distinguish between hemispheres. Hence, only the value of $|y|$ can be computed from $x_R$ by the expression:

$$|y| = \ln\left[\frac{\sqrt{s}}{2}\frac{x_R}{\sqrt{p_T^2 + m^2}} + \sqrt{\frac{s}{4}\frac{x_R^2}{p_T^2 + m^2} - 1}\right]. \tag{8}$$

Having determined the a function by fitting data to Equation (3), we can extract the $x_R$ dependence with our factorization ansatz by dividing out the $p_T$ dependence embedded in $A(\sqrt{s}, p_T, \Lambda_m)$ as follows:

$$f(\sqrt{s}, p_T, x_R) = \frac{1}{A(\sqrt{s}, p_T, \Lambda)}\frac{d^2\sigma}{2\pi p_T dp_T dy} = \exp\left(n_{xR}\ln(1 - x_R) + n_{xRQ}\ln^2(1 - x_R)\right). \tag{9}$$

Notice that $f = 1$ in the limit $x_R = 0$ is built in. The $F$-function depends on $\sqrt{s}$, $p_T$ and $y$ as well as $x_R$ and in general violates radial scaling because the power indices $n_{xR}$ and $n_{xRQ}$ are not constants. However, we will show that the power indices $n_{xR}$ and $n_{xRQ}$ are for inclusive jet data have a simple dependence on $\sqrt{s}$ and $p_T$ and are represented by:

$$\begin{aligned} n_{xR}(\sqrt{s}, p_T) &= \frac{D(\sqrt{s})}{p_T} + n_{xR0} \\ n_{xRQ}(\sqrt{s}, p_T) &= \frac{D_Q(\sqrt{s})}{p_T{}^2} + n_{xRQ0}, \end{aligned} \tag{10}$$

where the distortion parameters $D$ and $D_Q$ depend on $\sqrt{s}$ and $n_{xR0}$ and $n_{xRQ0}$ are constants. Thus, the remaining $p_T$ dependence is embodied in the $D$ and $D_Q$ terms that is the origin of the violation of radial scaling—mostly at low $p_T$, whereas the larger $p_T$ region is controlled by the constant parameters $n_{xR0}$ and $n_{xRQ0}$. In fact, with the $p_T$ behavior of Equation (10), the $x_R$ sector of the invariant cross section can be written in terms of a radial scaling violating term, controlled by the distortion parameters $D$ and $D_Q$ multiplied by a scaling term Thus, Equation (9) becomes:

$$f(\sqrt{s}, p_T, x_R) = \exp\left(\frac{D}{p_T}\zeta + \frac{D_Q}{p_T^2}\zeta^2\right)\exp\left(n_{xR0}\zeta + n_{xRQ0}\zeta^2\right), \tag{11}$$

where $\zeta = \ln(1 - x_R)$. The first exponential is almost independent on $p_T$ for low $x_R$, but is dependent on y and violates radial scaling, while the second exponential, the radial scaling term, is dependent only on $x_R$ and therefore for a fixed $\sqrt{s}$ obeys radial scaling. Note that positive $D$ and $n_{xR0}$ result in decreasing $f(\sqrt{s}, p_T, x_R)$ as $x_R$ increases, whereas positive $D_Q$ and $n_{xRQ0}$ result in increasing $f(\sqrt{s}, p_T, x_R)$ as $x_R$ increases. Therefore, if we compensate for the scale violating term in Equation (11), governed by the distortion parameters $D$ and $D_Q$ we should be left with the radial scaling second exponential term determined by the constants $n_{xR0}$ and $n_{xRQ0}$.

We will test this hypothesis by calculating a data-determined correction to the radial scaling limit so that what is left is a 'kernel' radial scaling function that has no $p_T$ or $y$ dependence, little $\sqrt{s}$ dependence, but is distinctly process dependent. The kernel end-product of this calculation is:

$$F(\sqrt{s}, x_R) = R(\sqrt{s}, p_T, y)f(\sqrt{s}, p_T, x_R). \tag{12}$$

We now show how the correction function $R(\sqrt{s}, p_T, y)$ in Equation (12) is calculated. Immediately, we note by comparing Equations (11) and (12) that we have:

$$R(\sqrt{s}, p_T, y) = \exp\left(-\frac{D}{p_T}\zeta - \frac{D_Q}{p_T^2}\zeta^2\right). \tag{13}$$

We find that $R$ is slowly dependent on $p_T$ in the limit of small $x_R$ but strongly dependent on y. Later, we will find that $D \sim \sqrt{s}$ and $D_Q \sim s$ so that the magnitude of the correction is roughly independent of $\sqrt{s}$.

Having eliminated the $D$ terms in $f(\sqrt{s}, p_T, x_R)$ by the correction factor $R$ of Equation (13), we expect that the $F$-function can be represented to good approximation for the expression:

$$F(\sqrt{s}, x_R) = \exp\left(n_{xR0}\ln(1 - x_R) + n_{xRQ0}\ln^2(1 - x_R)\right), \tag{14}$$

where $n_{xR0}$ and $n_{xRQ0}$ are constants defined in Equation (10) for a fixed value of $\sqrt{s}$. Hence, at a fixed value of $\sqrt{s}$ the $F$-function obeys radial scaling—namely the function only depends on $x_R$. On the other hand, complete 'radial scaling' is the limit when the power indices $n_{xR}$ and $n_{xRQ}$ are themselves constant for all $\sqrt{s}$. In this case, all the $p_T$ and $\sqrt{s}$ dependence of the invariant cross section is in the $A(\sqrt{s}, p_T, \Lambda_m)$ function and none is in the $F$-function. In this complete scaling case, it does not matter how $x_R$ is calculated—any

set of values of $\sqrt{s}$, $y$ and $p_T$ that computes to the same $x_R$ will yield the same non-$A$ part of the factorized cross section. This complete form of scaling has been shown to be violated by QCD evolution as a function of $\sqrt{s}$ [9].

In summary, we assert that the invariant cross section for inclusive jet, direct photon or particle production ($\pi$, K, $\Lambda$, J/$\psi$, D, B, $\Upsilon$, etc.) at a given value of $\sqrt{s}$, can be factorized into three sectors: (1) a $p_T - \sqrt{s}$ sector, (2) a $y - \sqrt{s}$ sector and (3) an $x_R - \sqrt{s}$ sector where:

$$\frac{d^2\sigma}{2\pi p_T dp_T dy} = A(\sqrt{s}, p_T, \Lambda_m)Y(\sqrt{s}, y)F(\sqrt{s}, x_R), \tag{15}$$

with the functions defined as:

$$\begin{aligned}
A(\sqrt{s}, p_T, \Lambda) &= \frac{\kappa(s)}{\left(p_T^2 + \Lambda_m^2\right)^{\frac{npT}{2}}} = \frac{\kappa(s)}{P_T^{npT}}, \\
Y(\sqrt{s}, y) &= \exp\left(\frac{D}{p_T}\zeta + \frac{D_Q}{p_T^2}\zeta^2\right), \\
F(\sqrt{s}, x_R) &= \exp\left(n_{xR0}\zeta + n_{xRQ0}\zeta^2\right).
\end{aligned} \tag{16}$$

We will show that $D$, $D_Q$, $n_{xR0}$ and $n_{xRQ0}$ are functions of $\sqrt{s}$ so that complete radial scaling is broken although it holds for fixed $\sqrt{s}$. The parameter $\Lambda_m$ is only significant when $p_T \sim m$, the mass of the heavy particle. We will test the assertion of Equation (16) and will show both agreements and violations to it in what follows.

*2.1. Theoretical Underpinnings of $x_R$*

The radial scaling variable $x_R$ was introduced to control the effect of the kinematic boundary and as such was useful in comparing cross section measurements at different values of $\sqrt{s}$ and different y regions. However, there is another value in that $x_R$ provides a window into the hard scattering of the primordial parton–parton system. For now, consider the relevant variables at the parton level. The $s$ value (total energy squared) of the parton–parton center of momentum collision in terms of the colliding partons longitudinal momentum fractions $x_1$ and $x_2$ is given by: $\hat{s} = sx_1x_2$. Hence, in terms of the colliding partons, the radial scaling variable is the Lorentz invariant that can be evaluated by:

$$x_{R0} = \sqrt{\frac{\hat{s}}{s}} = \sqrt{x_1x_2} = \frac{2p_T\cosh(\eta_0)}{\sqrt{s}}, \tag{17}$$

where $\eta_0$ is the true value of rapidity in the parton–parton COM frame and $p_T$ is the scattered parton transverse momentum. The difference between the p–p COM value of $x_R$ and the exact value $x_{R0}$ given by Equation (17) arises from the fact that the p–p COM value of $\eta$ is only approximately equal to the true value of $\eta_0$ because, in general, the parton–parton COM is moving with respect to the p–p COM. Of course, there are additional resolution effects to the actual measured value of $x_R$ from the fragmentation and hadronization processes, where the outgoing parton becomes the detected jet, photon or meson/baryon—effects we are neglecting in this parton-level discussion. Continuing, the Lorentz transformation from the parton COM to the p–p COM is controlled by:

$$\beta = \frac{x_1 - x_2}{x_1 + x_2}. \tag{18}$$

Therefore, the $x_R$ resolution is determined by not knowing the event-by-event value of $\beta$, even though its average for p–p collisions is zero. The resolution smearing is computed by remembering that the pseudo-rapidity transforms as $\eta = \eta_0 + \tanh^{-1}(\beta)$, where $\eta_0$ is the value in the parton–parton COM and $\eta$ is its value in the p–p COM and is given by:

$$\frac{\Delta x_R}{x_R} = \frac{(x_R - x_{R0})}{x_{R0}} = \frac{(1 - \beta\tanh(\eta_0))}{\sqrt{1 - \beta^2}} - 1 = \frac{\sqrt{1 - \beta^2}}{(1 - \beta\tanh(\eta))} - 1. \tag{19}$$

The relation between the β-smeared ('experimental') $x_R$ and the exact $x_{R0}$ is shown in Figure 1. Notice that there are 'good' kinematic regions, such as $\eta > 0$ and $\beta > 0$ and 'bad' regions when $\eta$ and $\beta$ have opposite signs. On average, the value of the measured $x_R$ tends to be larger for large $|\eta|$ than the true value, $x_{R0}$ denoted by the blue-dashed line in the figure. The resolution grows for increasing $|\eta|$ but saturates for $|\eta| \geq 3$.

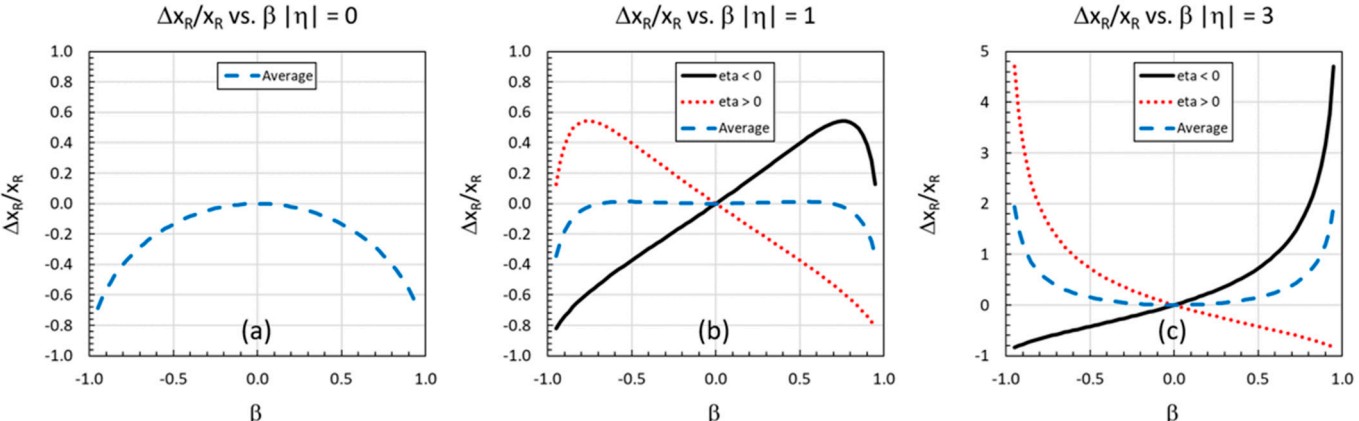

**Figure 1.** The error in $x_R$ is plotted vs. $\beta_{cm}$ for $|\eta| = 0$, $= 1$, $= 3$, in Figure 1a–c, respectively. The black solid ($\eta < 0$) and red dotted ($\eta > 0$) lines define the boundary of the error depending on the value of β and η, including its sign. The blue dashed line is the average error of both η cases as a function of β. The error is relatively small for small $|\eta|$ and grows with increasing $|\eta|$ but does not increase significantly beyond $|\eta| \geq 3$ because the tanh(η) $\to \pm 1$ for large $|\eta|$. Note that "1" on the vertical scale indicates 100% error. The data errors are constrained between the red dotted and solid black lines from this effect.

From our earlier publication [9], we find that the low $x_R$ behavior of inclusive cross sections has a $\sim (1 - x_R)^{nxR}$ behavior as in Equation (4), neglecting the $n_{xRQ}$ term. Unlike the high $p_T$ power law behavior of $A(\sqrt{s}, p_T, \Lambda_m) \sim 1/p_T^{npT}$, where the power index $n_{pT}$ is independent of scale calibration, the power index $n_{xR}$ is sensitive to both the $p_T$ and cosh(y) (cosh(η)) scales. Considering a putative change of scale of the form $x_R' = \zeta x_R$, which could be due to resolution errors in $p_T$ or $y$ or from fragmentation and hadronization following the hard parton–parton scattering, we find that for small $x_R$, the power index $n_{xR}$ is changed by:

$$nx_R' = nx_R \frac{\ln(1 - x_R)}{\ln(1 - \zeta x_R)} \approx nx_R/\zeta. \tag{20}$$

Hence, the power index $n_{xR}$ of the $(1 - x_R)$ distribution is sensitive to scale and is therefore a more stringent test of theory, especially parton fragmentation and hadronization, than the $p_T$ distribution measured by the a function.

Note (obviously) that in the case of pure dijets, the complete kinematics can be determined if both jets are measured. In this case, again neglecting the jet mass and any energy loss through fragmentation and hadronization, the exact value of $x_R$ is given by:

$$x_{R0} = \frac{2p_T \cosh\left(\frac{\eta_2 - \eta_1}{2}\right)}{\sqrt{s}}, \tag{21}$$

and in the case of heavy quarks where the quark mass cannot be neglected by:

$$x_{R0} = \frac{\sqrt{2(p_T^2 + m^2)[1 + \cosh(y_1 - y_2)]}}{\sqrt{s}}. \tag{22}$$

Similar expressions have been worked out by Feynman, Field and Fox some time ago [8]. In summary, $x_R$ provides a direct view of the underlying parton distributions with

an error that depends on the pseudo-rapidity and the unmeasured Lorentz factor β of the parton–parton COM.

### 2.2. Analysis of ATLAS Jets √s = 13 TeV

As described in our previous publication [9], the a function and the $x_R$ power indices are determined by the analysis of the invariant cross section for fixed $p_T$ extrapolated to $x_R = 0$. The parameters of the extrapolation are the power indices $n_{xR}$ and $n_{xRQ}$ of Equation (3) and the endpoint of the extrapolation is the value of the a function for that value of $p_T$. Namely, for fixed $p_T$, $\sqrt{s}$ and $\Lambda_m$, the a function value is determined by:

$$A(\sqrt{s}, p_T, \Lambda_m) \equiv \lim_{x \to 0} \left( \frac{d^2\sigma}{2\pi p_T dp_T dy} \right)_{p_T}. \tag{23}$$

An example of this analysis for inclusive jets at R = 0.4 and $\sqrt{s}$ = 13 TeV, measured by the ATLAS collaboration [17], for a few selected values of $p_T$ is shown in Figure 2 below. We have assigned errors for each data point as the sum of statistical and systematic errors added in quadrature. We have neglected the overall normalization error associated with the uncertainty of the luminosity (2.1%). In the construction of the evaluation of $x_R$, we have made a small jet mass correction since the ATLAS jet data were presented as a function of fixed y. We estimate this mass term [18] in the definition for $x_R$ by the expression:

$$x_R = 2p_T \sqrt{1 + (m_{jet}/p_T)^2} \cosh(y)/\sqrt{s} = 2p_T \sqrt{1 + (R/\sqrt{2})^2} \cosh(y)/\sqrt{s}, \tag{24}$$

where $R = (\Delta\phi^2 + \eta^2)^{1/2}$. We correct the $x_R$ value as shown but set $\Lambda_m = 0$ since this small mass correction (~3.8%) [9] has a neglectable effect on the power law fits to the $A(p_T)$ function.

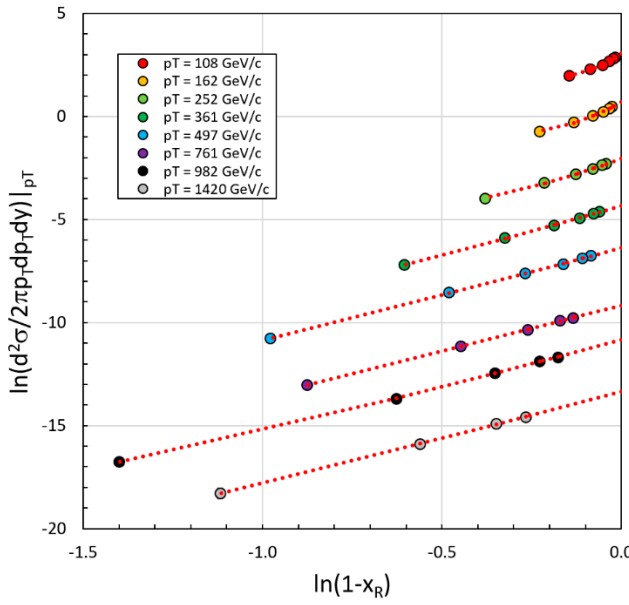

**Figure 2.** Demonstration of the ansatz of Equation (3) for some selected values of $p_T$ in GeV/c for inclusive jets defined by the anti-k$_T$ algorithm, with R = 0.4 measured by the ATLAS collaboration at $\sqrt{s}$ = 13 TeV. The plot demonstrates that the log of the invariant cross section at constant $p_T$ is a quadratic in $\ln(1 - x_R)$. The error bars are smaller than the data points. The red dotted lines indicate minimum $\chi^2$ fits to Equation (3). The extrapolation to $\ln(1 - x_R) = 0$ ($x_R = 0$) determines $\ln[A(p_T)]$. The right-most point of each constant $p_T$ line corresponds to y = 0 and the gap between this point and right-hand axis is the region beyond the kinematic boundary for given value of $p_T$ and $\sqrt{s}$. For the ensemble of fits at constant $p_T$, the $\chi^2$/d.f. = 14 for 79 degrees of freedom (p = 1.0).

Having determined the values for $A(p_T)$, $n_{xR}$ and $n_{xRQ}$ for each value of $p_T$, the entire inclusive cross section can now be described. The resultant $A(p_T)$ for 13 TeV ATLAS [17] and CMS [19] inclusive jets for R = 0.4 for jets determined by the anti-$k_T$ algorithm [20] is plotted in Figure 3.

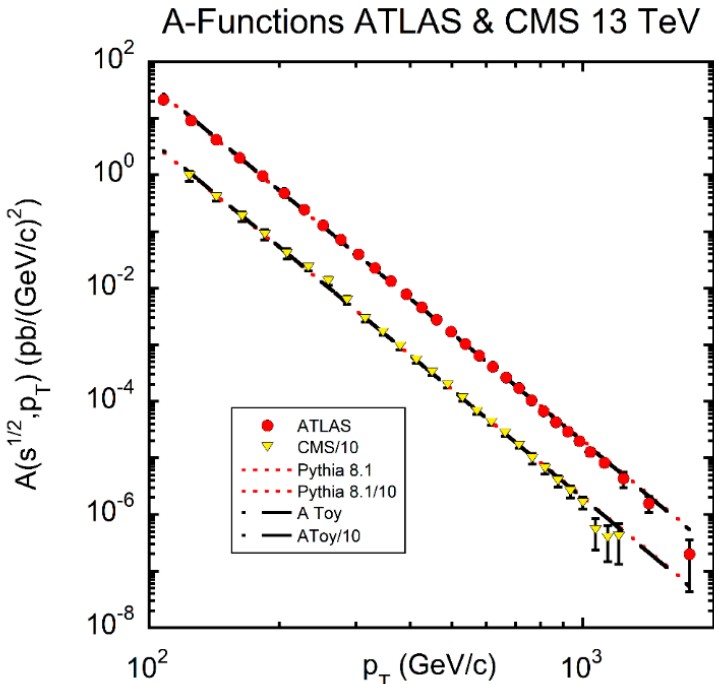

**Figure 3.** The a function $p_T$ dependence for a 13 TeV inclusive jet cross section (R = 0.4) measured by ATLAS (red circles) and CMS (yellow triangles) divided by 10. All data point errors—statistical and systematic—were added in quadrature. The dashed black lines represent power law fits of the Toy MC. The Pythia 8.1 simulations are indicated by the red dotted lines. Both simulations were normalized to data. The MC representations of the data are indistinguishable on this plot. The respective power law fits for 29 degrees of freedom (d.f.) for ATLAS are: fitting the data to a power law $\chi^2/$d.f. = 1.13, ($p$ = 0.288), Pythia 8.1 power law fit to data $\chi^2/$d.f. = 1.24 ($p$ = 0.18) and Toy power law fit to data $\chi^2/$d.f. = 1.13 ($p$ = 0.288). For CMS 25 d.f. the fit qualities are $\chi^2/$d.f. = 0.60 ($p$ = 0.94), 0.76 ($p$ = 0.80) and 0.66 ($p$ = 0.90), for data, Pythia 8.1 and Toy, respectively. The parameter $\Lambda_m$ in Equation (5) was set to 0.

As noted above, in addition to determining the $A$ value, the extrapolation to $x_R = 0$ also determines the power index parameters $n_{xR}$ and $n_{xRQ}$. These are shown in Figure 4.

Following the procedure embodied in Equation (12), we determine the $F(\sqrt{s}, x_R)$ function for 13 TeV ATLAS jets. In the calculation, it is important to use the actual a function values rather than its power law fit values since the small ($\sim \pm 30\%$) deviations from the pure power law over 8 orders of magnitude are critical. The result is shown in Figure 5 and the correction function given by Equation (13) is plotted in Figure 6. Note that the correction function is almost independent of $p_T$ for $|y| \le 1.5$, corresponding to low $x_R$ because, to a good approximation, $n_{xR} \sim 1/p_T$ and $n_{xRQ} \sim 1/p_T^2$, cancelling the $p_T$ dependence of $\ln(1 - x_R) \approx -x_R$ and $\ln(1 - x_R)^2 \approx x_R^2$, respectively.

Our formulation of inclusive jet production at the LHC at fixed $\sqrt{s}$ employs only six parameters ($\kappa$, $n_{pT}$, $D$, $n_{xR0}$, $D_Q$, and $n_{xRQ0}$) for a complete description of jet invariant differential cross sections—the a function characterizes the $p_T$ dependence in the limit $x_R \to 0$, the $F$-function describes the $x_R$ dependence at $y = 0$ and the $D$ and $D_Q$ terms track the scaling violation. Important corrections to the generation of the $F$-function are embodied in the $D$ and $D_Q$ terms which are related to the QCD evolution of the colliding parton PDFs. We summarize the results of fitting ATLAS and CMS 13 TeV R = 0.4 inclusive jets in Table 1. We note that the two data sets agree within about one standard deviation.

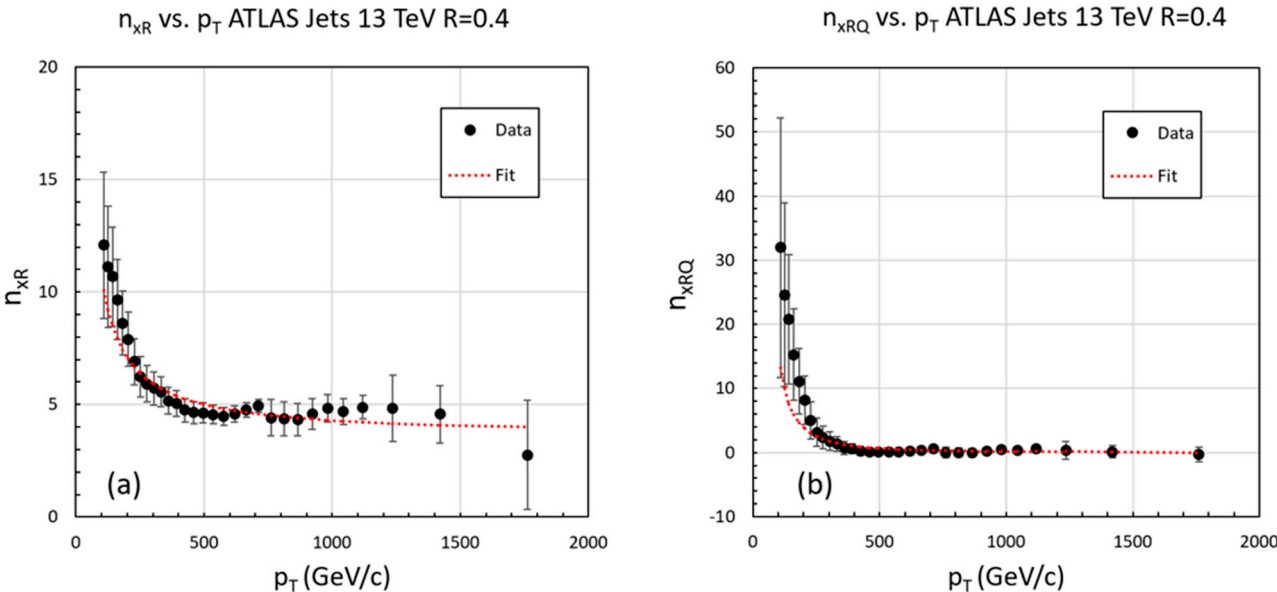

**Figure 4.** The $x_R$ power indices $n_{xR}$ (**a**) and $n_{xRQ}$ (**b**) are plotted as a function of $p_T$. The red dotted curves are the result of minimum $\chi^2$ fits to Equation (10). We find $D = (7.0 \pm 1.1) \times 10^2$ GeV/c, $n_{xR0} = 3.6 \pm 0.2$ with $\chi^2 = 14$ for 29 degrees of freedom ($p = 0.99$) and $D_Q = (1.5 \pm 0.4) \times 10^5$ (GeV/c)$^2$ and $n_{xRQ0} = 0.06 \pm 0.1$ with $\chi^2 = 24$ for 29 degrees of freedom ($p = 0.73$). While the fits have good $\chi^2$ values, they systematically underestimate the values of $n_{xR}$ and $n_{xRQ0}$ for $p_T \leq 200$ GeV/c.

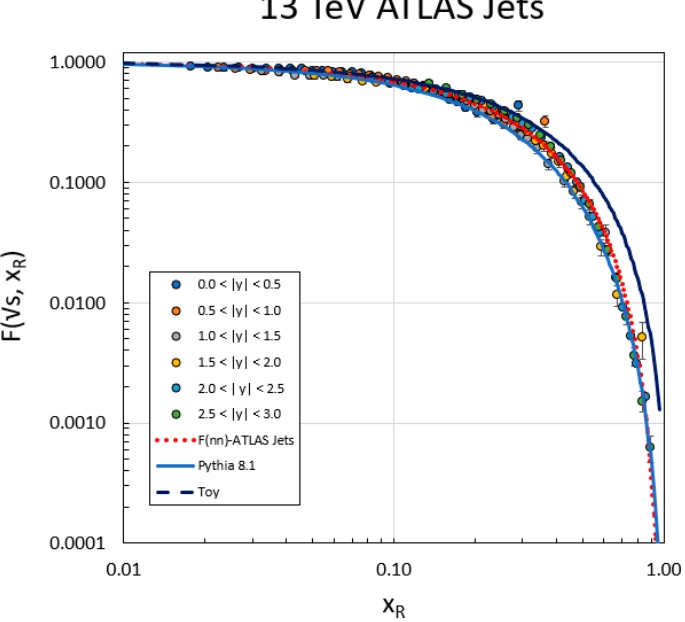

**Figure 5.** The $F(\sqrt{s}, x_R)$ for 13 TeV inclusive jets (R = 0.4) is plotted as a function of $x_R$ for various slices of $< |y| >$ indicated by the numbers in the legend. Note that all the data points at different y values fall on the same line and that the red dotted line represents Equation (14). The error bars represent the systematic and statistical errors added in quadrature. The fit of data by Equation (14) has a $\chi^2/\text{d.f.} = 1.06$ for 170 degrees of freedom ($p = 0.28$). The solid blue line represents Pythia 8.1 simulation ($\chi^2/\text{d.f.} = 3.5$, $p \sim 0$)) and the black line the prediction of the Toy MC ($\chi^2/\text{d.f.} = 36.6$ $p \sim 0$).

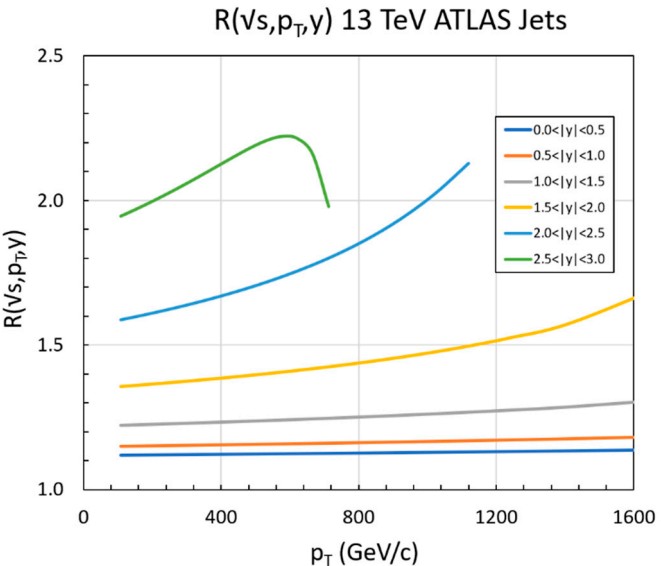

**Figure 6.** The $R(\sqrt{s}, p_T, y)$ correction function for 13 TeV inclusive jets determined by the measured values of $D$ and $D_Q$ is plotted as a function of $p_T$ for various values of $|y|$ given by Equation (13).

**Table 1.** Fit parameters of 13 TeV ATLAS and CMS jets. The parameters $\kappa$ and $n_{pT}$ describe the a function, and the parameters $D$, $n_{xR0}$, $D_Q$, and $n_{xRQ0}$ describe the $F$-function. We note that the ATLAS and CMS jet parameters are consistent within errors. In the analysis of both data sets, we have required $|y| \leq 3$ and have added systematic and statistical errors in quadrature. The overall normalization error in the luminosities determinations was neglected. The CERN MINUIT [21] fitting package was used. $D$ and $D_Q$ are correlated as approximately $0.5D \sim (D_Q)^{1/2}$.

| Experiment | Parameter | Value | $\chi^2$/d.f. | *p*-Value |
|---|---|---|---|---|
| ATLAS | $\kappa n_{pT}$ | $(2.1 \pm 0.3) \times 10^{14}$ pb/(GeV/c)$^{(2\text{-}n_{pT})}$ <br> $6.35 \pm 0.02$ | 33/29 | 0.28 |
| CMS | $\kappa n_{pT}$ | $(3.1 \pm 1.0) \times 10^{14}$ pb/(GeV/c)$^{(2\text{-}n_{pT})}$ <br> $6.41 \pm 0.05$ | 15/25 | 0.94 |
| ATLAS | $D$ <br> $n_{xR0}$ | $(7.0 \pm 1.1) \times 10^2$ GeV/c <br> $3.6 \pm 0.2$ | 14/29 | 0.99 |
| CMS | $D$ <br> $n_{xR0}$ | $(7.5 \pm 3.1) \times 10^2$ GeV/c <br> $3.3 \pm 0.6$ | 7/25 | 1.00 |
| ATLAS | $D_Q$ <br> $n_{xRQ0}$ | $(1.5 \pm 0.4) \times 10^5$ (GeV/c)$^2$ <br> $0.06 \pm 0.1$ | 24/29 | 0.73 |
| CMS | $D_Q$ <br> $n_{xRQ0}$ | $(2.0 \pm 1.3) \times 10^5$ (GeV/c)$^2$ <br> $0.08 \pm 0.4$ | 8/25 | 1.00 |

The ATLAS 13 TeV jet data [17] have an approximately 3% jet energy scale (JES) error. By using the Toy MC, to be described later, we find that, for a +3% JES change (jet energy measured to be larger than the actual energy), the a function parameter, $n_{pT}$, changes by only $-0.2\%$, whereas the parameters of the $[x_R - p_T - \sqrt{s}]$ sector are much more sensitive. For the same +3% JES increase, we find that $D$ changes by +6%, $n_{xR0}$ by $-5\%$, $D_Q$ by +7% and $n_{xRQ0}$ by $-4\%$. The magnitude parameter of the a function, $\kappa$, changes by +15% and is therefore quite sensitive to the JES. The $\pm$ signs indicate change of parameter, either increasing (+) or decreasing ($-$), when JES increased by +3%.

In summary, we have shown that inclusive jet production at $\sqrt{s} = 13$ TeV can be described with six parameters ($\kappa$, $n_{pT}$, $D$, $n_{xR0}$, $D_Q$, and $n_{xRQ0}$). The terms $D$ and $D_Q$

characterize the radial scaling violation and the parameters $n_{xR0}$ and $n_{xRQ0}$ determine the radial scaling term at a constant value of $\sqrt{s}$. In the next section we describe a Toy MC simulation that provides an intuitive physical picture of inclusive jet production.

## 3. Jet Simulations: Toy Model and Pythia 8.1

In order to gain a deeper understanding of how the $p_T$ and s dependences arise, we wrote a 'Toy' Monte Carlo (TMC) simulation in ROOT [22] that computes parton–parton elastic scattering weighted by the PDFs of the proton given by CT10 parameterization [23]. We take the scattered partons within $p_T$ and $\eta$ acceptance to approximate the jet as measured inclusively by ATLAS and CMS. A similar procedure is followed to simulate the detected photon in inclusive direct photon measurements. The program does not simulate any quark or gluon fragmentation or any "soft physics" of jet formation. In the simulation of inclusive jets, all events are dijets. The hard-scattering cross sections in the simulation are given in Owens, Reya and Gluck [24] and in the review by Owens [25]. The QCD evolution of the strong coupling constant $\alpha_s(Q)$ was parameterized by a fit to the PDG values [26] of the form $1/\alpha_s(Q) = 1.2104 \ln(Q) + 2.8827$ with Q in GeV/c resulting in $\alpha_s(Q)$ ~ 0.12 at $Q = M_z$.

For a more complete comparison with data, we deployed the HepSim Pythia 8.1 simulations [27] of inclusive jets with a jet radius R = 0.4 defined by the anti-$k_t$ algorithm [20] for COM energies through the LHC range even up to $\sqrt{s} = 100$ TeV, in order to check that s-dependent systematics continue to very high energies. The Pythia 8.1 MC "data" were analyzed in the same manner as described in [9]. However, for intuitive guidance, we find comparisons with the TMC to be useful.

### 3.1. Toy Model

The governing equations of our toy model are specified by the following. The s value (total energy squared) of the parton–parton center of momentum collision in terms of the colliding partons longitudinal momentum fractions $x_1$ and $x_2$ is given by:

$$\hat{s} = sx_1x_2, \tag{25}$$

where $x_1$ and $x_2$ are the momentum fractions of the colliding partons with respect to the incoming beam momenta. (For simplicity in notation in the equations to follow we have dropped the caret notation.) The Lorentz transformation $\beta$ value of the parton–parton COM is given by:

$$\beta = \frac{x_1 - x_2}{x_1 + x_2}. \tag{26}$$

The Mandelstam variables are for the parton–parton elastic scattering given by:

$$\begin{aligned} t &= -\tfrac{s}{2}(1 - \cos\theta) \\ u &= -\tfrac{s}{2}(1 + \cos\theta), \end{aligned} \tag{27}$$

where $\theta$ is the COM angle of the outgoing struck parton ($-1 \leq \cos\theta \leq 1$) with respect to the beam direction. Note that outgoing parton transverse momentum is $p_T^2 = ut/s$.

For example, in terms of these variables, the gluon elastic scattering cross section ($gg \rightarrow gg$) is given by:

$$\frac{d\sigma}{dt} = \frac{\pi\alpha_s^2}{s^2}\frac{9}{2}\left(3 - \frac{tu}{s^2} - \frac{su}{t^2} - \frac{st}{u^2}\right), \tag{28}$$

where $\alpha_s$ is the strong coupling constant and $s$, $t$ and $u$ are the Mandelstam variables in the parton–parton COM defined above. The cross section can be expressed in terms the scattered gluon transverse momentum, $p_T$, and s and is given by:

$$\frac{d\sigma}{dt} = \pi\alpha_s^2 \frac{9}{2}\left(\frac{1}{p_T^4} + \frac{3}{s^2} - \frac{p_T^2}{s^3} - \frac{3}{sp_T^2}\right). \tag{29}$$

In the limit of $p_T \ll \sqrt{\hat{s}}/2$, the cross section becomes s-independent. In that limit, the leading term is $\sim 1/p_T^4$. On the other hand, when $p_T = \sqrt{\hat{s}}/2$, corresponding to $\sin\theta = 1$ at the kinematic maximum ($\theta = \pi/2$, $t = u = -s/2$) the $g\,g \to g\,g$ elastic scattering cross section has the finite value of:

$$\frac{d\sigma}{dt} = \pi\alpha_s^2 \frac{243}{8}\left(\frac{1}{s^2}\right) = \pi\alpha_s^2 \frac{243}{128}\left(\frac{1}{p_T^4}\right). \tag{30}$$

A similar analysis can be performed for the other hard-scattering cross sections. These are tabulated in Table 2 below where we list the leading term and the value of the cross section at $p_T$ maximum.

**Table 2.** The cross sections for each hard-scattering process in jet production are listed showing the leading $p_T$ behavior at small $p_T$ and the values of the cross sections at the kinematic limit when $p_T = \sqrt{\hat{s}}/2$. The fractional coefficients result from the various color factors of the parton–parton interactions.

| Process | Leading $p_T$ Behavior | Value at $p_T = \sqrt{\hat{s}}/2$ |
|---|---|---|
| $gg \to gg$ | $\frac{d\sigma}{dt} \approx \pi\alpha_s^2 \frac{9}{2}\left(\frac{1}{p_T^4}\right)$ | $\frac{d\sigma}{dt} = \pi\alpha_s^2 \frac{243}{128}\left(\frac{1}{p_T^4}\right)$ |
| $gq \to gq$ $g\bar{q} \to g\bar{q}$ | $\frac{d\sigma}{dt} \approx 2\pi\alpha_s^2\left(\frac{1}{p_T^4}\right)$ | $\frac{d\sigma}{dt} = \pi\alpha_s^2 \frac{55}{144}\left(\frac{1}{p_T^4}\right)$ |
| $q\,q \to q\,q$ $\bar{q}\,\bar{q} \to \bar{q}\,\bar{q}$ | $\frac{d\sigma}{dt} \approx \pi\alpha_s^2 \frac{8}{9}\left(\frac{1}{p_T^4}\right)$ | $\frac{d\sigma}{dt} = \pi\alpha_s^2 \frac{11}{54}\left(\frac{1}{p_T^4}\right)$ |
| $q_a q_b \to q_a q_b$ $\bar{q}_a\bar{q}_b \to \bar{q}_a\bar{q}_b$ | $\frac{d\sigma}{dt} \approx \pi\alpha_s^2 \frac{8}{9}\left(\frac{1}{p_T^4}\right)$ | $\frac{d\sigma}{dt} = \pi\alpha_s^2 \frac{5}{36}\left(\frac{1}{p_T^4}\right)$ |
| $q\bar{q} \to gg$ | $\frac{d\sigma}{dt} \approx \pi\alpha_s^2 \frac{32}{27s}\left(\frac{1}{p_T^2}\right)$ | $\frac{d\sigma}{dt} = \pi\alpha_s^2 \frac{7}{108}\left(\frac{1}{p_T^4}\right)$ |
| $q\bar{q} \to q\bar{q}$ | $\frac{d\sigma}{dt} \approx \pi\alpha_s^2 \frac{8}{9}\left(\frac{1}{p_T^4}\right)$ | $\frac{d\sigma}{dt} = \pi\alpha_s^2 \frac{35}{216}\left(\frac{1}{p_T^4}\right)$ |
| $q_a\bar{q}_a \to q_b\bar{q}_b$ | $\frac{d\sigma}{dt} = \pi\alpha_s^2 \frac{4}{9s^2}\left(1 - \frac{2p_T^2}{s}\right)$ | $\frac{d\sigma}{dt} = \pi\alpha_s^2 \frac{1}{72}\left(\frac{1}{p_T^4}\right)$ |
| $gg \to q\bar{q}$ | $\frac{d\sigma}{dt} \approx \pi\alpha_s^2 \frac{1}{6s}\left(\frac{1}{p_T^2}\right)$ | $\frac{d\sigma}{dt} = \pi\alpha_s^2 \frac{7}{768}\left(\frac{1}{p_T^4}\right)$ |

There are three features of the parton–parton scattering equations that are relevant. The first is that the dominant hard-scattering processes have a $1/p_T^4$ behavior but those involving s-channel exchanges, such as $gg \to q\bar{q}$, have a $1/p_T^2$ behavior for fixed s in leading order or, in the case of $q_a\bar{q}_a \to q_b\bar{q}_b$, essentially flat in $p_T$ for constant s. In these channels, the cross sections are suppressed by a power of $1/s$. There is a slow additional $p_T$ dependence through the QCD evolution of the coupling constant $\alpha_s(Q^2)^2$. The second feature is the finite value of the cross sections at the kinematic limit when $p_T = \sqrt{\hat{s}}/2$. Additionally, the third feature is that for the t-channel exchanges, such as $g\,g \to g\,g$, the cross sections at low $p_T$ at small angles are independent of $\sqrt{\hat{s}}$.

All of the processes in Table 2 were considered in exact form, such as given in Equation (23) for $gg \to gg$ scattering, in our Toy MC program and are added in appropriate weight to simulate the jet $p_T$ spectrum as measured at ATLAS and CMS at the LHC. The PDFs were taken from the CT10 [23] fits which we parameterized at each μ value by an eighth-order polynomial of the natural logarithm of the PDF as a function of $\ln(\ln(1/x))$ in

the interval $1 \times 10^{-5} \leq x \leq 0.988$. This parameterization was motivated by the observation that the log of the gluon PDF, ln(xG(x)), is approximately linear in the double log ln(ln(1/x)). Hence, the higher-order terms of the fit are small perturbations about this dominant linear dependence. The parameterizations are accurate to a fraction of a percent except at very high x where the accuracy is a few percent even for the quark PDFs where the log–log approximation is less exact.

In the simulations, we have generally taken $\mu \sim \sqrt{s}$ for the PDF shapes and the $\alpha_s(Q^2)$ renormalization scale as either $Q \sim p_T$ or $Q \sim \sqrt{\hat{s}} = \sqrt{s x_1 x_2}$. Since we use our Toy MC to give rough physics guidance and not precision tests of QCD, our results are not strongly dependent on our particular choices of scale. For jets, we take all parton masses to be zero. Thus, the rapidity, $y$, and pseudo-rapidity, $\eta$ are equal. For the simulation of inclusive $B^{0,\pm}$ and Z-boson productions, for example, we do account for quark/boson masses and distinguish y from $\eta$.

As mentioned, the Toy MC does not account for the 'soft physics' of jet formation involving gluon and quark fragmentation governed by Sudakov form factors and subsequent hadronization, nor does it include NLO and higher-order evolution of $\alpha_s(Q^2)$. This neglect may seem alarmingly incomplete, but for the fact that a power law followed by the underlying $p_T$ distribution is manifestly independent of scale factors and quite insensitive to fragmentation and parton splitting[4]. Further, since the a function is determined by the limit $x_R \rightarrow 0$, it essentially avoids the 'soft physics' operative at finite $x_R$. This insensitivity to 'soft physics' is one of the main utilities of the a function.

Inclusive jet production is a sum over several channels of hard scattering in addition to the dominant $g\,g \rightarrow g\,g$ term. Using the Toy MC at $\sqrt{s}$ = 13 TeV, we studied the $p_T$ distribution for each hard-scattering channel (and the corresponding antiquark ones) listed in Table 2. We generated Monte Carlo data samples and analyzed them in the same manner as we did for data in order to determine the $p_T$ dependence of the cross section characterized by the function $A(\sqrt{s}, p_T)$ and the power of $(1 - x_R)$ for each constant $p_T$. The cross sections, given by the sum:

$$\sigma = \sum_{i,j} \left( \frac{d^2\sigma}{dp_T dy} \Delta p_T(i) \Delta y(j) \right) \tag{31}$$

of each process for 106 GeV/c $\leq p_T \leq$ 1423 GeV/c, $|y| \leq 3$, are shown in Table 3 normalized to the total of all process. Additionally, tabulated are the a function $p_T$ power indices, $n_{pT}$ for different production channels. We find that the $\Lambda_m$ term is unnecessary at the high $p_T$ values where $p_T \gg m_{jet}$.

### 3.2. The Power Law Indices

As expected, processes involving gluon–gluon, gluon–antiquark and antiquark–antiquark interactions have the larger $n_{pT}$ and $n_{xR}$ corresponding to the steeper shape of their respective PDFs, whereas those involving quark–quark scattering have the smaller values. The overall jet production is dominated by gluon–gluon elastic scattering with that process at $\sqrt{s}$ = 13 TeV making up 66% of the total inclusive jet cross section in our Toy MC simulations. The average value of $n_{pT}$ varies only $\pm13\%$ over the various processes listed in the table.

From Table 3, we note in detail that jets at 13 TeV are dominated by $gg \rightarrow gg$ scattering. The power indices, $n_{pT}$, are concentrated at approximately ~6. Those processes involving gluons and antiquarks have larger power indices correlating with their steeper PDF x dependences than those involving quarks, such as quark–quark elastic scattering, which has the smallest index driven by the flatter parton x distribution. Gluon–gluon elastic scattering has the largest (steepest) power index. The power law index $n_{pT}$, while varying somewhat between different hard-scattering processes, has a weighted average value that is quite close to the ATLAS and CMS data. Hence, the very simple Toy MC correctly predicts $A(p_T) \sim 1/p_T^6$, which we note is far from the dimensional limit $1/p_T^4$ as we observed

in Figure 3. The invariant cross section, however, has the dimensions of $\text{pb}/(\text{GeV/c})^2$ or $1/(\text{GeV/c})^4$, whereas $A(p_T) \sim 1/p_T^6$ at fixed $\sqrt{s}$. This presents a puzzle as to what corrects for this extra power $\sim 1/(\text{GeV/c})^2$. Later, we will show that the s dependence of $\kappa(s)$ acts as a dimensional custodian, thereby insuring the invariant cross section has the correct dimensions.

**Table 3.** The power law indices, $n_{pT}$ of $A(\sqrt{s}, p_T)$ with $\Lambda_m \equiv 0$, given in Equation (3) are tabulated for the Pythia 8.1 simulation of 13 TeV jets (R = 0.4) and for our Toy MC simulation broken down for each hard-scattering process listed in Table 2. The values of $n_{pT}$ in the Toy MC were determined from power law fits $106\,\text{GeV/c} \leq p_T \leq 1423\,\text{GeV/c}$, roughly matching ATLAS data. Note that processes involving gluons and antiquarks have a larger power index than those involving quarks as would be expected from their respective PDF shapes. The power indices are constrained at $5.3 \leq n_{pT} \leq 6.7$. The cross section ratios for the various subprocesses to total are given in the second column. The total cross section is dominated by $gg \to gg$ and $gq \to gq$ scatterings (66% and 13% of total, respectively).

| Process | $\sigma/\sigma$ (all) | $n_{pT}$ |
|---|---|---|
| ATLAS | 1 | $6.35 \pm 0.02$ |
| CMS | 1 | $6.41 \pm 0.05$ |
| Pythia 8.1 | 1 | $6.31 \pm 0.01$ |
| All Toy | 100% | $6.35 \pm 0.02$ |
| $gg \to gg$ | 66.20% | $6.76 \pm 0.03$ |
| $gq \to gq$ | 13.09% | $6.09 \pm 0.02$ |
| $qq \to qq$ | 5.95% | $5.43 \pm 0.03$ |
| $q_a q_b \to q_a q_b$ | 3.27% | $5.33 \pm 0.02$ |
| $q\bar{q} \to gg$ | 0.54% | $5.85 \pm 0.03$ |
| $q\bar{q} \to q\bar{q}$ | 1.98% | $6.03 \pm 0.03$ |
| $gg \to q\bar{q}$ | 1.30% | $6.66 \pm 0.03$ |
| $q_a\bar{q}_a \to q_b\bar{q}_b$ | 0.07% | $5.54 \pm 0.02$ |
| $\bar{q}\,\bar{q} \to \bar{q}\,\bar{q}$ | 1.43% | $6.62 \pm 0.04$ |
| $\bar{q}_a\bar{q}_a \to \bar{q}_b\bar{q}_b$ | 0.81% | $6.54 \pm 0.04$ |
| $g\bar{q} \to g\bar{q}$ | 5.37% | $6.64 \pm 0.03$ |

The a function is directly controlled by the energy in the parton–parton COM, $\sqrt{\hat{s}} = \sqrt{sx_1x_2}$ which fixes the maximum $p_T$ for that particular parton–parton scattering and therefore the entire $p_T$ spectrum for the collision. Hence, the morphing of the underlying hard $\sim 1/p_T^4$ parton–parton scattering cross sections shown in Table 2 to the observed and Monte Carlo-simulated behavior of $\sim 1/p_T^6$ has a simple explanation. Noting that the low $p_T$ behavior of the elastic scattering cross section has little s dependence, as demonstrated by Equation (29) and shown in Table 2, and that the cross section is finite at the kinematic limit $p_T = \sqrt{\hat{s}}/2$, the observed $p_T$ spectrum can be thought of a sum of overlapping, power law-segments each following the power law $\sim 1/p_T^4$ independent of s, at the experimentally chosen minimum $p_T$ stretching out to the kinematic maximum of $p_T = \sqrt{\hat{s}}/2$. Each line segment has an amplitude given by the cross sections of the table above and contributes to the overall $p_T$ distribution by the weighting of the $\sqrt{\hat{s}} = \sqrt{sx_1x_2}$—distribution determined by the colliding parton PDFs.

Hence, there are two major factors that determine $n_{pT}$ of the $A(\sqrt{s}, p_T)$ power law: (1) the underlying hard-scattering $p_T$ dependence of $d\hat{\sigma}/d\hat{t}$ given in Table 2, and (2) the parton $x$ distribution that determines the $\hat{s} = sx_1x_2$ distribution, which is dominated by the gluon distribution in $g\,g \to g\,g$ scattering at high energies. There are a third and fourth effect present: (3) the QCD evolution of the parton distribution functions as $\sqrt{s}$ increases (especially at $x \leq 10^{-4}$) and (4) the running of $\alpha_s(Q^2)$ as the $Q^2$-scale changes. However, at the LHC energies, the factors (3) and (4) are growing smaller as s increases and their influence on the a function are dominated by the first two effects.

The parton distribution determines the $\hat{s}$ distribution through Equation (25). For inclusive jet production, it is the very low-x behavior of the gluon distribution that most strongly affects the power law of $A(\sqrt{s}, p_T)$. The value of x has to satisfy $x \geq 4p_T^2/s \approx 2.4 \times 10^{-4}$

for the ATLAS 13 TeV inclusive jet data where the minimum jet $p_T \approx 100$ GeV/c. No $2 \to 3$ scattering is necessary as implied in our earlier publication [9]—just the underlying hard scattering and the parton distributions are needed. Our unsophisticated Toy MC simulates this behavior quite well.

The hard scattering of partons to produce inclusive jets and particles is very well known and has been understood since the early days of the quark-parton model [7,8]. What is new is that the a function developed here is a particularly simple measure of the underlying hard-scattering physics. The data, Pythia 8.1, and the toy model including all channels are well represented by Equation (10). Those involving gluons and antiquarks have larger values of $D$ and $D_Q$ whereas those involving quarks have smaller $D$ and $D_Q$ values because they have less steeply falling PDFs with increasing x. The later processes are less well represented by Equation (10).

The distortion "$D$" and "$D_Q$" terms are quite descriptive of the inclusive cross section and have a strong dependence on the low-x behavior of the colliding partons as shown in Table 4, but are also influenced by the sampling of the cross section along lines of constant $|\eta|$ ($|y|$) and reflects the $|\eta|_{max}$ and $|\eta|_{min}$ constraints in the $x_R - p_T$ plane. As a consequence, these constraints have to be accounted for in comparing the $x_R$ distributions of different experiments that have different $\eta$ acceptance regions. However, in the table, we have fixed $|\eta| \leq 3$.

**Table 4.** The power law index of $[p_T - x_R]$ sector given in Equation (9) are tabulated for Pythia 8.1 and our Toy MC simulation for each hard-scattering process at $\sqrt{s}$ = 13 TeV. Notice that the parameters are strongly dependent on the hard-scattering process and the underlying PDFs. Additionally, note that $n_{xR0}$ is strongly correlated with $n_{xRQ0}$. The ATLAS data for $D$ and $D_Q$ fall between the Pythia 8.1 and the Toy MC. The ATLAS, Pythia 8.1 and All Toy fits were performed by a minimum $\chi^2$ fit with MINUIT. The subprocesses were fit with linear regression (LR)—which does not minimize $\chi^2$ but does go through the points. The errors quoted for these processes are those of the LR.

| Process | $D$ (GeV/c) | $n_{xR0}$ | $D_Q$ (GeV/c)$^2$ | $n_{xRQ0}$ |
|---|---|---|---|---|
| ATLAS | $700 \pm 110$ | $3.6 \pm 0.2$ | $(1.5 \pm 0.4) \times 10^5$ | $0.1 \pm 0.1$ |
| CMS | $750 \pm 307$ | $3.3 \pm 0.6$ | $(2.0 \pm 1.3) \times 10^5$ | $0.1 \pm 0.4$ |
| Pythia 8.1 | $322 \pm 30$ | $4.3 \pm 0.1$ | $(6.5 \pm 1.3) \times 10^4$ | $0.4 \pm 0.04$ |
| All Toy | $1170 \pm 92$ | $3.1 \pm 0.2$ | $(2.0 \pm 0.3) \times 10^5$ | $0.4 \pm 0.1$ |
| $gg \to gg$ | $969 \pm 17$ | $7.1 \pm 0.1$ | $(3.2 \pm 0.03) \times 10^5$ | $1.0 \pm 0.1$ |
| $gq \to gq$ | $-25 \pm 46$ | $4.2 \pm 0.1$ | $(8.9 \pm 0.9) \times 10^4$ | $-0.2 \pm 0.1$ |
| $qq \to qq$ | $-34 \pm 55$ | $2.7 \pm 0.2$ | $(5.4 \pm 1.1) \times 10^4$ | $-1.1 \pm 0.2$ |
| $q_a q_b \to q_a q_b$ | $-65 \pm 52$ | $3.3 \pm 0.1$ | $(7.3 \pm 1.1) \times 10^4$ | $-0.8 \pm 0.2$ |
| $q\bar{q} \to gg$ | $253 \pm 28$ | $4.7 \pm 0.1$ | $(1.2 \pm 0.1) \times 10^5$ | $0.3 \pm 0.1$ |
| $q\bar{q} \to q\bar{q}$ | $223 \pm 40$ | $4.1 \pm 0.1$ | $(1.7 \pm 0.1) \times 10^5$ | $-0.2 \pm 0.1$ |
| $gg \to q\bar{q}$ | $1362 \pm 15$ | $7.4 \pm 0.1$ | $(4.3 \pm 0.03) \times 10^5$ | $1.1 \pm 0.1$ |
| $q_a\bar{q}_a \to q_b\bar{q}_b$ | $-169 \pm 30$ | $6.8 \pm 0.1$ | $-(2.7 \pm 0.7) \times 10^4$ | $0.6 \pm 0.1$ |
| $\bar{q}\,\bar{q} \to \bar{q}\,\bar{q}$ | $397 \pm 34$ | $8.2 \pm 0.1$ | $(1.4 \pm 0.1) \times 10^5$ | $0.6 \pm 0.1$ |
| $\bar{q}_a\bar{q}_a \to \bar{q}_b\bar{q}_b$ | $468 \pm 28$ | $8.7 \pm 0.1$ | $(1.8 \pm 0.1) \times 10^5$ | $0.9 \pm 0.1$ |
| $g\bar{q} \to g\bar{q}$ | $720 \pm 18$ | $8.0 \pm 0.1$ | $(2.4 \pm 0.1) \times 10^5$ | $1.2 \pm 0.1$ |

Contributions from parton–parton scattering with less peaked shapes at low x will result in a smaller value of $D$. A similar argument applies to the s dependence of $D_Q(s)$. For a rough estimate of the effect of the QCD scale for the inclusive jet simulation, we ran the Toy MC using 7 TeV PDFs to simulate the 13 TeV data—instead of using the appropriate 13 TeV PDFs. We found that $n_{pT}$ changes by only 0.32%, whereas the change to $D$ was 6% and for $n_{xR0}$ of order 5% and the changes to $D_Q$ and $n_{xRQ0}$ were, 22% and 16%, respectively. Hence, the $[x_R - p_T]$ sector is much more sensitive to the QCD scale than the a function.

The $A(p_T)$ functions are not influenced by the η acceptance regions. In our analyses of inclusive reactions, the data are sampled in the $[p_T - x_R]$ plane defined by a quadrilateral with the four constraint equations listed below:

$$x_R \leq 1; p_{T\min} \leq p_T \leq p_{T\max}; x_R \leq \frac{2p_T}{\sqrt{s}} \cosh(\eta_{\max}); x_R \geq \frac{2p_T}{\sqrt{s}}. \tag{32}$$

As an estimate of the |η| boundary constraints, we simulated $g\,g \to g\,g$ scattering in our Toy MC. In this exercise, instead of fixing |η| to various values in order to histogram the $p_T$ distribution as the data are parsed, we fixed $x_R$ to a set of discrete values to determine the $p_T$ distribution. In essence, we are simulating the invariant cross section:

$$\frac{d^2\sigma}{2\pi p_T dp_T dx_R} = G(\sqrt{s}, p_T, x_R), \tag{33}$$

where $G(\sqrt{s}, p_T, x_R)$ is another function of those variables from which we can extract an a function and an *F*-function. Note that $p_T$ and $x_R$ are independent when the simulated kinematic point on the $[p_T - x_R]$ plane is within the quadrilateral region given by the constraints of Equation (32) and only become coupled on the boundaries. The radial scaling variable is symmetric between hemispheres in p–p and AA collisions, whereas the $x_R$ distribution may in fact be different in the pA case. The results the with |η| ≤ 3 simulations for these two cross section definitions are given in Table 5 below.

**Table 5.** Shown are the values of the jet parameters $n_{xR0}$ for $g\,g \to g\,g$ scattering subprocess at $\sqrt{s}$ = 13 TeV inclusive jet simulation by our Toy MC for two cross section definitions. The quoted errors were determined by the consistency of the fits and not by the statistics of the MC simulation. No finite bin corrections were applied. Note that η is double valued, either > 0 or < 0, whereas $0 < x_R \leq 1.0$.

| Cross Section Definition | $\frac{d^2\sigma}{2\pi p_T dp_T d\eta}$ | $\frac{d^2\sigma}{2\pi p_T dp_T dx_R}$ |
|---|---|---|
| $n_{pT}$ | $6.76 \pm 0.03$ | $6.59 \pm 0.02$ |
| $n_{xR0}$ | $6.83 \pm 0.04$ | $7.3 \pm 0.1$ |
| $n_{xRQ0}$ | $1.04 \pm 0.04$ | $0.1 \pm 0.1$ |
| $D$ (GeV/c) | $948 \pm 17$ | $950 \pm 37$ |
| $D_Q$ (GeV/c)$^2$ | $(3.01 \pm 0.03) \times 10^5$ | $(2.98 \pm 0.08) \times 10^5$ |

The power indices are somewhat different, but the distortion parameters $D$ and $D_Q$ are essentially the same. We take these results as being consistent for the different cross section ($x_R$ vs. |η|) schemes of the two calculations within the phase space samplings of the two calculations. One would expect that future data sets will have higher statistics and consequently more refined binning so that the experimental form of the $x_R$ behavior can be better measured.[5]

### 3.3. Deconstruction of PDF Shape

We have seen in Tables 3 and 4 that there is a close connection between the PDFs of the colliding partons and the jet parameters κ, $n_{pT}$, $D$, $D_Q$, $n_{xR0}$ and $n_{xRQ0}$. The a function, in particular, has a direct connection to the underlying parton distributions—especially to their very low-x behavior. In order to gain insight, we revert to our Toy MC by probing the underlying dependences with greatly simplified one-parameter models of the colliding parton PDFs. We consider only $g\,g \to g\,g$ scattering and greatly simplify the gluon PDF in three forms in order to determine which of the shape parameters of the radial scaling jet description strongly depends on the simplified gluon PDF parameters. The three forms are one that emphasizes the low-x behavior, one that emphasizes the high-x behavior and the Pomeron [28] which describes the gluon distribution at very low x in a simplified form. This study is the first step towards probing hard scattering of the colliding partons

as expressed by our six-parameter formulation of the inclusive jet scattering differential cross section.

In this study, we consider the Pomeron form of the gluon PDF which gives us a simplified view of the very low-x behavior:

$$xG(x, Q^2) \sim \exp\sqrt{\frac{48}{11 - \frac{2}{3}n_f} \ln\left(\frac{\ln\left(\frac{Q^2}{\Lambda^2}\right)}{\ln\left(\frac{Q_0^2}{\Lambda^2}\right)}\right) \ln\left(\frac{1}{x}\right)}. \tag{34}$$

We set $\Lambda_{QCD}^2 = (0.34 \text{ GeV/c})^2$, $Q^2 = s = (13 \times 10^3 \text{ GeV/c})^2$, $Q_0^2 = (3.2 \text{ GeV/c})^2$ and $n_f = 6$, forcing $xG(x, Q^2)$ to follow the CT10 [23] gluon PDF $xG(x, s)$ distribution in the interval $10^{-5} < x < 10^{-3}$ within an overall normalization factor. The low-x gluon distribution in the Pomeron approximation can be expressed in the form $\sim 1/x^\mu$ with an effective power $\mu(x, Q)$ given by:

$$\mu(x, Q) = \frac{1}{2}\sqrt{\frac{48}{11 - \frac{2}{3}n_f} \ln\left(\frac{\ln\left(\frac{Q^2}{\Lambda^2}\right)}{\ln\left(\frac{Q_0^2}{\Lambda^2}\right)}\right)} / \sqrt{\ln\left(\frac{1}{x}\right)}, \tag{35}$$

which closely tracks the effective power of the CT10 gluon distribution for low x.

Thus, guided by the behavior of the Pomeron, we consider two extreme forms of the colliding parton PDFs. We take the form emphasizing the low-x behavior governed by the power index $\mu$ to be:

$$xG(x, Q^2) \sim (1/x)^\mu. \tag{36}$$

Additionally, for the simplified gluon high-x behavior, we follow the expectation of the valence quark distribution to explore a form below controlled by the power index $\nu$:

$$xG(x, Q^2) \sim (1 - x)^\nu. \tag{37}$$

Note that by taking the logarithmic derivatives of the respective $xG(x, Q^2)$ forms, the power indices are related by: $\mu/\nu = x/(1 - x)$. Thus, for example, an extreme value of $\nu$ is required to emulate the low-x behavior determined by $\mu$ and vice versa. Therefore, the two behaviors are essentially independent.

The toy simulation program was executed with these choices of the gluon PDF for $\sqrt{s} = 13$ TeV. In the simulations, we allowed $\alpha_s(Q^2)$ to evolve by $Q = p_T$. As usual, the MC 'data' were analyzed in the same manner as data, other toy simulations and Pythia 8.1 simulations with $|\eta| \leq 3$. The $p_T$ range considered was $106 \text{ GeV/c} \leq p_T \leq 1440 \text{ GeV/c}$ corresponding to the ATLAS 13 TeV data, where the upper $p_T$ cutoff ensures at least four rapidity bins of the 'data' being within $x_R \leq 0.9$.

The shape parameters $\mu$ and $\nu$ were varied and the resulting inclusive cross section parameters given by Equation (4) studied. Most striking is that the a function power index, $n_{pT}$, is an almost-linear function of $\mu$ which controls the low-x PDF shape. At the other extreme of high $x$, we find that the $n_{xR0}$ parameter is approximately linear in $\nu$ with an almost one-to-one correspondence $n_{xR0} \sim \nu$. These two dominant behaviors are shown in Figure 7, furnishing a rough interpretation of the observed jet data behavior.

However, we find that all six of our parameters ($\kappa$, $n_{pT}$, $D$, $n_{xR0}$, $D_Q$, and $n_{xRQ0}$) depend on $\mu$ and $\nu$. Further, the distortion parameters $D$ and $D_Q$ are complicated functions of $\mu$ and $\nu$ We find that both $D$ and $D_Q$ have peak values of 400 GeV/c and $1 \times 10^5$ $(\text{GeV/c})^2$, respectively, at approximately $\mu \sim 0.5$. Additionally, both $D$ and $D_Q$ are negative, with minimum values at approximately $\nu \sim 8$ to 10. Their complete behaviors are shown in Appendix B.

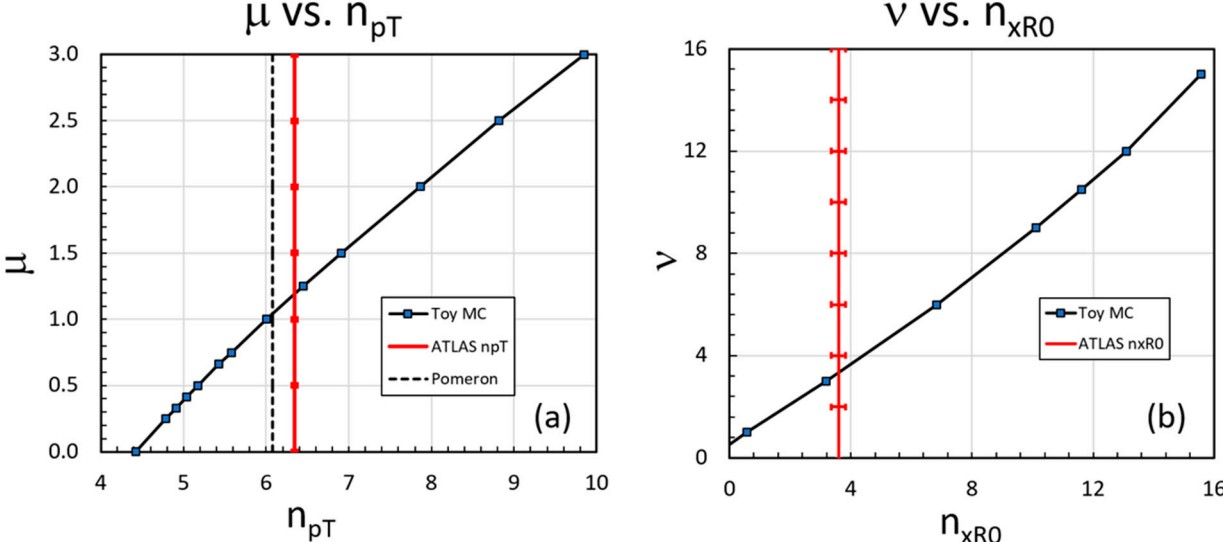

**Figure 7.** Shown are the values of the toy inclusive jet simulations as a function of the power indices $\mu$ and $\nu$ of our two simple models of the gluon PDF. The sensitivity of $n_{pT}$ to $\mu$, the power of $(1/x)^\mu$ is shown in (a). The black dotted line indicates the $n_{pT}$ value resulting from the Pomeron and the red solid line and horizontal error bars indicate the central value of $n_{pT}$ of the ATLAS 13 TeV inclusive jet data and errors, respectively. The sensitivity of $n_{xR0}$ to the power index $\nu$ of $(1-x)^\nu$ is shown in (**b**). The red lines indicate the experimental value from the ATLAS jet data with error bars. Notice that the power index of the a function is mostly controlled by the low x-peaking behavior of the PDF shown in (**a**), whereas the $n_{xR0}$ behavior is controlled by the parameter $\nu$ that shapes the high-x behavior shown in (**b**).

This study confirms the strong sensitivity of $n_{pT}$ on the low-x behavior of the PDFs of the colliding partons as shown in Figure 7a, implying that the 'operative' $\mu \sim 1.2$. Expressing the CT10 [23] gluon distribution as $xG(x, Q^2) \sim 1/x^{\mu(x,Q)}$, we find $\mu \sim 1.2$ for $x \sim 4 \times 10^{-2}$. The Pomeron has a $\mu$ value that is always smaller than the CT10 distribution for larger x. Hence, the power index of the a function in the Pomeron case is smaller than that of the gluon distribution.

Because of the double-log approximation, there is a very slow evolution of $\mu(x, Q)$ with increasing $Q \sim \sqrt{s}$. Hence, the value of $\mu(x, Q)$ for the Pomeron approximation of the low-x gluon distribution at $\sqrt{s} = 2.76$ TeV is not much different from that of $\sqrt{s} = 13$ TeV consistent with the observation that $n_{pT}$ is nearly independent of $\sqrt{s}$. In fact, both the CT10 parameterization of the gluon and quark PDFs at low x roughly follow a linear $1/(\ln(1/x))^{1/2}$ dependence and have $\mu$ values at the same x that increase by only $\sim 6\%$ between $Q \sim \sqrt{s} = 2.76$ TeV and 13 TeV.

### 3.4. Consequent F-Function

The corresponding F-functions of the $\mu$ - $\nu$ study by the algorithm of Equation (12) are shown in Figure 8. Both PDF extremes result in slowly increasing F-functions for $\mu = \nu = 0$ and decreasing F-functions for finite values of $\mu$ and $\nu$, with the $\nu$ case imposing the largest influence as expected by Figure 7b. Hence, the high- $x_R$ behavior is determined chiefly by the large $x$ shape of the colliding parton PDFs.

### 3.5. Summary

We have performed this study considering only $g\,g \to g\,g$, but, according to Table 2, many other types of parton–parton scattering have roughly the same $p_T$ behavior so the conclusions here are more general than pertaining to just $g\,g \to g\,g$ scattering. The gluon distribution dominates for $x < 0.1$ and the quark distribution dominates for $x > 0.1$. In broad terms, it is the behavior of the colliding parton PDFs at low x that controls the power $n_{pT}$ and the $D$ and $D_Q$ parameters. Since the gluon distribution is most peaked at low x, gluon scatterings are primarily responsible for determining the values of the $n_{pT}$ and $D$

and $D_Q$ parameters in inclusive jet production. The high-x region is the domain of the quark parton PDFs—especially the valence quarks at very high x. This study indicates that the high-x behavior of the parton PDF controls $n_{xR0}$ and has little influence on $n_{pT}$.

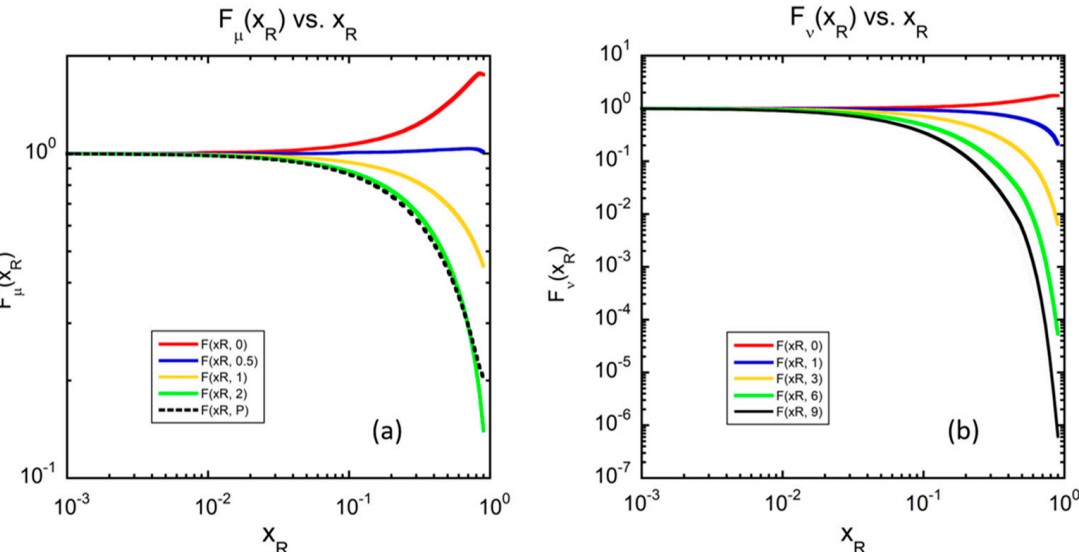

**Figure 8.** The resultant *F*-functions in the two cases are shown. For (**a**), the red through green lines represent the $F(x_R)$ functions for $\mu = 0, 0.5, 1, 2$, respectively, and the Pomeron is shown by the dotted black line. In (**b**), we show the *F*-functions for $\nu = 0$ (red line), 1, 3, 6 and 9 (black line). ATLAS data lie between the yellow and green lines in (**b**). The *F*-functions were calculated from their respective $n_{xR0}$ and $n_{xRQ0}$ values for each $\mu$ and $\nu$.

Finally, we compute the effective $\mu$ and $\nu$ values as a function of x for the CT10 gluon and quark PDFs at $\sqrt{s} = 13$ TeV. The results are shown in Figure 9.

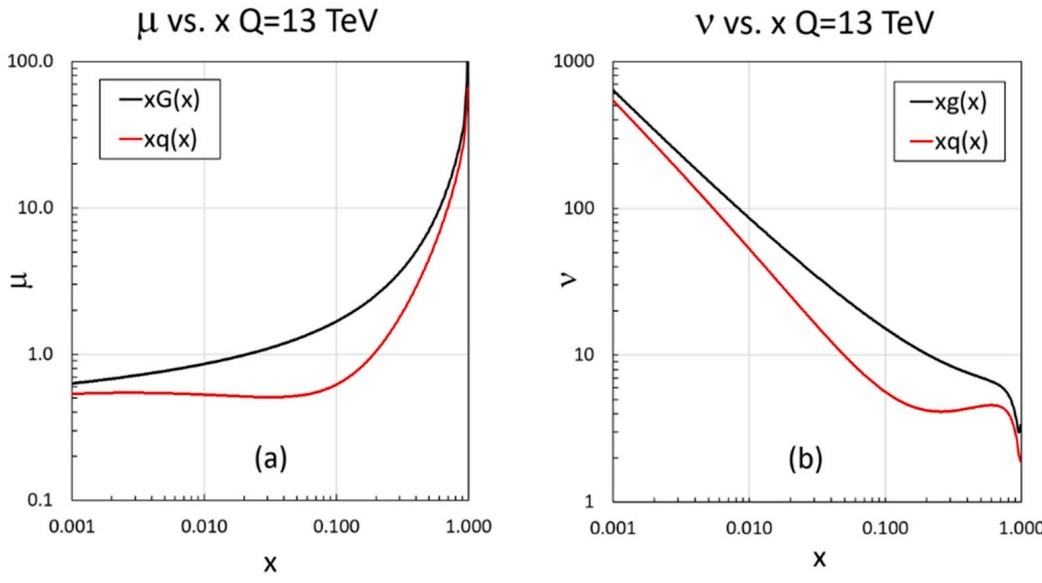

**Figure 9.** The effective $\mu$ (**a**) and $\nu$ values (**b**) for the CT10 gluon (black lines) and quark (red lines) PDFs at $\sqrt{s} = 13$ TeV. Notice that $\mu$ for quarks is smaller than that for gluons implying that $n_{pT}$ for quark scattering is smaller than the corresponding values for gluons—consistent with the discussion above. Further, we note that since the quark distribution dominates at high x, the value of $n_{xR0}$ is largely determined by the quark PDFs at x > 0.1 where there is a 'valence shelf' $\nu \sim 4$ consistent with the $n_{xR0}$ value measured in jets (ATLAS $n_{xR0} = 3.6 \pm 0.2$) by the $n_{xR0} - \nu$ relation of Figure 7b. Therefore, our analysis using $p_T$ and $x_R$ sheds light on the shape of the PDFs in different regions by the different sensitivities of the parameters $n_{pT}$, $D$, $D_Q$, $n_{xR0}$ and $n_{xRQ0}$ and are nicely correlated with the shapes of the colliding PDFs.

### 4. The s—Dependence of Inclusive Jets

Using the data of Tables II, VII and VIII and Figure 11 of our earlier publication [9], we concluded that the power of $p_T$, characterized by the parameter $n_{pT}$, for the invariant inclusive cross section for jets is roughly independent of $\sqrt{s}$ and that the magnitude of the jet invariant cross section governed by the parameter κ(s) grows approximately linearly with s. We also noted in [9] the value of $n_{pT} = 6.5 \pm 0.3$ for inclusive jets—a value that is ~ 8 standard deviations above the expected dimensional limit of 4 that is mandated by the dimensional definition of the inclusive invariant cross section (Equation (1)) and that of the underlying parton–parton hard-scattering cross section $d\hat{\sigma}/d\hat{t}$.

Here, we have refined our analysis using HEPData (https://www.hepdata.net/, accessed on 28 April 2021) not available at the time of our early publication using the ATLAS inclusive jet data for R = 0.4 from $\sqrt{s}$ = 2.76, 5.02, 7, 8 and 13 TeV [17,29–32], respectively. We have analyzed each data set in the same manner as demonstrated above. These findings are tabulated in Appendix A.

We treat the data at each $\sqrt{s}$ as being analyzed by the same algorithms, jet energy scale calibration, pileup corrections, etc., although the data span the 2013 to 2018 time period corresponding to the early days of commissioning the LHC and the ATLAS detector through to their more mature operating periods. We have analyzed the data conservatively by taking statistical and systematic errors in quadrature—even so, these errors may not represent all the errors between different $\sqrt{s}$ data sets.

The s dependence of the jet parameters is shown in Figure 10. It is interesting to note that the parameters κ, D and $D_Q$ increase as $\sqrt{s}$ increases. While the scatter of the data is large, κ, D and $D_Q$ appear to follow power laws in $\sqrt{s}$, such as of the form $\kappa(s) \sim \kappa_0 \left(\sqrt{s}\right)^{ns}$, where $\kappa_0$ and $n_s$ are constants. In order to estimate the constant term $\kappa_0$ and the power law index $n_s$ for κ(s), as well the corresponding parameters for the parameters D(s) and $D_Q$(s), we fit to the following log equations:

$$
\begin{aligned}
\ln(\kappa(s)) &\sim n_s \ln\left(\sqrt{s}\right) + \ln(\kappa_0), \\
\ln(D(s)) &\sim n_D \ln\left(\sqrt{s}\right) + \ln(D_0), \\
\ln\left(D_Q(s)\right) &\sim n_{DQ} \ln\left(\sqrt{s}\right) + \ln\left(D_{Q0}\right),
\end{aligned}
\tag{38}
$$

where $n_s$, $n_D$ and $n_{DQ}$ are the power indices and $\ln(\kappa_0)$, $\ln(D_0)$ and $\ln(D_{Q0})$ are the constant terms. The resulting fits of data and two Monte Carlo simulations (to be described later) are shown in Table 6 below. What is of most interest are the power indices, $n_s$, $n_D$ and $n_{DQ}$. It appears that $\kappa(s)$ and $D_Q(s)$ increase with $\sqrt{s}$ with a power ~2, whereas $D(s)$ increases with a power ~1, that is linearly in $\sqrt{s}$. Later, we will show that the power index, $n_s$, that governs how the a function magnitude parameter, κ(s), increases with increasing s, is key to maintaining the overall correct dimension of the invariant cross section.

The resulting simulation of inclusive dijets by our Toy MC is displayed in Figure 10a for κ vs. $\sqrt{s}$ and Figure 10b for $n_{pT}$ vs. $\sqrt{s}$. The data points suffer from considerable scatter, but, from the figure, we conclude that data, Pythia 8.1 and Toy MC roughly agree that the magnitude of the cross section governed by the parameter $\kappa(\sqrt{s})$ grows nearly linearly with increasing s and that the $p_T$ power of the a function is consistent with $n_{pT} = 6.3 \pm 0.1$ of the average value for ATLAS jets and is essentially independent of $\sqrt{s}$. The power indices, $n_{pT}$, $n_{xR0}$ and $n_{xRQ0}$ in Equations (6) and (10) show no systematic variation in $\sqrt{s}$ although their errors are large and their correlations may be important in determining the shape of the *F*-function.

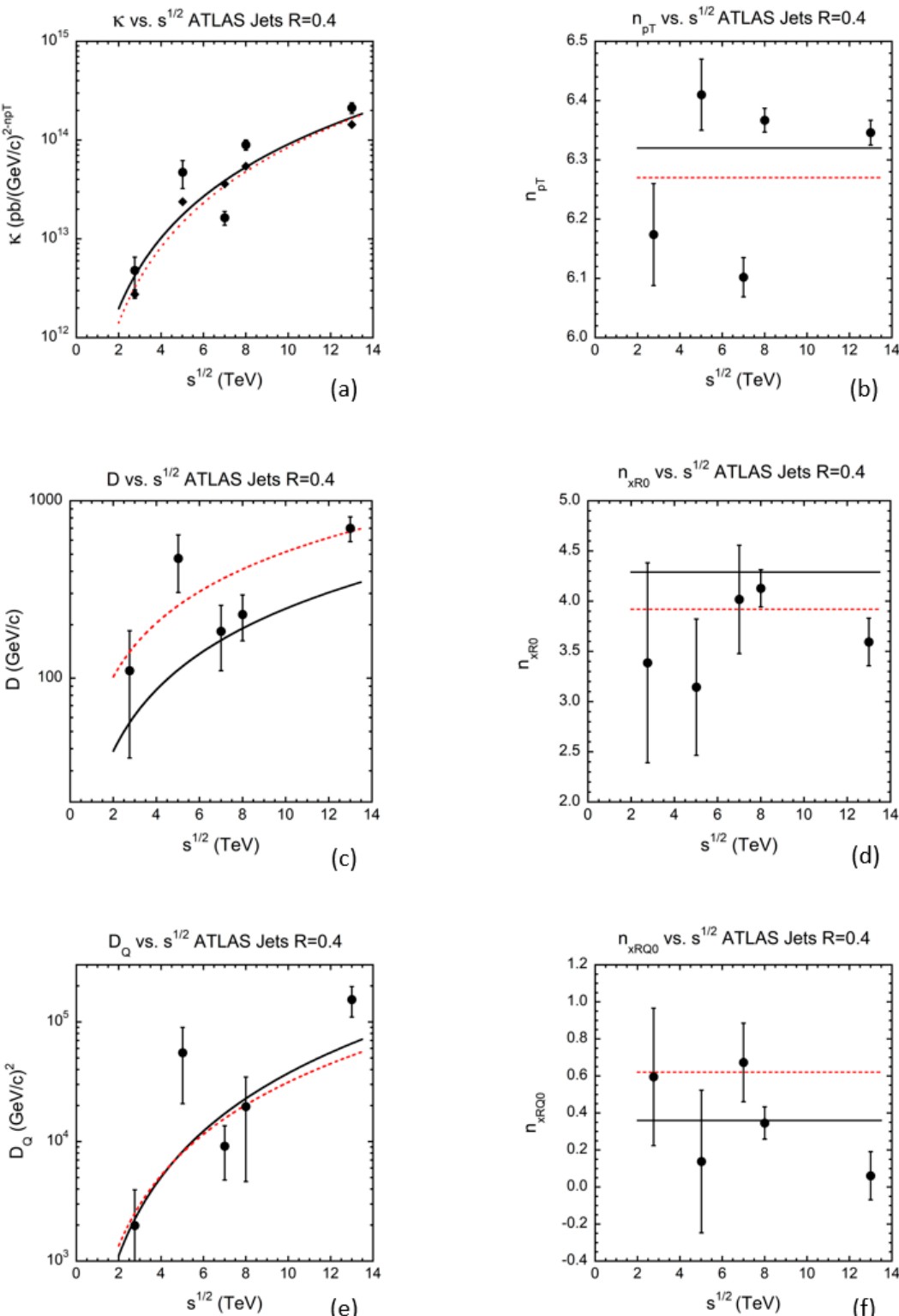

**Figure 10.** The fit values for each of the inclusive jet measurements performed by the ATLAS collaboration are plotted as a function of $\sqrt{s}$. The red dotted lines are power law fits ((**a**) $\kappa(\sqrt{s})$, (**c**) $D(s)$, (**e**) $D_Q(s)$, respectively) to the toy model MC described in the text and the solid black lines are the corresponding power law fits of the Pythia 8.1 simulations. The power indices are plotted in (**b**) $n_{pT}$, (**d**) $n_{xR0}$, (**f**) $n_{xRQ0}$, respectively. The simulations show little s dependence, so only the average power indices are shown. Since the power index, $n_{pT}$, shows no systematic $\sqrt{s}$ dependence, we show the $\kappa$ values computed by fixing $n_{pT}$ to its average. Those points are shown as diamonds in (**a**).

**Table 6.** The $\sqrt{s}$, expressed in GeV, dependence of the parameters $\kappa(\sqrt{s})$, $D(s)$ and $D_Q(s)$ are tabulated. The data were fit assuming all five $\sqrt{s}$ points are of equal statistical weight. For the $\kappa(s)$ parameter, the Monte Carlo simulations (Pythia 8.1, Toy) were normalized to the $\sqrt{s}$ = 13 TeV data point. In summary, it appears that $\kappa(s)$ and $D_Q(s)$ increase with a power ~2, whereas $D(s)$ increases with a power ~1, that is linearly in $\sqrt{s}$. It is apparent from Figure 10 that there is considerable scatter in the parameter values of the jet fits so for the logarithmic behavior of the parameters with respect to $\ln(\sqrt{s})$ we quote the regression value, $R^2$, instead of the $\chi^2/\text{d.f.}$

| Parameter | Power Index | Constant Term | $R^2$ |
|---|---|---|---|
| $\kappa(s)$ | $n_s$ | $\ln(\kappa_0)$ | |
| Data | $2.3 \pm 0.7$ | $11 \pm 6$ | 0.79 |
| Pythia 8.1 | $2.4 \pm 0.2$ | $10.2 \pm 1.7$ | 0.98 |
| Toy | $2.5 \pm 0.3$ | $9 \pm 3$ | 0.96 |
| $D(s)$ | $n_D$ | $\ln(D_0)$ | |
| Data | $0.9 \pm 0.5$ | $-2 \pm 5$ | 0.51 |
| Pythia 8.1 $\sqrt{s} \geq 7$ TeV | $1.1 \pm 0.1$ | $-5.1 \pm 0.8$ | 0.99 |
| Toy | $1.0 \pm 0.1$ | $-3.0 \pm 0.1$ | 0.96 |
| $D_Q(s)$ | $n_{DQ}$ | $\ln(D_{Q0})$ | |
| Data | $2.3 \pm 1.0$ | $-10 \pm 9$ | 0.63 |
| Pythia 8.1 $\sqrt{s} \geq 7$ TeV | $2.2 \pm 0.1$ | $-9.6 \pm 0.9$ | 0.99 |
| Toy | $2.0 \pm 0.5$ | $-8 \pm 5$ | 0.84 |

The general behavior of the $x_R$ sector of the inclusive cross sections is characterized by the shape of $(1 - x_R)^{n_{xR}}$ which is mostly controlled by the power index $n_{xR}$ at low $x_R$. (We consider the quadratic term $\exp\left[n_{xRQ} \ln^2(1 - x_R)\right]$ controlled by $n_{xRQ}(p_T) = D_Q/p_T + n_{xRQ0}$ to be a perturbation.) Considering two nearby points in $y$, $y_1$ and $y_2 > y_1$, and noting that $\ln(1 - x_R) \approx -x_R$ for small $x_R$, we estimate that the power $n_{xR}$ of $(1 - x_R)$ should be approximately:

$$n_{xR} \sim \left(\frac{\sqrt{s}}{2p_T}\right) \ln\left(\frac{\sigma(p_T, y_1)}{\sigma(p_T, y_2)}\right)\left(\frac{1}{\cosh(y_2) - \cosh(y_1)}\right), \tag{39}$$

where $\sigma(p_T, y)$ denotes the inclusive invariant differential cross section given by Equation (2). Hence, we expect that the $n_{xR}$ power index should be proportional to $\sqrt{s}/2p_T$ at low $x_R$—especially when dominated by $g\,g \to g\,g$ scattering. This behavior is captured in the $D$ term defined by Equation (10). From Equation (39), we find that:

$$D(s) \sim -\frac{\sqrt{s}}{2}\left(\frac{d\ln(\sigma(p_{T\min}, y))}{d\cosh(y)}\right), \tag{40}$$

where the derivative is evaluated at the lowest measured $p_T$ for a given $\sqrt{s}$ data set. Note that the minus sign enforces the sign convention of Equation (39). By this formulation, the value of $D$ should grow with increasing $\sqrt{s}$ if the derivative $d\ln(\sigma(p_{T\min}))/d\cosh(y)$ has little s-dependence—roughly true when the kinematic point is near the rapidity plateau. The data, Pythia 8.1 and the Toy MC all follow this behavior (see Figure 10c).

Note that the s dependence of the a function, $A(\sqrt{s}, p_T)$, is the same as the inclusive differential cross section at $x_R = 0$. In our formulation, the dimension of the invariant cross section is determined by the term $A(\sqrt{s}, p_T) = \kappa(s)/p_T^{n_{pT}}$ given by Equation (6). Since $\kappa(s)$ for inclusive jets is proportional to s[GeV$^2$] as shown in Figure 10a and $n_{pT} \sim 6$ [(GeV/c)$^{-6}$], the overall dimensions of the inclusive cross section are [(GeV/c)$^2$] [(GeV/c)$^{-6}$] ~$1/p_T^4$[(GeV/c)$^{-4}$] ~ [cm$^2$/(GeV/c)$^2$], thus the same dimensions of the hard-scattering cross section d$\sigma$/dt ((GeV/c)$^{-4}$), as it must be by dimensional analysis of

Equation (1). Later, we will refine the relationship between $\kappa(s)$ and $n_{pT}$ which we call the "dimensional custodian".

One might ask why the exponent of the $p_T$ power is approximately independent on the value of $\sqrt{s}$. One factor is that the leading term in the hard $g\,g \to g\,g$ ($2 \to 2$) scattering cross section $d\hat{\sigma}/d\hat{t}$ at small $p_T$ is independent of $\sqrt{s}$. Another factor is that the evolution of the PDFs enhances the low-x region as $\sqrt{s}$ increases, which is partially compensated by the decrease in the $\alpha_s(Q)^2$ term of the hard-scattering cross sections as the scale Q increases. In fact, we find that the fractions of subprocesses given in Table 3 for 13 TeV jets are nearly the same for $\sqrt{s} = 2.76$ TeV with the ATLAS experimental cuts. For example, the $(g\,g \to g\,g)/(g\,q \to g\,q)$ channels at 2.76 TeV are 68.8%/13.6%, respectively, vs. 66.2%/13.1% for 13 TeV. The overall conclusion is that our formulation of the inclusive invariant cross section given by Equations (2)–(4) suggests that the $A(\sqrt{s}, p_T)$ function is a less sensitive way to study QCD and, as will be discussed later, the $x_R$ dependence of the cross section, primarily through the distortion parameters $D$ and $D_Q$, is a much more sensitive measure of theory, hard parton scattering and the nucleon PDFs. We find that the power indices $n_{pT}$, $n_{xR0}$ and $n_{xRQ0}$ have little s-dependence, making their average values meaningful. Most of the s dependence is in the magnitude factor $\kappa(s)$ of the a function. The averages are tabulated in Table 7 below for ATLAS jets.

**Table 7.** The power indices $n_{pT}$, $n_{xR0}$ and $n_{xRQ0}$ of data averaged over $2.76 \leq \sqrt{s} \leq 13$ TeV of inclusive jets (R = 0.4) measured by ATLAS are compared with two MC simulations and are tabulated. The quoted errors are the standard deviation about the average. The Monte Carlo simulations, Pythia 8.1 and Toy MC are in good agreement with $n_{pT}$ data and are consistent with the $< n_{xR0} >$ and $< n_{xRQ0} >$ data values, for the later within 50% errors.

| Parameter | Average Value | Data/MC |
|---|---|---|
| | $< n_{pT} >$ | |
| Data | $6.3 \pm 0.1$ | |
| Pythia 8.1 | $6.28 \pm 0.04$ | $1.00 \pm 0.02$ |
| Toy MC | $6.27 \pm 0.06$ | $1.00 \pm 0.02$ |
| | $< n_{xR0} >$ | |
| Data | $3.7 \pm 0.4$ | |
| Pythia 8.1 | $4.4 \pm 0.2$ | $0.8 \pm 0.1$ |
| Toy MC | $3.92 \pm 0.04$ | $0.9 \pm 0.1$ |
| | $< n_{xRQ0} >$ | |
| Data | $0.4 \pm 0.2$ | |
| Pythia 8.1 | $0.4 \pm 0.1$ | $0.8 \pm 0.6$ |
| Toy MC | $0.62 \pm 0.01$ | $0.6 \pm 0.4$ |

As another way of envisioning the s dependence of inclusive jets arising from $\kappa(s)$, we normalize the $A(p_T)$ functions by multiplying them by $p_T{}^{6.35}$, the reciprocal of the $p_T$ dependence of the 13 TeV ATLAS R = 0.4 jet data set, as is frequently done for cosmic ray spectra—in Figure 11a. Plotted in the figure are the two Monte Carlo simulations, normalized to the 13 TeV ATLAS data. The strong s dependence is evident. It is of note that the Toy MC follows the much more sophisticated Pythia 8.1 simulation up to $\sqrt{s} = 100$ TeV, indicating that, at least for the kinematic region of the simulation, the hard scattering of partons dominates.

It is of course true that the s dependence of the $A(p_T)$ function is not the complete story of the cross section s dependence. We have therefore computed the integral inclusive cross section of the ATLAS R = 0.4 jet data in the kinematic region measured and normalized

for $|y| \leq 3$ by using our parameterizations given in Tables A2–A4 in Appendix A. The integration is defined as:

$$\sigma\left(\sqrt{s}\right) = \pi \int\limits_{0}^{3} dy \int\limits_{p_T^L}^{p_T^H} \left( \frac{d^2\sigma\left(\sqrt{s}, p_T, y\right)}{2\pi p_T d p_T dy} \right) dp_T^2, \tag{41}$$

where the same $p_T$ interval $100 \leq p_T \leq 3000$ GeV/c is used for all values of $\sqrt{s}$. In the integration, $x_R < 0.9$ where $F(x_R) > 10^{-3}$. The cross section integral is compared to the integral of the $A(p_T)$ function for the same $p_T$ range. In order to study the behavior over the range of measured $\sqrt{s}$ values, we choose the same lower and upper $p_T$ limits independent of $\sqrt{s}$. The result is shown in Figure 12, where we conclude that most the s dependence of the integrated cross section is in the $A(p_T)$ function, and that the overall integrated cross section rises faster than the integral of $A(p_T)$.

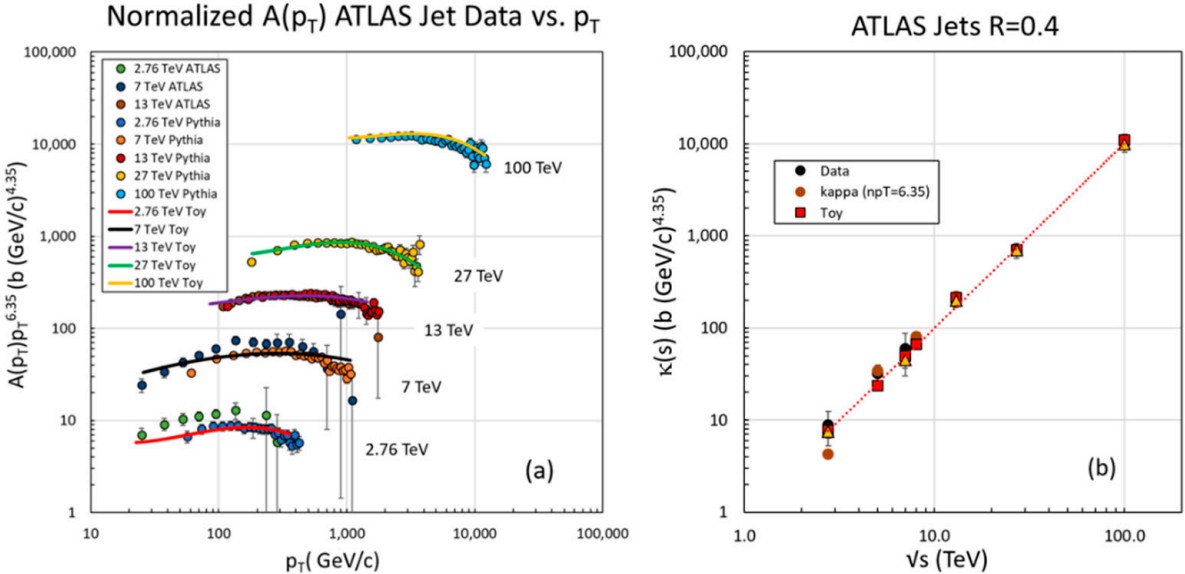

**Figure 11.** (**a**) The $A(p_T)$ functions multiplied by $p_T^{6.35}$ for ATLAS inclusive jet data (R = 0.4) are plotted vs. $p_T$. The Toy MC simulations for each energy are displayed by the red dotted lines and the Pythia 8.1 simulations are indicated by the circles with black outlines. The Toy MC has been smoothed by a cubic polynomial fit in $p_T$. It follows Pythia 8.1 for the three overlapping $\sqrt{s}$ values, but both tend to underestimate the data at 2.76 (green circles) and 7 TeV (blue circles). All three functions overlap at $\sqrt{s}$ = 13 TeV, where the simulations were normalized. The vertical axis is a measure of the magnitude parameter $\kappa(s)$ in Equation (6) given in units of barns (GeV/c)$^{4.35}$. On the right, (**b**), $\kappa(\sqrt{s})$, computed by the average of each normalized $A(p_T)$ is plotted as a function of $\sqrt{s}$ for data and the two MC simulations. The errors of the averaged are the standard deviation about the averages. Also shown is the value of $\kappa(\sqrt{s})$ of data computed by fixing $n_{pT}$ = 6.35. The red dotted line is a fit to the Pythia 8.1 simulation consistent with $\kappa(\sqrt{s}) \sim (\sqrt{s})^2$. Note that each normalized a function is not a constant indicating a violation of the pure power law.

The resulting $F(x_R)$ functions are plotted in Figure 13. The Toy MC gives the better fit for $\sqrt{s}$ = 2.76 and 5.02 TeV, but has approximately the same quality as the Pythia 8.1 simulation for 7 TeV. On the other hand, Pythia 8.1 gives the better fits for 8 and 13 TeV. The resulting $\chi^2$ are shown in Table 8.

Since it is difficult to see any s dependence of $F(\sqrt{s}, x_R)$ in Figure 13, we plot the fitted functions to Equation (16) of the analysis of the 2.76, 7 and 13 TeV ATLAS data in Figure 14. It is apparent that as the COM energy increases, the $F$-function becomes steeper but all data follow a simple power law $\sim (1 - x_R)^{n0}$ at low $x_R$.

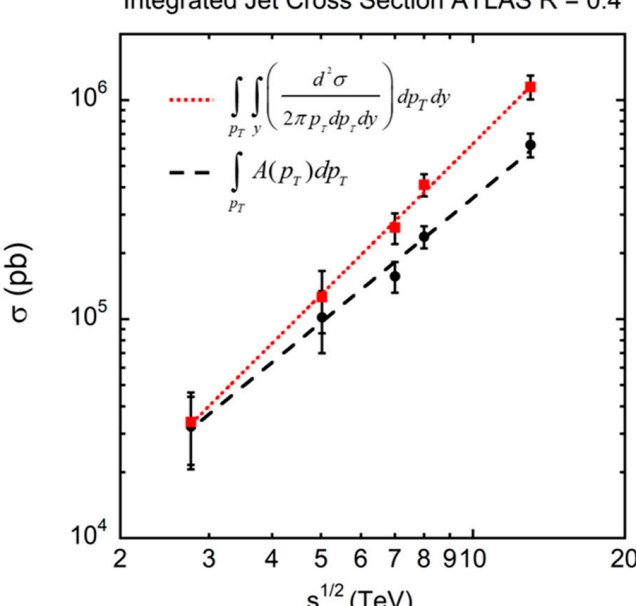

**Figure 12.** The integrated jet cross section for $100 \leq p_T \leq 3000$ GeV/c, $x_R \leq 0.9$, and $|y| \leq 3.0$ for the parameterization of the inclusive cross section as measured by ATLAS (red squares) and of the integral of the corresponding a functions (black circles). The error bars were estimated from the relative errors of $\kappa(s)$, the a function magnitude parameter. The red dotted line and dashed black line represent a power law growth $(\sqrt{s})^{2.29 \pm 0.05}$ and $(\sqrt{s})^{1.89 \pm 0.09}$ with $\chi^2$/d.f. of 1.3/5 ($p = 0.93$) and 0.62/5 ($p = 0.99$), respectively. The integrated cross section grows faster in $\sqrt{s}$ than the integrated a function because of the broadening of the rapidity distribution with increasing $\sqrt{s}$.

**Table 8.** The $\chi^2$ values of comparison of data with themselves vs. data with the Toy MC and Pythia 8.1 simulations of data shown in Figure 13. In all comparisons, the $n_{xR0}$ and $n_{xRQ0}$ values were used for the model. The column "Data" is the comparison of actual data points with the model determined by Equation (13). The $\chi^2$ values and the numbers of degrees of freedom are shown as ratio and value for each comparison: data vs. data, data vs. Toy MC and data vs. Pythia 8.1. The errors used in the $\chi^2$ computation include the errors of the fit as well as data systematic and systematic errors added in quadrature. $p$-Values less than $10^{-8}$ were set to 0.

| $\sqrt{s}$ (TeV) | Data | | Toy MC | | Pythia 8.1 MC | |
|---|---|---|---|---|---|---|
| | $\chi^2$/d.f. | $p$-Value | $\chi^2$/d.f. | $p$-Value | $\chi^2$/d.f. | $p$-Value |
| 2.76 TeV ATLAS | 21.2/52 | 1.00 | 71/52 | 0.04 | 130/52 | $1.3 \times 10^{-8}$ |
| 5.02 TeV ATLAS | 14.1/72 | 1.00 | 149/72 | $2.7 \times 10^{-7}$ | - | - |
| 7 TeV* ATLAS | 15.0/83 | 1.00 | 51.3/83 | 0.99 | 50.3/83 | 0.99 |
| 8 TeV ATLAS | 161/155 | 0.35 | 3464/155 | 0 | - | - |
| 13 TeV ATLAS | 137/171 | 0.97 | 1161/171 | 0 | 771/171 | 0 |
| 13 TeV CMS | 143/154 | 0.73 | 248/154 | $2.4 \times 10^{-6}$ | 494/154 | 0 |

* Three outlier points have been eliminated in calculation of $\chi^2$. We see that the data are consistent with themselves, consistent with the Toy MC up to 7 TeV, whereas the adequacy of Pythia to provide a good fit shows no systematic s dependence, but the $\chi^2$ are generally poor, except for 7 TeV. Overall, only in the case of data vs. data are the $\chi^2$ values reasonable, which indicates that the shape of $F(\sqrt{s}, x_R)$ is compatible with Equation (16) and that the MC simulations are being strongly tested.

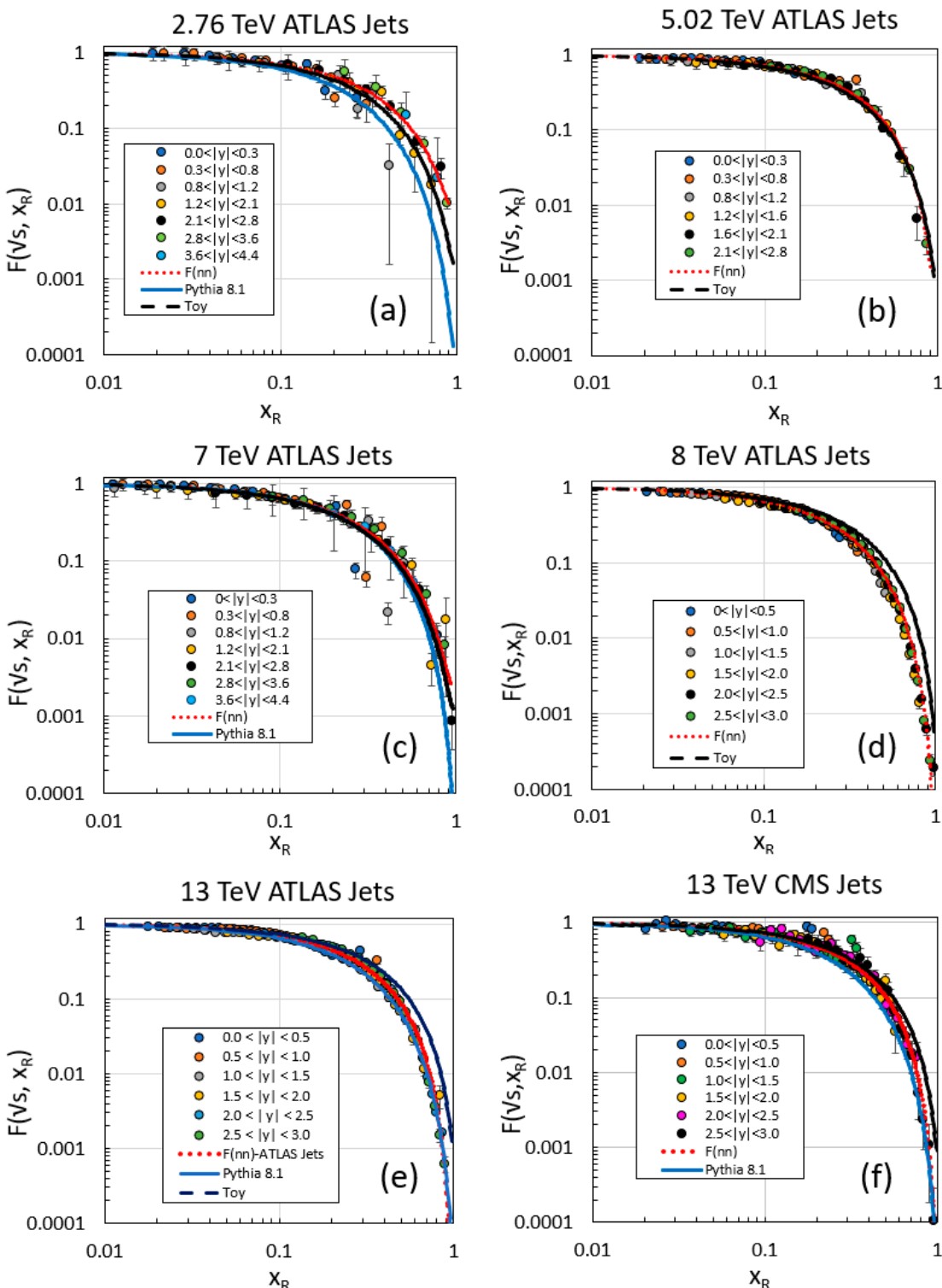

**Figure 13.** The $x_R$ distributions for five values of $\sqrt{s}$ for ATLAS inclusive jets ((**a**–**e**) for $\sqrt{s}$ = 2.76, 5.02, 7, 8, 13 TeV, respectively) and for CMS jets at 13 TeV—(**f**). The red dotted line represents a fit to the data (Equation (16)), the solid blue line the results of a Pythia 8.1 simulation and the dashed black line the results of the toy model simulation. Pythia agrees roughly with data for 7 and 13 TeV but underestimates the data at 2.76 TeV. The Toy model generally overestimates data at high $x_R$ for 8 and 13 TeV.

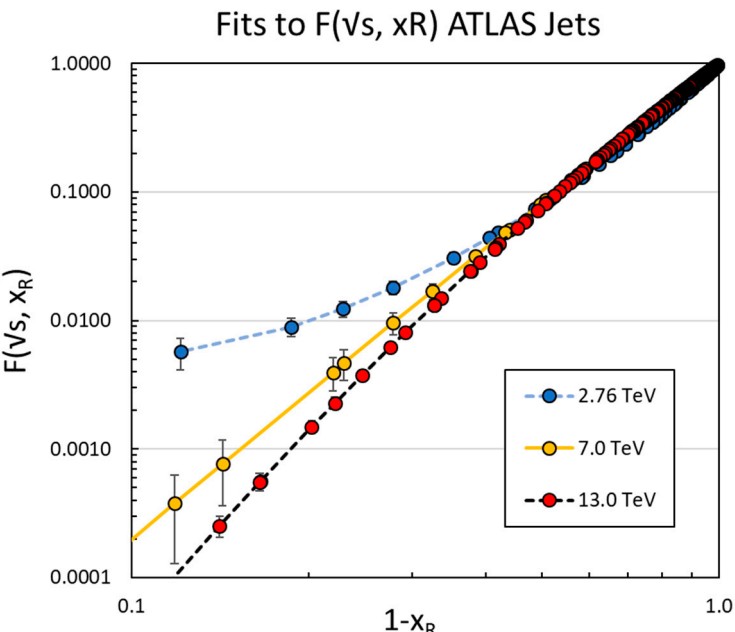

**Figure 14.** Shown are the fits to Equation (16) of the *F*-function of ATLAS inclusive jet data (R = 0.4) taken at 2.76, 7 and 13 TeV indicated by the blue dotted, solid gold and dashed black lines, respectively. The 'data' points plotted are the values of the fit with the error bars indicating the error of the fit plotted at the same $(1 - x_R)$ values of the data. The plot shows that the data follow a power law $\sim (1 - x_R)^{nxR0}$ for low $x_R$, but deviate significantly at higher $x_R$—especially for at 2.76 TeV. The deviation is controlled by the $n_{xRQ0}$ term. Some s dependence is expected since the gluon, quark and antiquark contents of the proton evolve with $\sqrt{s}$.

## 5. Analysis of Inclusive Isolated Photons

The production of photons by either parton–parton annihilation or by parton–parton Bremsstrahlung in p–p collisions has been of long-term interest [33,34]. It provides a useful window into the gluon and quark distributions of the proton without the complications of hadronization of particles in the final state [35]. However, there is a third, and complicating process, where the detected photon arises from a higher-order fragmentation process into a photon from the quark legs of the collision. Thus, the analysis of the data on this process is subtle and important corrections have to be made in order to isolate the direct photon signal from these background fragmentation processes as well as from that from $\pi^0 \to \gamma\gamma$ decay. This isolation cut is typically performed by demanding that the transverse energy in a hollow cone centered on the detected photon be less than some empirical functional value.

As we did for jet production in Table 2, we list the dominant processes that contribute to direct photon production in Table 9 [24,25]. Note that both quark Bremsstrahlung and quark–antiquark annihilation cross sections have a leading $1/p_T^2$ behavior at low $p_T$ for fixed $\sqrt{\hat{s}}$. In the case of Bremsstrahlung $gq \to \gamma q$ we note (dropping the caret designation of the parton–parton COM variables):

$$\frac{d\sigma}{dt} = \frac{\pi\alpha_e\alpha_s}{s^2}\left(-\frac{e_q^2}{3}\right)\left(\frac{u}{s}+\frac{s}{u}\right) \approx \frac{\pi\alpha_e\alpha_s}{s}\left(\frac{e_q^2}{3}\right)\frac{1}{p_T^2}, \tag{42}$$

where $\alpha_e$ is the fine structure constant, $\alpha_s$ is the strong interaction coupling strength and $e_q$ is the electric charge of the radiating quark. At the maximum $p_T = \sqrt{s}/2$ limit for a given $s$, the differential cross section is finite and has the value:

$$\frac{d\sigma}{dt} = \frac{5}{2}\frac{\pi\alpha_e\alpha_s}{s^2}\left(\frac{e_q^2}{3}\right) = \frac{\pi\alpha_e\alpha_s}{p_T^4}\left(\frac{5e_q^2}{96}\right). \tag{43}$$

This cross section and those for other parton–parton scatterings are tabulated in Table 9.

**Table 9.** The various parton–parton scattering processes (carets not shown) that contribute to direct photon production in p–p scattering are listed. The leading $p_T$ dependences at small $p_T$ and the values of the cross sections at the kinematic limit where $p_T = \sqrt{\hat{s}}/2$ are shown. The $gg \to \gamma\gamma$ cross section depends on both $\alpha_e$ and $\alpha_s$ and on s, t, and u through the terms $T_i$. This channel was neglected since it contributes $\leq 0.1\%$ of the overall production cross section. In all cases, the exact expressions for the scattering cross sections were used in the Toy MC simulations. The expressions shown here are only to give a rough idea of the $\hat{s}$ and $p_T$ dependences. Notice that, unlike the cross sections of Table 2, the photon-producing cross sections are $\hat{s}$ dependent and at low $p_T$ and fall with increasing $p_T$ as $1/p_T^2$.

| Process | Leading $p_T$ | Value at $p_T=\sqrt{\hat{s}}/2$ |
|---|---|---|
| $gq \to \gamma q$ | $\frac{d\sigma}{dt} \approx \frac{\pi\alpha_e\alpha_s}{s}\left(\frac{e_q^2}{3}\right)\frac{1}{p_T^2}$ | $\frac{d\sigma}{dt} = \frac{\pi\alpha_e\alpha_s}{p_T^4}\left(\frac{5e_q^2}{96}\right)$ |
| $q\bar{q} \to \gamma g$ | $\frac{d\sigma}{dt} \approx \frac{\pi\alpha_e\alpha_s}{s}\left(\frac{8e_q^2}{9}\right)\frac{1}{p_T^2}$ | $\frac{d\sigma}{dt} = \frac{\pi\alpha_e\alpha_s}{p_T^4}\left(\frac{e_q^2}{9}\right)$ |
| $q\bar{q} \to \gamma\gamma$ | $\frac{d\sigma}{dt} \approx \frac{\pi\alpha_e^2}{s}\left(\frac{2e_q^4}{3}\right)\frac{1}{p_T^2}$ | $\frac{d\sigma}{dt} = \frac{\pi\alpha_e^2}{p_T^4}\left(\frac{e_q^4}{12}\right)$ |
| $gg \to \gamma\gamma$ | $\frac{d\sigma}{dt} \sim \frac{\alpha_s^2}{8\pi^2}\frac{\pi\alpha_e^2}{s^2}\left(\sum_{i=1}^{nf}eq_i^2\right)^2\sum_i T_i$ $\sim (1 \times 10^{-3})\frac{d\sigma(q\bar{q}\to\gamma g)}{dt}$ | See Owens [25] (neglected) |

The leading $p_T$ power for constant s is $(1/s)(1/p_T^2)$ rather than the steeper $1/p_T^4$ that governs the underlying hard-scattering dominant terms in jet production. Therefore, we would expect to see a reflection of this $1/p_T^2$ behavior predicting that inclusive photons will have a flatter $A(p_T)$ spectrum. We also expect that the $x_R$ sector, characterized by the power indices $n_{xR}(p_T)$ and $n_{xRQ}(p_T)$, will be different from inclusive jet production because photon creation tends to be in the direction of the incoming electric fields causing a peaking along the incoming beam direction. However, the peaking behavior will be modulated by the hadronic part of the photon creation process. Hence, the $x_R$ distribution for inclusive photons is the result of a competition of the peaking by QED and the flattening of QCD.

We have analyzed ATLAS 8 [36] and 13 TeV [37] photon data in the same manner as we did for inclusive jets, namely using Equation (4) as the ansatz. (There are ATLAS 7 TeV data [38] that cover $0.0 \leq |\eta| \leq 1.81$ in only three bins making them insufficient coverage for our full analysis.) The results are shown in Figure 15. The power law of the isolated photon a function is quite evident. Hence, the inclusive isolated photon cross section can be factorized into a $[p_T - \sqrt{s}]$ sector and a $[p_T - x_R]$ sector as we found for inclusive jets but we find that the $[p_T - x_R]$ sector is significantly different.

The power indices of the $A(p_T)$ photon fits are: $n_{pT} = 5.81 \pm 0.02$ for 8 TeV, $5.91 \pm 0.04$ for 13 TeV data and $5.85 \pm 0.12$ for 13 TeV theory [39]—all three values being significantly smaller $(9.8\sigma)$ than the corresponding values for inclusive jets $(6.35 \pm 0.02)$ discussed earlier. The corresponding $\kappa(s)$ values for the ATLAS photon measurements are: $(5.0 \pm 0.5) \times 10^9$ pb/GeV/c$^2$, $(1.8 \pm 0.4) \times 10^{10}$ pb/GeV/c$^2$ and $(1.3 \pm 0.9) \times 10^{10}$ pb/GeV/c$^2$ for 8 TeV data, 13 TeV data and 13 TeV simulation, respectively. As in the case of inclusive jets, we find that the $\kappa(s)$ value for inclusive isolated photons increases with increasing $\sqrt{s}$.

Turning to the $[p_T - x_R]$ sector, we plot in Figure 16 the power indices $n_{xR}(p_T)$ and $n_{xRQ}(p_T)$ as a function of $p_T$ for data and the simulation based on an NLO pQCD predictions from Jetphox based on the MMHT2014 PDFs taken for the posted HepData of the paper [37].

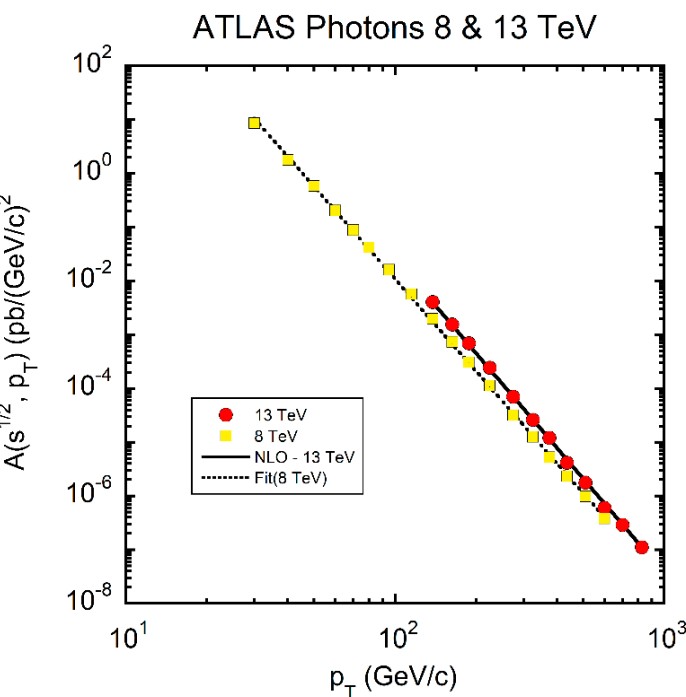

**Figure 15.** The $A(p_T)$ functions for direct photons measured in the ATLAS detector for $\sqrt{s}$ = 8 TeV (yellow squares) and $\sqrt{s}$ = 13 TeV (red circles). The dotted line is a power law fit to the 8 TeV data and the solid black line is a fit to the NLO simulation for the 13 TeV data with parameters $\kappa = (1.28 \pm 0.95) \times 10^{10}$ pb/(GeV/c)$^{2-npT}$ and $n_{pT}$ = 5.85 $\pm$ 0.12.

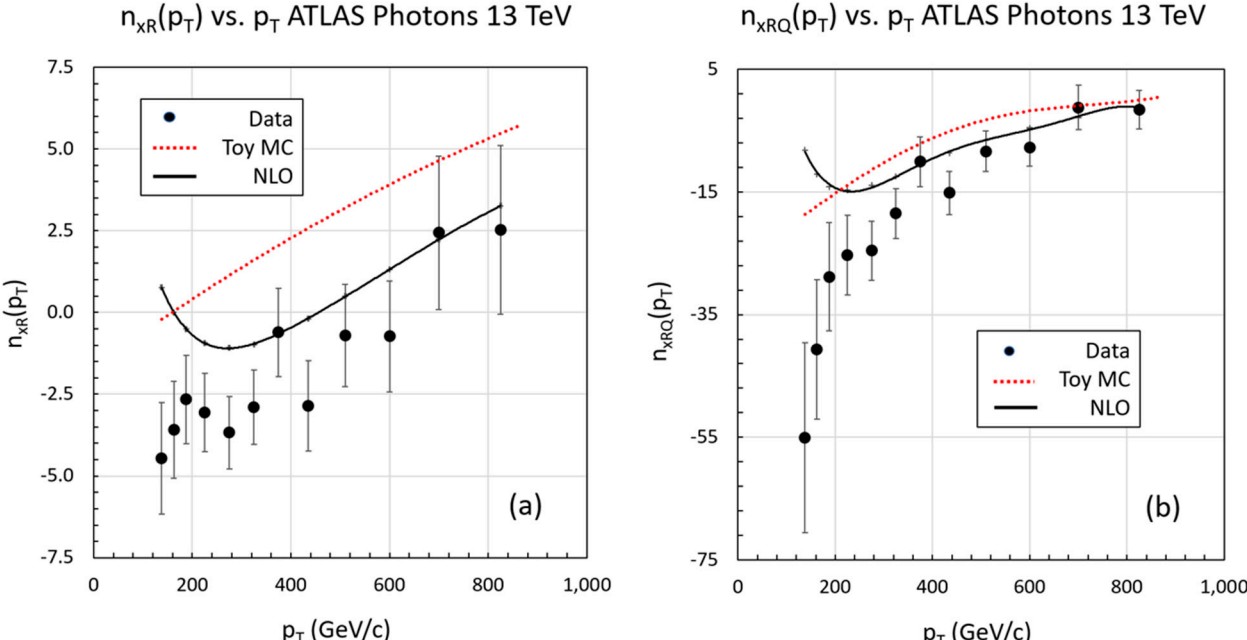

**Figure 16.** The power indices $n_{xR}$ and $n_{xRQ}$ ((**a**,**b**) respectively) for inclusive isolated photons measured in the ATLAS detector for $\sqrt{s}$ = 13 TeV compared to simulations. The red dotted lines represent the results of our Toy model simulation and the solid black lines the theory simulation shown in the ATLAS paper [37]. Both the Toy MC and NLO simulation tend to overestimate the power indices. The NLO simulation is a better representation of $n_{xR}(p_T)$ but is approximately the same quality as the Toy MC for $n_{xRQ}(p_T)$. The large systematic errors of the NLO simulation are not shown.

We have simulated direct photons in the same manner as we did for inclusive jets by considering only the underlying hard scattering of gluons and quarks. We neglect the so-call fragmentation production of photons and higher level QCD contributions [39]. The dominant underlying hard-scattering cross sections are proportional to $\alpha_e \alpha_s$, hence the first order in electromagnetic and hadronic interactions. In the simulation we take $\alpha_e(M_Z) = 1/128$ [26] to be a constant and $\alpha_s(Q^2)$ to evolve as described above. The contributing underlying parton–parton scattering cross sections are tabulated below. There are two major types—those involving Bremsstrahlung and those involving quark–antiquark annihilation into a photon–gluon pair. A third contribution involves quark–antiquark annihilation into a photon pair. Unlike purely hadronic processes, which are roughly independent of $\hat{s}$, the photon-producing cross sections fall with increasing $\hat{s}$ and have a $1/p_T^2$ dependence at low $p_T$. Hence, it is the very low $\hat{s} = s x_1 x_2$ region that dominates the inclusive photon cross sections.

The Toy MC uses the CT10 PDFs but does not account for photon identification efficiency, radiative corrections or isolation effects—hence is only a rough guide to the data. The results of the simulations in comparison to the 13 TeV ATLAS data, where the statistical and photon ID errors were added in quadrature, are shown in Table 10 for our Toy MC. Our simulation involves only the various hard-scattering processes listed in the table and the corresponding parton distributions from a parameterization of CT10 [23]. We have not simulated fragmentation photons or the effect of photon isolation cuts.

**Table 10.** The contributions of each process, $\sigma(i)$, operative in the production of direct photons at $\sqrt{s}$ = 8 and 13 TeV integrated over $34.8 \leq p_T \leq 990\,\mathrm{GeV/c}$ and $138.3 \leq p_T \leq 1932\,\mathrm{GeV/c}$ for our Toy MC 8 and 13 TeV simulations, respectively, and over $0 \leq |y| \leq 2.5$ for both energies are shown along with their corresponding power law indices. The power law indices were calculated for MC data in the range $34.8 \leq p_T \leq 600\,\mathrm{GeV/c}$ and $138.3 \leq p_T \leq 1310\,\mathrm{GeV/c}$ for 8 and 13 TeV data, respectively, which roughly corresponds to the ATLAS data ranges. The $gg \to \gamma\gamma$ process is neglected. The Toy MC is only useful as a rough guide—it underestimates the value of $n_{pT}$ by approximately 11%. There is a small $\sqrt{s}$ dependence in the fractional contributions to the total cross section but the values of $n_{pT}$ for all processes change by only 0.56%.

| | **8 TeV** | | **13 TeV** | |
|---|---|---|---|---|
| **Process** | $\sigma\,(i)/\sigma\,(\mathrm{All})$ | $n_{pT}$ | $\sigma\,(i)/\sigma\,(\mathrm{All})$ | $n_{pT}$ |
| All | 100% | $5.35 \pm 0.01$ | 100% | $5.32 \pm 0.01$ |
| $g\,u \to \gamma\,u$ $g\,d \to \gamma\,d$ $g\,s \to \gamma\,s$ $g\,\bar{u} \to \gamma\,\bar{u}$ $g\,\bar{d} \to \gamma\,\bar{d}$ $g\,\bar{s} \to \gamma\,\bar{s}$ | 89.01% | $5.40 \pm 0.01$ | 85.52% | $5.35 \pm 0.01$ |
| $u\,\bar{u} \to \gamma\,g$ $d\,\bar{d} \to \gamma\,g$ $s\,\bar{s} \to \gamma\,g$ | 10.80% | $5.11 \pm 0.01$ | 14.19% | $5.19 \pm 0.01$ |
| $u\,\bar{u} \to \gamma\,\gamma$ $d\,\bar{d} \to \gamma\,\gamma$ $s\,\bar{s} \to \gamma\,\gamma$ | 0.19% | $4.95 \pm 0.01$ | 0.29% | $5.01 \pm 0.01$ |

Note that processes involving Bremsstrahlung at $\sqrt{s}$ = 13 TeV comprise approximately 86% of the cross section for $E_T \geq 100$ GeV at $\sqrt{s}$ = 13 TeV, whereas the sum of the annihilation cross sections is 14%. Ichou and d'Enterria [35] estimate the same fractions at $\sqrt{s}$ = 14 TeV to be 84% and 16%, respectively, with an isolation cut R = $(\Delta\eta^2 + \Delta\phi^2)^{1/2} = 0.4$.

It is interesting to compare the ratio of isolated prompt photons at 13 TeV to those measure at 8 TeV. The ATLAS collaboration has performed such a calculation and has compared the results to a NLO QCD calculation using the program [40]. One would expect

that some of the simplicity of the Toy MC simulation, such as the absence of common systematic errors would cancel in taking the ratio. In the ATLAS paper, the ratio of the 13 TeV/8 TeV data is plotted as a function of $p_T$ in four separate $|y|$ bins. Displaying the ratio in this manner implies that the comparison is made between an $x_R$ value at 13 TeV and a larger $x_R$ value at 8 TeV given by $x_R(8) = 13/8\, x_R(13)$.

Referring to Equation (4), immediately we notice that since $n_{pT} \sim$ constant, most of the variation of the ratio is in the $[p_T - x_R]$ sector. Since the cross section falls with increasing $x_R$, comparing the 13 and 8 TeV data with this $x_R$ relation between the two $\sqrt{s}$ values ensures that the $R^\gamma_{13/8}(p_T, \eta)$ ratio increases with increasing $p_T$ and increasing $|\eta|$, namely for increasing $x_R$. Most of the $p_T$ dependence in the ratio is therefore due to the decrease in the cross section as the kinematic point approaches the kinematic boundary, $x_R = 1$ with the decrease larger for the 8 TeV data than the 13 TeV data. Thus, this test of theory has a strong kinematic component that is relatively easy to simulate.

In terms of our formulation of inclusive cross sections and their comparison at the same $p_T$ and $|\eta]$, we can express the ratio $R^\gamma_{13/8}$ as the product of three ratios:

$$R^\gamma_{13/8}(p_T, \eta) = R_A(p_T) R(p_T, x_R) R_Q(p_T, x_R), \tag{44}$$

where

$$
\begin{aligned}
R_A(p_T) &= \left( \frac{\kappa(13)}{\kappa(8)} \right) \left( p_T^{\Delta n pT} \right), \\
R(p_T, x_R) &= \left( \frac{(1-x_R)^{n x_R(13, p_T)}}{(1-13/8 x_R)^{n x_R(8, p_T)}} \right), \\
R_Q(p_T, x_R) &= \exp\left( n_{xRQ}(13, p_T) \ln^2(1 - x_R) - n_{xRQ}(8, p_T) \ln^2(1 - 13/8 x_R) \right),
\end{aligned} \tag{45}
$$

and $\Delta n_{pT} = n_{pT}(8) - n_{pT}(13) \approx -0.1 \pm 0.04$. This near-equality of the $n_{pT}$ exponents makes $R_A(p_T)$ slowly varying—in fact $< R_A > = 1.9 \pm 0.1$ for $100 < p_T < 1310$ GeV/c.

We show the $R^\gamma_{13/8}$ ratio for different $\eta$ slices as a function of $p_T$ in Figure 17. The Toy MC represents the data rather well over the entire kinematic range. It underestimates the ratio for the two lower $\eta$ bins, but is remarkably close to the data and the more sophisticated NLO QCD simulation for the two higher ones. The NLO simulation is a better representation of the data than the Toy simulation with $\chi^2/\text{d.f.} = 31/47$ ($p = 0.97$), while the Toy simulation has $\chi^2/\text{d.f.} = 175/47$ ($p = 1.2 \times 10^{-16}$), where most of the contribution to the $\chi^2$ comes from the lower two $|\eta|$ bins.

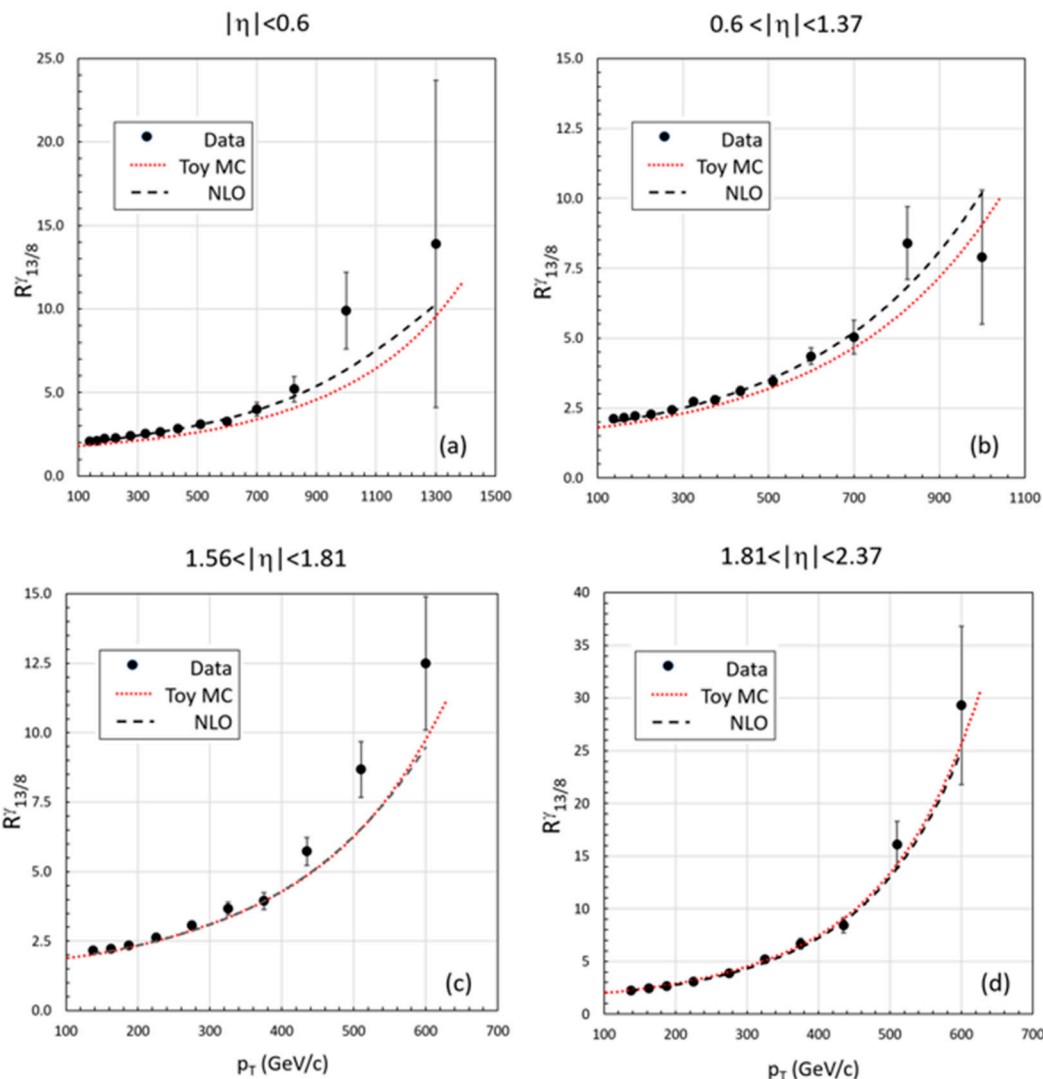

**Figure 17.** The ratio of the inclusive isolated photon cross sections measured by the ATLAS collaboration at $\sqrt{s}$ = 8 and 13 TeV is plotted as a function of $p_T$ ($E_T$) for various slices of $|\eta|$ in (**a**–**d**). The dashed black curves represent the NLO QCD calculation and the red dotted curves are the results of our Toy MC simulation. The NLO calculation is a better representation of the data but it is noteworthy that the Toy simulation is so close to the data—especially at the two larger $|\eta|$ bins. This is an indication that most of the $p_T$ dependence is due to the decrease in the cross section as $x_R \to 1$, the high $x_R$ kinematic boundary.

## 6. Analysis of Heavy Mesons and Baryons

The study of heavy quark final states (charm and bottom) offers tests of both perturbative QCD as well as non-perturbative corrections. The literature is extensive and there are highly developed MC simulations which replicate the data quite well. Because the mass of the bottom quark, m, defines the scale of the strong coupling in such processes and is much larger than $\Lambda_{QCD}$, perturbative calculations can be conducted. The same is roughly true for charm quark states despite being lighter and closer to the $\Lambda_{QCD}$. Higher-order QCD diagrams ($\sim\alpha_s(m^2)^3$) are important since the cross section for $gg \to gg$ scattering is several orders of magnitude larger than $gg \to Q\overline{Q}$, thereby permitting heavy quark pair production to occur by fragmentation of one of the gluon lines into $Q\overline{Q}$. These processes are of order $\alpha_s{}^3(Q^2)$ [41,42]. Hence, we would expect our very elementary lowest-order simulation to be only a rough guide.

In order to gain a theoretical foundation of heavy quark (meson/baryon) production, we first examine the underlying parton–parton scattering processes that contribute. We

consider both open charm and bottom states, as well as "onium" states (J/ψ, ψ(2S), Υ(1S)). There are two main processes—gluon–gluon scattering into a heavy quark–antiquark pair and light quark–antiquark annihilation into a heavy quark–antiquark pair. The appropriate cross sections are shown in Table 11 in the small $p_T$ approximation as well as at the maximum $p_T$ kinematic limit.

**Table 11.** The leading order cross sections and their behavior at low $p_T$ and at the kinematic boundary are tabulated. Again, the carets denoting variables of parton–parton scattering have been omitted. The equations were derived from [43] and checked with those in the PDG [26]. The modified transverse momentum $P_T = \sqrt{p_T^2 + m^2}$.

| Process | Leading $p_T$ | Value at $p_T = \frac{1}{2}\sqrt{\hat{s}-4m^2}$ $\hat{s} = 4P_T^2$ |
|---|---|---|
| $gg \to Q\overline{Q}$ | $\frac{d\sigma}{dt} = \frac{\pi\alpha_s^2}{s^2}\left(\frac{s}{6(m^2+p_T^2)} - \frac{3}{8}\right)\left[1 - 2\frac{m^4+p_T^4}{s(m^2+p_T^2)}\right]$ $\frac{d\sigma}{dt} \approx \frac{\pi\alpha_s^2}{s}\left(\frac{1}{6(m^2+p_T^2)}\right)$ | $\frac{d\sigma}{dt} = \frac{7}{768}\frac{\pi\alpha_s^2}{P_T^4}\left(1 + \frac{2m^2}{P_T^2} - \frac{2m^4}{P_T^4}\right)$ |
| $q\overline{q} \to Q\overline{Q}$ | $\frac{d\sigma}{dt} = \frac{4\pi\alpha_s^2}{9}\frac{1}{s^2}\left(1 - \frac{2p_T^2}{s}\right)$ | $\frac{d\sigma}{dt} = \frac{\pi\alpha_s^2}{72P_T^4}\left(1 + \frac{m^2}{P_T^2}\right)$ |

Notice that the $gg \to Q\overline{Q}$ cross section has a $1/P_T^2$ behavior at small $p_T$, indicating that the data should follow a power law in the modified transverse momentum $P_T$ with $\Lambda_m$ = m rather than in $p_T$. We expect the a function to be a power law in $1/P_T$ and that the power $n_{p_T}$ should be less than that of inclusive jets since the dominant hard-scatteringhard-scattering cross section goes as ~$1/P_T^2$ rather than ~$1/p_T^4$, as in the case of jets. Since $\hat{s} \geq 4m^2$, the cross sections are finite throughout their kinematic ranges and, as before, we examine the behavior of the cross sections through their approximations. The first approximation of the operative hard-scattering cross sections is to determine the leading $p_T$ term for the case when $\hat{s}$ is well above threshold. Note that $\hat{s} = 2P_T^2(1 + \cosh(y_1 - y_2))$ for $Q\overline{Q}$ production, where the heavy quarks are produced at $y_i$, $i$ =1, 2, respectively, expresses the fact that the rapidity of the b-mesons tend to be correlated [44].

The $gg \to Q\overline{Q}$ process dominates the $q\overline{q} \to Q\overline{Q}$ reaction at low $p_T$ and high $\sqrt{s}$ since the gluon PDFs dominate the quark and antiquark PDFs, while their respective cross sections at low $p_T$ are nearly equal. We see that hard-scattering cross sections for $gg \to Q\overline{Q}$ and $q\overline{q} \to Q\overline{Q}$ are power laws in the modified transverse momentum (transverse mass) $P_T = \sqrt{p_T^2 + m^2}$ as indicated by the data when m ~ $\Lambda_m$. Furthermore, the cross sections are larger when $|y_1 - y_2|$ is small. Both differential cross sections are finite in the limit of the maximum value of $p_T = \sqrt{\hat{s} - 4m^2}/2$ and decrease with increasing s. For the two cross sections we expect the empirical term $\Lambda_m$ to be determined by the mass, m, of the detected particle, but for small m we expect that the $\Lambda_m$ parameter generally will be larger than m because of gluon radiation and parton intrinsic transverse momentum.

We have applied our formulation of inclusive cross sections to the production of heavy mesons and baryons in p–p collisions. Just as in the case of inclusive jets and direct photons, we determine the A and *F*-functions for inclusive heavy quark final states, thereby providing new tools to study them. We note that for those processes at low transverse momentum, $p_T$, of order of the mass of the heavy particle produced, we must engage the parameter $\Lambda_m$ in Equation (5) in order to determine the transverse momentum part of the invariant cross section $A(p_T, \sqrt{s}, \Lambda_m)$ in terms of the modified transverse momentum. In the case of direct production of charm/bottom mesons and baryons, the $\Lambda_m$ value is determined mostly by the mass of the detected heavy particle itself. Additionally, in the case of indirect production of heavy particles where the detected particle is the result of a decay, the $\Lambda_m$ value is determined by the parent particle mass and value of Q ~ (m(parent)−− m(daughter)) of the decay and is generally larger than the direct production case. Unfortunately, we find the data are not extensive enough to include the $n_{xRQ}$ term in Equation (3) so the analysis of heavy mesons to follow is performed with $n_{xRQ} \equiv 0$. In the following, we discuss both the $p_T$ and $x_R$ behavior of heavy particle production.

### 6.1. $P_T$ Dependence of Heavy Particle Production

An example of this application is shown in Figure 18, where we plot the a function of the invariant $B^{\pm}$ inclusive cross section measured by the LHCb collaboration at p–p collisions at $\sqrt{s} = 7$ TeV [45] as a function of both $p_T$ and of the mass-modified transverse momentum $P_T \equiv \sqrt{p_T^2 + \Lambda_m^2}$. We determine the value of $\Lambda_m$ by a minimum $\chi^2$ power law fit to the hypothesis that the modified transverse momentum distribution follows a power law distribution. The fitting process determines $\kappa$, $\Lambda_m$ and $n_{pT}$. For $B^{\pm}$ data shown, we find the $P_T$ power index $n_{pT} = 5.5 \pm 0.2$ for $\Lambda_m = 6.3 \pm 0.3$ GeV/c and $\kappa = (8.8 \pm 4.7) \times 10^3$ µb $(\text{GeV/c})^{n_{pT}-2}$. It is important to note that our formulation not only determines the operative mass term, $\Lambda_m$, in the production of the heavy meson, but also estimates the underlying $P_T$ power law, thereby enabling comparisons with other processes—especially at higher momentum, where $P_T \gg \Lambda_m$.

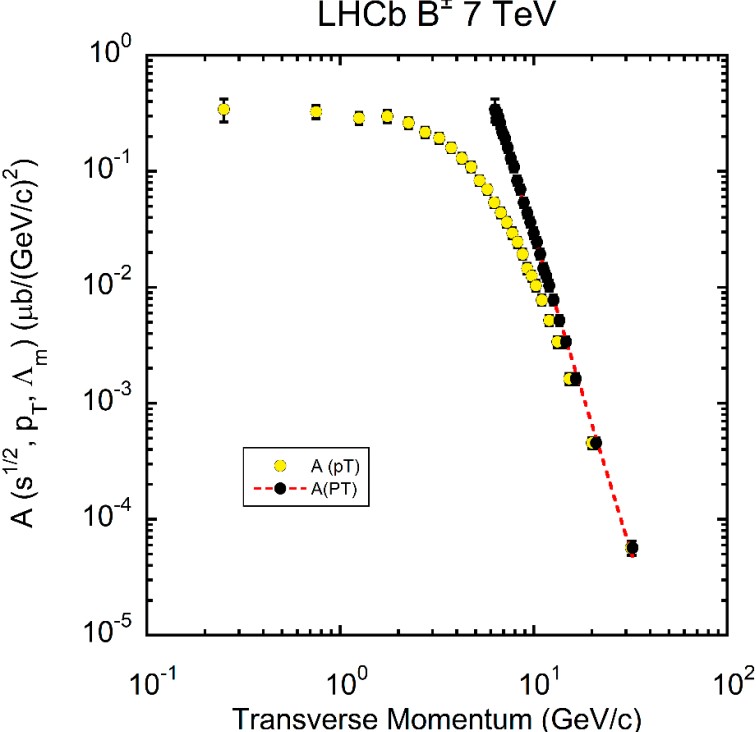

**Figure 18.** Shown is the transverse momentum $p_T$ and modified transverse momentum $P_T$ dependences of the a function for $B^{\pm}$ production at 7 TeV (LHCb [45]). It is evident that the transverse momentum distribution $p_T$ indicated by the yellow circles can be corrected by the empirical mass term $\Lambda_m = 6.3 \pm 0.3$ GeV/c to reveal the underlying power law indicated by the black circles as given in Equation (3). The power index of the $P_T$ distribution is $n_{pT} = 5.5 \pm 0.2$, smaller than that of the inclusive (u-d-s-g) jet production cross sections shown in Figure 3. The 3-parameter fit determines $\kappa = (8.8 \pm 4.7) \times 10^3$ µb $(\text{GeV/c})^{n_{pT}-2}$ with $\chi^2/\text{d.f.} = 6.6/24$ ($p = 1.0$).

Open quark flavor mesons ($\pi^{\pm,0}$, $K^{\pm}$, $D^0$, Ds, D*, $B^{\pm,0}$, $Bs^0$ mesons), vector mesons (such as $\phi$, $J/\psi$ and $\psi(2S)$) and baryons (antiprotons and $\Lambda_b$ baryons) can be analyzed using the $\Lambda_m$ parameter to reveal the underlying $A(P_T)$ power law. Although well known, this correlation of $\Lambda_m$ with the mass of the particle produced is not frequently referenced because inclusive cross sections are presented as $d^2\sigma/dp_T dy$, which distorts the $\Lambda_m$ dependence by simple kinematics, rather than the differential cross section in the invariant phase space form, $d^2\sigma/2\pi p_T dp_T dy$, where a power law in $P_T$ is manifest. Determining the $\Lambda_m$ term in the modified transverse momentum, $P_T$, from data is an important test of the production cross section and potentially yields information of the mother–daughter relationship for particles produced indirectly.

The a function, by definition, should be independent of $y$. Thus, it enables the $p_T$ distributions of data in different y ranges to be compared. In Figure 19, we show the 8 TeV LHCb J/ψ data taken at higher |y| ($2 \leq$ |y| $\leq 4.5$) [46] compared with ATLAS data [47] at lower |y| $\leq 2$. A simultaneous minimum $\chi^2$ fit to both data sets yields $\kappa = (2.32 \pm 0.13) \times 10^6$, $n_{pT} = 6.62 \pm 0.02$ and $\Lambda_m = 3.93 \pm 0.02$ GeV/c with $\chi^2$/d.f. = 73.9/35. Both data sets are at $\sqrt{s} = 8$ TeV, but their respective y regions do not overlap.

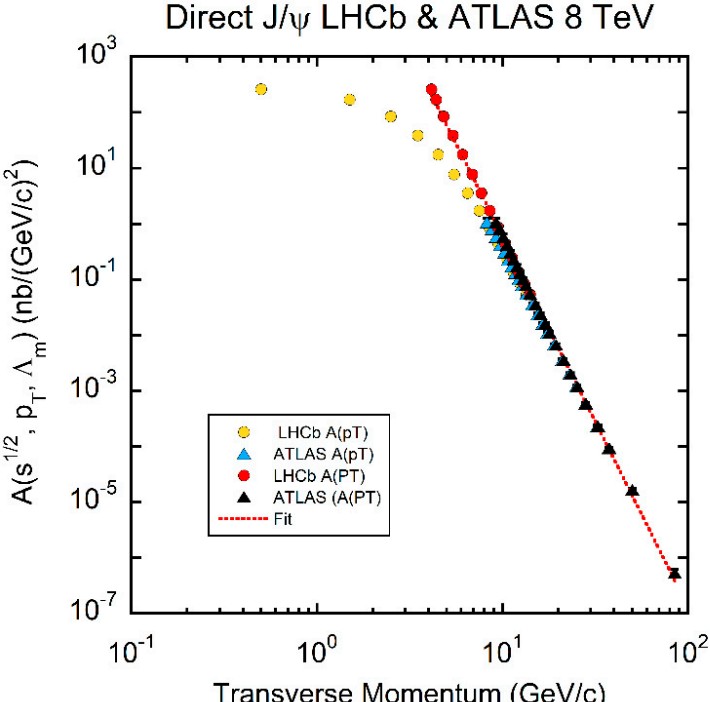

**Figure 19.** The $A(\sqrt{s}, p_T, \Lambda_m)$ values for J/ψ prompt mesons as a function of transverse momentum are plotted for both LHCb (circles) and ATLAS data (triangles) sets at $\sqrt{s} = 8$ TeV. Two values of the transverse momentum are plotted for each point—the $p_T$ value and the modified transverse momentum value derived from $p_T$ given by $P_T \equiv \sqrt{p_T^2 + \Lambda_m^2}$. The yellow circles and blue triangles (mostly covered by black triangles) are $A(\sqrt{s}, p_T, \Lambda_m)$ versus $p_T$ for the LHCb and ATLAS data, respectively, and the red circles (LHCb) and black triangles (ATLAS) are the same data plotted versus the modified transverse momentum, $P_T$. Note that all data have the same power index $n_{pT} = 6.53 \pm 0.03$ indicated as a simultaneous fit to both data sets. The common fit value results in $\Lambda_m = 3.93 \pm 0.02$ GeV/c. The ATLAS data were taken in the interval $0 \leq$ |y| $\leq 2.0$ and the LHCb data in the non-overlapping region $2.0 \leq$ |y| $\leq 4.5$, thereby demonstrating the independence on |y| of $A(\sqrt{s}, p_T, \Lambda_m)$ as a function of $P_T$.

The values of $\Lambda_m$ for other single particle inclusive cross sections are approximately linearly dependent on the rest mass (PDG value [26]) of the produced particle as indicated in Figure 20 (left). The data were taken from Table VIII of reference [9], along with other data [48–62]. However, the linear $\Lambda_m$ –m relation appears to be broken for the Υ(nS). The ATLAS value and CMS values are consistent with the linear relation of the lower mass data, whereas the LHCb values lie below the extrapolated line. The ATLAS data cover |y| $\leq 2.0$ and the CMS data are even more central with |y| $\leq 1.2$, whereas the LHCb data range over $2.0 \leq$ |y| $\leq 4.5$. More data are needed to resolve this discrepancy—especially from the ATLAS and CMS collaborations covering the central |y| range. We have averaged the ATLAS and CMS data with those of the LHCb collaboration in Figure 20 (left). The red dotted line is a minimum $\chi^2$ fit $\Lambda_m = (1.17 \pm 0.04)$ m + $(0.40 \pm 0.04)$ to all data points except the inclusive photon points and the Υ(nS) values. The $\chi^2$/d.f. = 29.7/11 ($p = 1.8 \times 10^{-3}$).

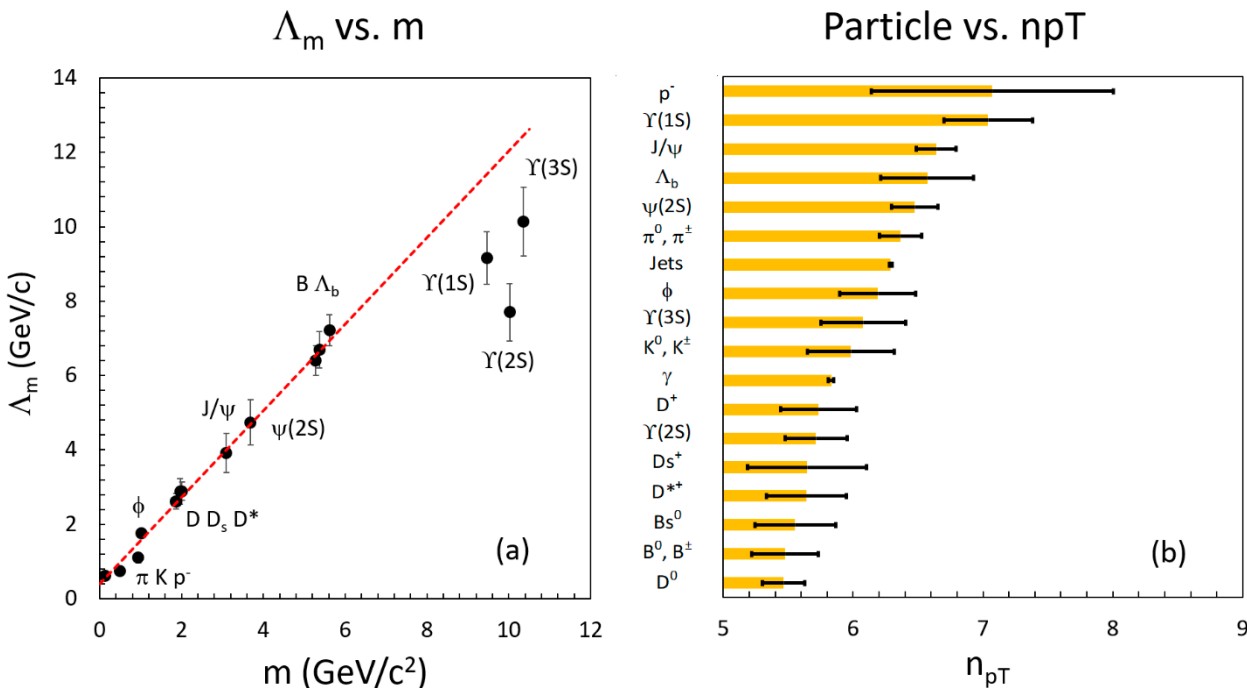

**Figure 20.** Shown in (**a**) is the transverse mass parameter $\Lambda_m$ as a function of the single particle mass in single particle inclusive production as illustrated in the fit for B$^\pm$ shown above. The red dotted line represents a fit to the data, not including the $\Upsilon$(nS) data resulting in $\Lambda_m$ = (1.17 ± 0.04) m + (0.40 ± 0.04) where $\Lambda_m$ is in momentum units (GeV/c) and m is in mass units (GeV/c$^2$). The photon point, where m = 0 with $\Lambda_m$ = 0, was not included in the fit. The LHCb $\Upsilon$(nS) data are inconsistent with the general trend for lower masses suggesting that the $\Lambda_m$ –m relation breaks down for high mass. In (**b**) are the $n_{pT}$ values of the corresponding $A(P_T)$ fits in ascending magnitude vs. the particle species. The photon and jet points are the values of the spine fits—hence the small errors.

Heavy quark pair production is sensitive to the gluon distribution of the proton, quark masses in the low $p_T$ region and is a laboratory for testing QCD. There is a large body of work in simulating the inclusive cross sections for heavy quark production, such as the FONNL code [5]. LHCb data taken on heavy meson production are especially interesting in the low $p_T$ region where the $\Lambda_m$ term is important. For example, in this low $p_T$ regime, not only is the intrinsic transverse momentum of the partons potentially important, but also the very low-x parton behavior is critical. (The 7 TeV LHCb B data probes down to x ~ 5 × 10$^{-3}$.) In the simulations, there are large $\ln(p_T/m)$ terms that must be resumed. Additionally, higher-order $\alpha_s{}^3$ terms are important at low $p_T$.

In the spirit of the discussion of the $p_T$ distributions of inclusive jets and photons given above, it is interesting to see whether there is an underlying simplicity in the measured cross sections that would be evidence of the initial hard parton–parton scattering with appropriate mass terms considered. One simplicity already evident has been shown in Figure 20 that indicates a linear relationship between the effective mass term $\Lambda_m$ which makes the $P_T \equiv \sqrt{p_T^2 + \Lambda_m^2}$ distribution a pure power law.

As an example, we study the $A(p_T, \sqrt{s}, \Lambda_m)$ behavior of B$^\pm$ measured by the LHCb collaboration at the LHC (see Figure 21). From equations in Table 11, the a function becomes quite flat in $p_T$ for small $p_T$ since the modified transverse momentum $P_T$ is essentially constant ~ m ($\Lambda_m$). Our Toy MC simulates the flat region at very low $p_T$ due to this transverse mass effect. The simulation also shows that the power law index of 1/ $P_T$ is smaller than that of inclusive jets following the $p_T$ dependence of the underlying parton–parton hard scattering. As before, we note that the value of $n_{pT}$ is not dependent on the details of the fragmentation (no fragmentation/fragmentation ~1) but the value of $\Lambda_m$ does

depend on fragmentation in ourIe model since the lower $p_T$ values following fragmentation can fall below the lower $p_T$ cut.

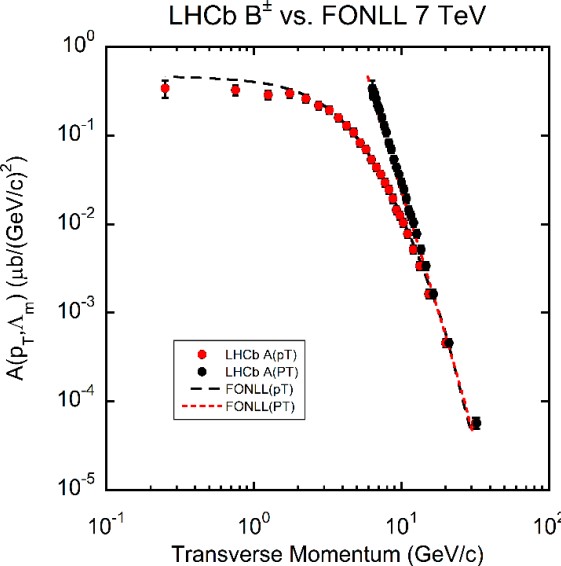

**Figure 21.** The resulting $A(\sqrt{s}, p_T, \Lambda_m)$ function for the FONLL simulation of inclusive production of B-mesons at $\sqrt{s} = 7$ TeV (orange diamonds and red dotted line) superimposed on the LHCb data (red and black circles) for MC and data plotted vs. the transverse momentum $P_T$ and the modified transverse momentum, $P_T$. The MC is in good agreement with the data and both MC and data follow a power law in the modified transverse momentum, $P_T$.

Our toy model is quite simple and differs from data in several significant ways. One discrepancy is that the resultant power of $P_T$ is larger than that of the data, although smaller than the corresponding jet value, where the toy model is successful in simulating the power indices $n_s$ and $n_{pT}$. In the simulation, the QCD coupling was evolved by $\hat{s}$ and the input mass of the b-quark was set to 4.75 GeV/$c^2$. The salient points of our toy formulation of inclusive reactions captures the underlying power law in $P_T$ expected from $gg \rightarrow Q\overline{Q}$ and $q\overline{q} \rightarrow Q\overline{Q}$ hard scattering to be smaller than that of inclusive jets and the suppression of the low $p_T$ values of $A(\sqrt{s}, p_T, \Lambda_m)$ by the heavy quark mass terms in the hard-scattering cross sections.

Unlike our Toy MC simulations, higher-order effects are considered in the FONLL program [5]. Its application to LHCb inclusive B$^\pm$ data is shown in Figure 21, where we find that the $\Lambda_m$ parameter in the modified transverse momentum is larger than the PDG rest mass (m value) [26] as in data and the $n_{pT}$ parameter is smaller than that of u-d-s-g jets as expected. The simulated values for LHCb measurements of B-mesons at 7 TeV yields $\Lambda_m = 5.9 \pm 0.3$ GeV/c and $n_{pT} = 5.6 \pm 0.2$ and for D$^0$ mesons at 13 TeV determines $\Lambda_m = 2.9 \pm 0.2$ GeV/c and $n_{pT} = 5.7 \pm 0.1$, both consistent with data (B: $\Lambda_m = 6.3 \pm 0.3$ GeV/c, $n_{pT} = 5.5 \pm 0.2$; D: $\Lambda_m = 2.7 \pm 0.1$ GeV/c, $n_{pT} = 5.3 \pm 0.1$).

Furthermore, it is interesting to observe that $A(\sqrt{s}, p_T, \Lambda_m)$ functions for LHCb B$^\pm$, B$^0$, B$_s^0$ mesons [45], after the appropriate $\Lambda_m$ corrections ($\Lambda_m = 6.3 \pm 0.3$ GeV/c), and b-jets, measured by ATLAS at 7 TeV [63], have the same power law index, $n_{pT}$. The relation is shown in Figure 22 below where the b-jets have been normalized by an empirical factor of $1.4 \times 10^{-4}$. The red dotted line represents a minimum $\chi^2$ fit to the LHCb data and ATLAS b-jets combined ($\kappa = (9.2 \pm 0.7) \times 10^3$ $\mu$b GeV/c$^{npT-2}$ and $n_{pT} = 5.51 \pm 0.03$, $\chi^2$/d.f. = 14.3/52, $p = 1.0$). It is apparent that the three processes plotted have the same $p_T$ dependence (other than the normalization factor) suggesting that the soft processes in b-jet formation and those in the fragmentation of the b-quark to B$^0$, B$^\pm$ hadrons have little effect on the $p_T$ distributions of the a function. This is one of the salient simplifying powers of the a function.

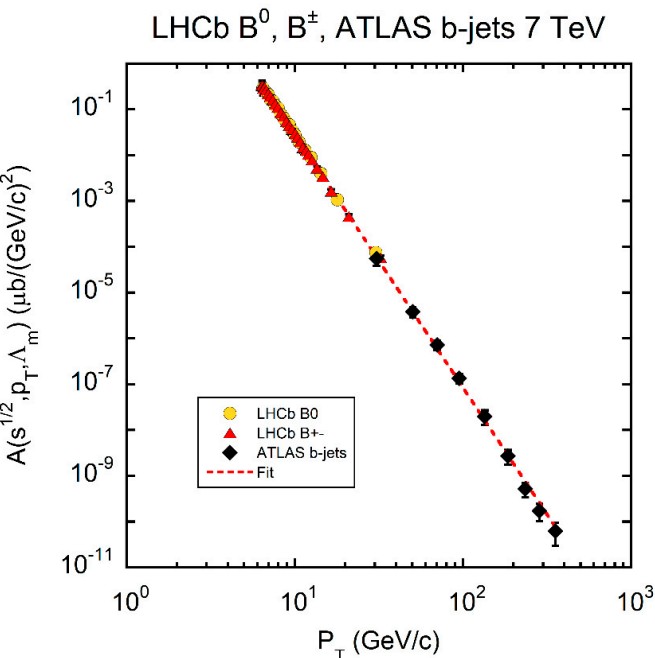

**Figure 22.** Comparison of the $A(P_T)$ distributions of b-jets (black diamonds) measured in ATLAS normalized by an empirically determined multiplicative factor of $1.4 \times 10^{-4}$ with LHCb B-meson data (B$^0$ and B$^\pm$ shown as yellow circles and red triangles, respectively). Each data set is plotted versus the modified transverse momentum $P_T = (p_T^2 + \Lambda_m^2)^{1/2}$. The red dotted line is a simultaneous power law fit in the modified transverse momentum, $P_T$, of all three data sets resulting in $n_{pT} = 5.51 \pm 0.03$ and $\Lambda_m = 6.3 \pm 0.3$ GeV/c ($\chi^2$/d.f. = 14.3/52, p = 1.0).

Our formulation of the invariant cross sections in terms of the a function and the $[p_T - x_R]$ sector enables such a comparison to be made between diverse data sets. In fact, given the common $n_{pT}$ value for B$^0$, B$^\pm$ and b-jets demonstrated in Figure 22, and the dimensional custodial to be discussed in Section 7, we expect that all three processes will have $n_s = 1.24 \pm 0.05$ making their respective a functions grow with increasing $\sqrt{s}$ as $(\sqrt{s})^{(1.24 \pm 0.05)}$.

### 6.2. $X_R$ Dependence of Heavy Particle Production

The $n_{xR}$ behavior for inclusive B$^\pm$ production as measured by the LHCb collaboration [45] is shown in Figure 23. The data have been analyzed in the same manner as the inclusive jets and inclusive photons discussed above. In the analysis, we have used the PDG [26] rest mass value of the B$^\pm$ meson ($5.27929 \pm 0.00014$ GeV/c$^2$) for the expression for $x_R$, and the $\Lambda_m$ term of the $p_T$ distribution was set to the measured value $\Lambda_m = 6.3 \pm 0.6$ GeV/c as shown in Figure 20.

From the figure, we note that $n_{xR}(p_T)$ for inclusive B production is quite different from that of jets (Figure 5) and that of direct photons (Figure 16), but the momentum regions of the measurements are quite different. We observe that the FONLL [5] simulation shown in Figure 23 overestimates the power $n_{xR}$ at low $p_T$ although the experimental errors are large.

Since we have observed that the $P_T$ distributions of B$^\pm$ production at the LHCb and b-jets as measured by the ATLAS collaboration are consistent as shown in Figure 22, it is interesting to see how the $F(\sqrt{s}, x_R)$ functions compare. In Figure 24, we plot on the left the 7 TeV LHCb B$^\pm$ inclusive data and on the right b-jets measured at 7 TeV by the ATLAS collaboration. In both cases, the F-functions were determined in the same manner as those for inclusive jet production discussed above, but with the simplification of setting the $D_Q$ and $n_{xRQ0}$ terms to zero since the data are not extensive enough for good estimates of their values. Notice that the two F distributions in Figure 24 are nearly the same—in fact in

terms of Equation (16) with $n_{xRQ0} = 0$, we find $n_{xR0} = 14.0 \pm 0.4$ for B$^{\pm}$ and $n_{xR0} = 12 \pm 3$ for b-jets—in agreement, but very different from light parton jets indicated by the blue dotted line in the figure on the left ($n_{xR0} = 4.0 \pm 0.5$, $n_{xRQ0} = 0.7 \pm 0.2$). Applying a $\chi^2$ test of the b-jet fit to B$^{\pm}$ *F*-function, we find $\chi^2 = 50$ for 134 d.f.; and for B$^{\pm}$ with itself, $\chi^2 = 101$ for 134 d.f. Similarly, applying the fit of B$^{\pm}$ *F*-function to b-jets, we find $\chi^2 = 91$ for 35 d.f.; and for b-jets with itself, $\chi^2 = 67$ for 35 d.f. The $F(7, x_R)$ for g-u-d-s jets discussed above is much different from b-jets.

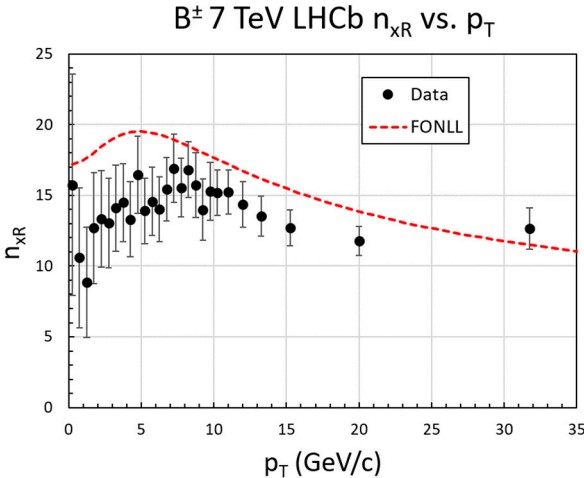

**Figure 23.** The power terms, $n_{xR}$ for $(1 - x_R)^{nxR}$ are plotted vs. $p_T$ for LHCb B$^{\pm}$ data (black points) and for the FONLL simulation (red dotted line). The errors were computed from the experimental statistical and systematic errors added in quadrature and the fitting errors in the determination of $n_{xR}(p_T)$.

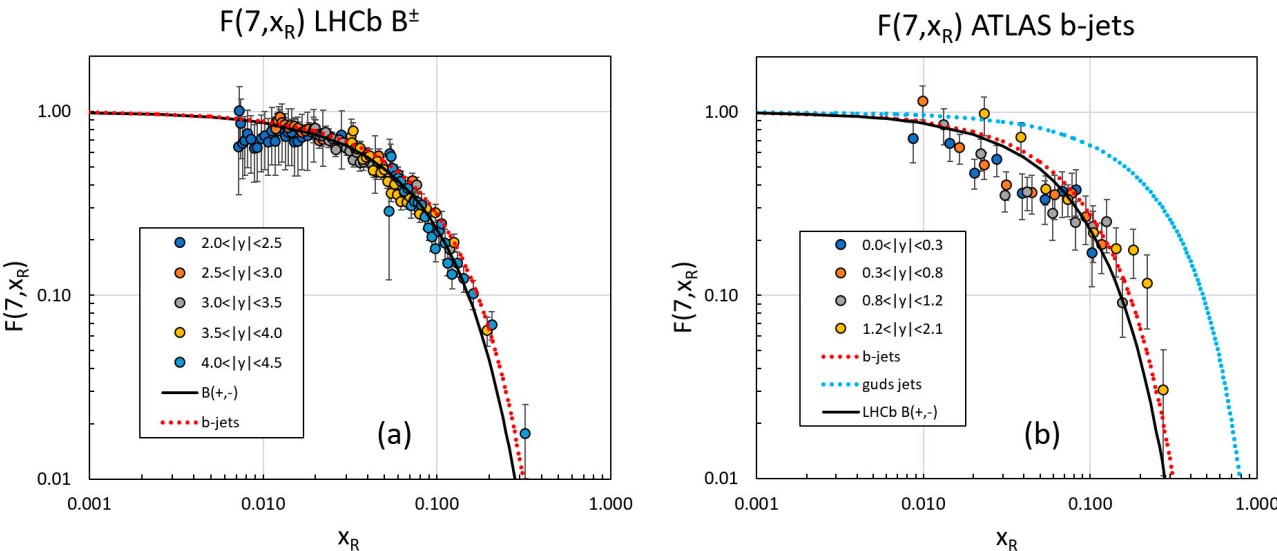

**Figure 24.** On the left, (**a**), we show the $F(7, x_R)$ function for B$^{\pm}$ data measured at 7 TeV by the LHCb collaboration and on the right (**b**) the b-jet $F(x_R)$ measured by ATLAS at the same $\sqrt{s}$. Notice that the rapidity and $p_T$ regions of the LHCb and ATLAS data sets are disjoint. We have plotted LHCb vs. ATLAS: $2.0 \leq |y| \leq 4.5$ vs. $0 \leq |y| \leq 2.1$, and $0.25 \leq p_T \leq 31.75$ GeV/c vs. $30 \leq p_T \leq 355$ GeV/c, respectively. The $x_R$ regions of the two functions are roughly the same, $0.008 < x_R < 0.3$. The *F*-functions for B$\pm$ and b-jets are very different from g-u-d-s jets which are indicated by the blue dotted line in the right-hand graph.

The observed consistency of the a functions, determined by the invariant differential cross section extrapolation $x_R \to 0$, for inclusive B$^{\pm}$ and b-jets suggests that the underlying

parton–parton scatterings for the two processes are the same. What is noteworthy is that the *F*-functions are also quite similar but quite different for those of light quark/gluon jets despite the fact that soft processes, such as fragmentation and hadronization, are at work. In the case of $B^{\pm}$ production, the b-quark has to hadronized into a B-meson, whereas for b-jets, there only has to be collimated gluon and quark radiation around the struck b-quark direction to form a jet. The steeper fall-off as the kinematic boundary is approached is indicative of the dominance of gluons and sea quarks in the production process.

## 7. Analysis of Z-Boson Inclusive Production

The production of the Z-boson in p–p collisions is one of the important tests of the standard model in that the production cross section involves not only QCD physics but also the electroweak sector. The production cross section is usually thought of as a continuum of the Drell–Yan process, where an initial state quark–antiquark pair annihilates to a heavy $J^{PC} = 1^{--}$ state to become a Z-boson. On the other hand, Z-boson production is also related to direct photon production, where, for example, in the process $q + \bar{q} \rightarrow Z + g$ the 'heavy photon' in the final state becomes the Z-boson and the radiated gluon provides a transverse momentum kick to the Z that would not be present in simple quark–antiquark annihilation with no gluon radiation. Since we have already analyzed some of our properties of vector meson production, such as $J/\psi$, $\psi(2S)$ and $\Upsilon(nS)$ and direct photon production, it is of interest to analyze the inclusive Z-boson production with our radial scaling phenomenology.

Following Schott and Dunford [64], who have reviewed Z-boson production at 7 TeV, there are two regions of the spectrum of the transverse momentum, $p_T$ of the Z-boson that have distinct signatures. In the high $p_T \gg M_z$ region, the cross section is expected to be of the form:

$$\frac{d^2\sigma}{dp_T^2} \sim \frac{\alpha_s(p_T^2)}{p_T^4} \tag{46}$$

which is at the dimensional limit of the inclusive cross section $\sim 1/p_T^4$. As we will see in the next section, this $p_T$ dependence implies a slow growth of the a function magnitude $\kappa(s)$ with $\sqrt{s}$. In the intermediate transverse momentum range, where the Z-boson transverse momentum is larger than the intrinsic parton transverse momentum ($k_T \sim 0.7$ GeV/c) $k_T < p_T < M_z/2$, gluon emission is important in the initial quark–antiquark state. When the gluon is colinear with the incoming quark or antiquark line, the effect of gluon emissions can become quite large and has to be treated by a resummation technique (i.e., "Sudakov form factor"). Again, following Schott and Dunford [64] the normalized Z-boson cross section at low $p_T$ becomes:

$$\frac{1}{\sigma}\frac{d^2\sigma}{dp_T^2} \sim \frac{d^2}{dp_T^2}\left(\exp\left[\frac{-2\alpha_s}{3\pi}\ln^2\left(\frac{M_Z^2}{p_T^2}\right)\right]\right) \tag{47}$$

The exponential term imposes a large damping at small $p_T$ of the cross section by 'robbing' energy of the annihilating quark–antiquark collision, thereby pushing the production of the Z-boson closer to its $\sqrt{\hat{s}}$ threshold. The Z-boson a function shows these two characteristics—a suppression at low $p_T$ controlled by colinear gluon emission and an emergent $\sim 1/p_T^4$ power law at high $p_T$.

In Figure 25, we show the a functions for inclusive $Z/\gamma^*$ production at $\sqrt{s} = 8$ [65] and 13 TeV [66] measured by the CMS collaboration and a measurement by the ATLAS collaboration at 7 TeV [67]. The 8 TeV data correspond to an integrated luminosity of 19.7 fb$^{-1}$ and range $0 \leq |y| \leq 2.4$ and $10 \leq p_T \leq 600$ GeV/c central bin values by measuring the $Z \rightarrow \mu^+ \mu^-$ channel. Additionally, the CMS 13 TeV data of the absolute inclusive cross section for $Z \rightarrow \mu^+ \mu^-$, $e^+ e^-$ cover the $0 \leq |y| \leq 2.4$ region in five bins and range over $0.5 \leq p_T \leq 950$ GeV/c in thirty four bins, corresponding to an integrated luminosity of 35.9 fb$^{-1}$. The 7 TeV ATLAS data (4.7 fb$^{-1}$) of the double differential cross section has been normalized by a fiducial cross section and cover $1.0 \leq p_T \leq 800$ GeV/c in

26 bins with $0 \leq |y| \leq 2.4$ in three bins. These extensive data enable us to calculate the a functions as well as determine t–e $(1 - x_R)$ power index function, $n_{xR}(p_T)$.

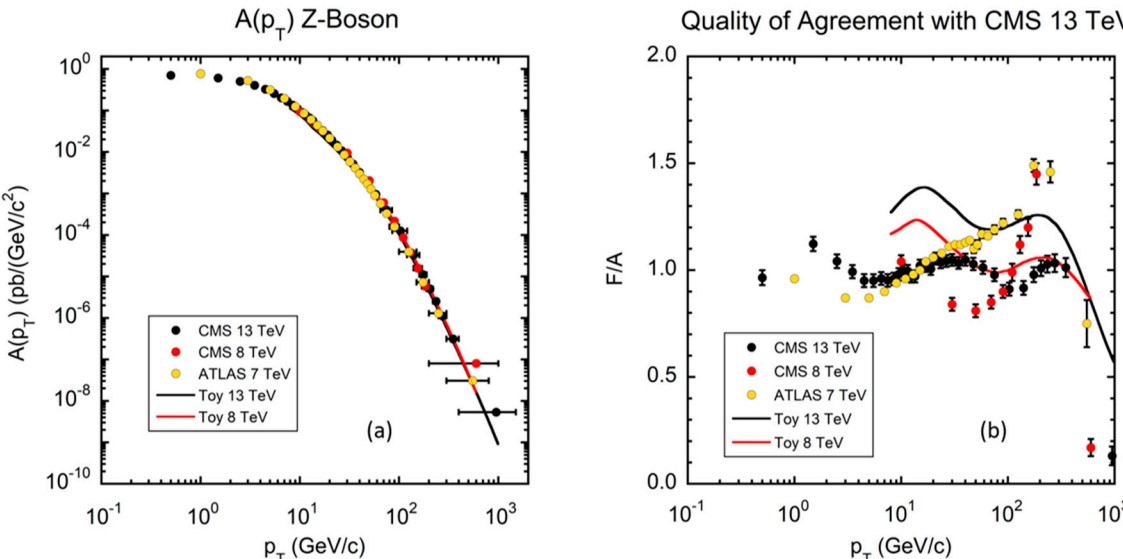

**Figure 25.** In the left plot, (**a**), the a function of inclusive Z-boson production measured by the CMS collaboration at 8, 13 TeV are compared with the ATLAS 7 TeV data and Toy MC simulations at 8 and 13 TeV. All data are normalized to the CMS 13 TeV data set. The $p_T$ bin widths have been plotted for the highest ~5 bins of the data. There are two salient features of the a functions apparent in the plots: (1) the turnover at low $p_T$ and (2) the emergent power law behavior at large $p_T$. On the right (**b**) are shown the ratio of the normalized data and MC to 13 TeV CMS data set. The black circles in the right plot indicate the quality $^{\text{of}}$ a 4th-order log(A)-log($p_T$) polynomial fit to 13 TeV data used to compare data sets. The overall agreement is within $\pm 50\%$ for all quality measures (fit, data-data, data-MC) except at highest $p_T$ points.

One might think that the $\Lambda_m$ –m relation, shown in Figure 20a, would be operative for inclusive Z-boson production but we find that relation to be strongly broken. The data have a turnover at low $p_T$ that cannot be 'corrected' by a single value of $\Lambda_m$ consistent with gluon radiation in the initial state. However, we do find that the a functions are consistent with a $p_T$ power law at large $p_T \geq 100$ GeV/c. Fitting the region $102.5 \leq p_T \leq 950$ GeV/c for the CMS 13 TeV data set, we find $A(p_T) \sim 1/p_T^{np_T}$, where $n_{pT} = 4.68 \pm 0.03$ ($\chi^2$/d.f. = 112/6 d.f., p ~ 0.)—close to the dimensions $(\text{GeV/c})^{-4}$ required by the definition of the invariant cross section, albeit with a large $\chi^2$ value since there was no finite bin correction for the largest $p_T$ bin.

The a function for Z-boson inclusive production can be approximated by treating the Z-boson as a heavy photon in the process $q + \bar{q} \to Z + g$, as noted above. We have deployed our direct photon simulation but modified the kinematics to accommodate the mass of the Z-boson and have ignored the Z-boson width. As with the direct photon simulations, we have used the CT10 parton distributions with QCD scale set to $Q^2 \sim s$ and considered all $q\bar{q}$ channels weighted by their respective Z-boson branching fractions. The results of the simulation are shown in the figure as red (8 TeV) and black (13 TeV) lines normalized to the data. In the momentum range $p_T(Z) > 7$ GeV/c and neglecting the very highest $p_T$ point of the three measurements, the R(Toy/Data) = $1.0 \pm 0.5$ over up to 6 orders of magnitude. We conclude that $p_T > 7$ GeV/c regions of the $A(p_T)$ functions can be roughly simulated by the heavy photon approximation with gluon emission and have little shape change as a function of $\sqrt{s}$.

A much more demanding test of the data is to examine the power $n_{xR}(p_T)$ as in $(1 - x_R)^{nxR}$ as a function of $p_T$. Our Toy simulation indicates that $n_{xR}(p_T)$ is quite sensitive to the mixture of $q\bar{q}$ annihilations contributing to the production cross section. Thus, the simulation of the $n_{xR}(p_T)$ dependence, just as in inclusive jet production, is a much more stringent test of theory than the $A(p_T)$ function.

We have deployed only the single power $n_{xR}$ analysis since the 8 TeV CMS and 7 TeV ATLAS data are insufficient to determine the quadratic term, $n_{xRQ}(p_T)$, as in Equation (9). Furthermore, the ATLAS 7 TeV data set has only three |y| points so the determination of the $x_R$ behavior is minimal. When we analyze the $x_R$ dependence of these three data sets we find a large s dependence in the structure of $n_{xR}(p_T)$ as a function of $p_T$. Not surprisingly, our simulation of the $n_{xR}(p_T)$ by the Toy MC fails badly. For example, in the 13 TeV case, the TMC predicts $n_{xR}(p_T) \sim 6$ and independent of $p_T$, whereas the CMS data fall from $n_{xR} \sim 8$ at $p_T \sim 5$ GeV/c to $n_{xR} \sim 1$ for $p_T > 200$ GeV/c, with shape changes in between. Remember that the $A(p_T)$ function is insensitive to 'soft physics', whereas the $x_R(p_T)$ dependence is sensitive to both 'hard' and 'soft' physics. Hence, it is possible to have agreement of the Toy MC simulation for $A(p_T)$, but disagreement with data for $x_R$ behavior. This disagreement begs for a full simulation of Z production, with all the QCD, parton PDF and electroweak effects operative, analyzed in the $[p_T - x_R]$ framework with finer |y| and $p_T$ binning and standardized background corrections.

In summary, the study of Z-boson inclusive production in our $[p_T - x_R]$ framework offers stringent test of the modeling. Measurements of the double differential cross section in bins of $p_T$ and $y$ with standardized cuts and signal definitions are mandated. Others have made this point clear [68,69].

## 8. The Dimensional Custodian

The inclusive invariant cross sections, $d^2\sigma/(2\pi p_T dp_T dy)$ have the dimension cm$^2$/(GeV/c)$^2 \sim 1/$(GeV/c)$^4$—by definition. It is therefore interesting to see how this dimension is maintained by computing the logarithmic derivatives with respect to all the energy/momentum variables. (We take energy and momentum to be equivalent when measured in either GeV or GeV/c with c = 1). The sum of the resultant power indices should compute to the dimension of the cross section, that is, the sum of the power indices should result in the value – 4, the dimension of the invariant cross section in units of GeV/c.

In order to refine the analysis for inclusive jets and photons, we deploy another power index, introduced in Table 6, to explicitly express the s dependence of the a functions. In that terminology, we write $\kappa(s) = \kappa_0 (\sqrt{s})^{ns}$, with $n_s$ being the power index of its $\sqrt{s}$ dependence. We then fit the normalized a functions of data taken at different $\sqrt{s}$ values to the form:

$$S_{j,\gamma}(p_T) \equiv \frac{A(\sqrt{s}, p_T)}{(\sqrt{s})^{ns}} = \frac{\kappa_0}{p_T^{npT}} \, , \tag{48}$$

where $n_s$, $\kappa_0$ and $n_{pT}$ are fit parameters with $n_{pT}$ being the usual a function power index. We call these normalized $p_T$ distributions the spine functions for jets and photons. They should be approximate functions of only $p_T$, having eliminated the s dependence by the power index, $n_s$.

The a functions so normalized for ATLAS inclusive jets is shown in Figure 26. We expect that a determination of the power indices, $n_{pT}$ and $n_s$, of the spine functions to be a straightforward test of the dimensional constraint that the invariant cross section, as expressed by the a functions, should have the dimension of [(GeV/c)$^{-4}$]. Thus, the power indices, $n_{pT}$ and $n_s$, for both inclusive jets and photons, respectively, should obey the dimensional constraint equation: $n_{pT} - n_s = 4$. However, when we measure this simple constraint equation we find that it is not satisfied and discovery (not unexpected) that the residual power of $p_T$ forcing the dimensional constraint is in fact directly related to the QCD evolution of the distribution functions of the colliding partons and thus to the strong coupling constant $\alpha_s(Q^2)$. Therefore, the spine functions provide a rough test of the Q$^2$ evolution of the colliding PDFs and coupling constant controlled by QCD.

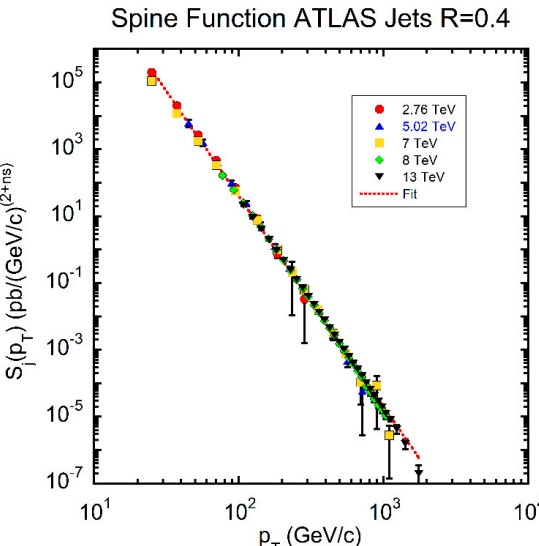

**Figure 26.** The measured $p_T$ dependence $A(\sqrt{s}, p_T)$ of inclusive jets (R = 0.4) measured at the LHC by ATLAS divided by $s^{ns/2}$ (called the 'spine function') is shown. The data are: $\sqrt{s}$ = 2.76 TeV red circles, 5.02 TeV inverted blue triangles, 7 TeV yellow squares, 8 TeV green diamonds, 13 TeV black triangles. Note that all the data follow the same power law with $n_{pT}$ = 6.29 ± 0.01. The error bars represent the statistical and systematic errors of the measurements added in quadrature. The resulting errors are mostly smaller than the plotted points. The red dotted line represents $A(\sqrt{s}, p_T)/s^{ns/2} \sim \kappa_o/p_T^{6.3}$, where $n_s$ = 2.0 ± 0.1. The data follow this power law with residuals $\leq \pm 50\%$ over 12 orders of magnitude. For plotting purposes, the error bars for 5 points were limited to no larger than 95% of the point's value.

Combining the $n_s$ power index with the value of $n_{pT}$ and demanding that the dimension of the inclusive jet cross section $d^2\sigma/(2\pi p_T dp_T dy) \sim 1/(\text{GeV}/c)^4$, we find $n_{pT} - n_s - 4 \neq 0$ but rather that there is a non-zero residual power $n_r = n_{pT} - n_s - 4 = (6.29 \pm 0.02) - (1.99 \pm 0.04) - 4 = 0.30 \pm 0.04$ (7.5σ). Hence, the simple dimension equation is not satisfied without the residual power $n_r$. For inclusive photons, we find similarly that the residual power $n_r = n_{pT} - n_s - 4 = (5.83 \pm 0.02) - (1.56 \pm 0.04) - 4 = 0.27 \pm 0.05$ (5.4σ).

We have checked our understanding of the spine function for inclusive jets by calculating the function for both Toy MC as well as Pythia 8.1 simulations. The comparison of data with these two simulations is shown in Table 12 below. We note that the consistencies of the simulations with data are reasonably good but that the simulated value of the residual power, $n_r$, tends to be smaller than the data. Since the determination of the residual power involves a subtraction of two experimental (simulated) numbers, the quoted error is probably smaller than the actual error since we have not included the luminosity errors, $p_T$ finite binning, and other experimental details in the simulations.

The spine functions follow power laws in $1/p_T$ to a good approximation. For inclusive jets, we find the residuals of the $1/p_T$ power law fit to data are $\leq \pm 50\%$ over 12 orders of magnitude. It is somewhat of a surprise that the normalized a functions are so congruent, since in the primordial parton hard-scattering cross sections the PDF and $\alpha_s(Q^2)$ evolutions are operative. However, we note that at very high energy the QCD $Q^2$ evolution becomes smaller fractionally. In fact, a close examination of the residuals of each data set that comprise the spine functions reveals that they have a systematic pattern of negative residuals at the low and high $p_T$ limits of the distribution and positive residuals between the two $p_T$ extremes as shown in Figure 18 of our earlier publication [9]. The peak position of the positive residual $p_T$ (max residual) is roughly the hyperbolic mean of the lowest and highest momentum of the distribution $\sim <p_T> = \sqrt{p_{T\min}p_{T\max}}$. Both Pythia 8.1 and our Toy MC follow this behavior.

**Table 12.** Spine function parameters for inclusive jets. Data are compared with Toy MC and Pythia 8.1 simulations. The Toy simulation had the relative errors of simulated invariant cross section 'data' points fixed to 2%, whereas the errors of the Pythia 8.1 simulation itself were used. Since the MCs have no absolute normalizations, the values of $\kappa_0$ for them are arbitrary and therefore not tabulated. The MC fits ranged from $2.76 \leq \sqrt{s} \leq 13$ TeV matching the range of data. The $p_T$ and $|y|$ binning of the MC simulations were forced to be the same as data.

| Inclusive Jets | Parameter | Value |
|----------------|-----------|-------|
| Data | $\kappa_0$ | $(10 \pm 3) \times 10^5$ (pb GeV$^{(npT-ns-2)}$) |
| Data | $n_s$ | $1.99 \pm 0.04$ |
| Toy MC | $n_s$ | $2.084 \pm 0.004$ |
| Pythia 8.1 MC | $n_s$ | $2.028 \pm 0.005$ |
| Data | $n_{pT}$ | $6.29 \pm 0.01$ |
| Toy MC | $n_{pT}$ | $6.286 \pm 0.002$ |
| Pythia 8.1 MC | $n_{pT}$ | $6.243 \pm 0.003$ |
| Data | $n_r$ | $0.30 \pm 0.04$ |
| Toy MC | $n_r$ | $0.203 \pm 0.004$ |
| Pythia 8.1 MC | $n_r$ | $0.216 \pm 0.005$ |
| Data | $\chi^2/$d.f. | $221/94$ ($p = 3 \times 10^{-12}$) |

We proceed in the same manner as we followed for inclusive jets for the analysis of direct photon data taken at 8 TeV [36] and 13 TeV [37] by the ATLAS collaboration in order to determine the spine function for direct photons. (There are ATLAS data at 7 TeV [38], but they cover only three $|y|$ bins, thereby precluding a full $n_{xR}$ and $n_{xRQ}$ analysis.) For each $\sqrt{s}$ value we determined the $A(\sqrt{s}, p_T)$ function as in Equation (3) with $\Lambda \equiv 0$. These $A(\sqrt{s}, p_T)$ values were simultaneously considered by using Equation (36) in a global fit to determine the three spine parameters $\kappa_0$, $n_s$ and $n_{pT}$. The resulting spine function for direct photon production is shown in Figure 27. As in the case of the jet spine function analysis, we display the spine function normalized to the 13 TeV data set. The resulting fit values are given in Table 13. We applied the normalization parameters determined by the ATLAS data to the plotted UA1 data [70] but did not include them in the fit.

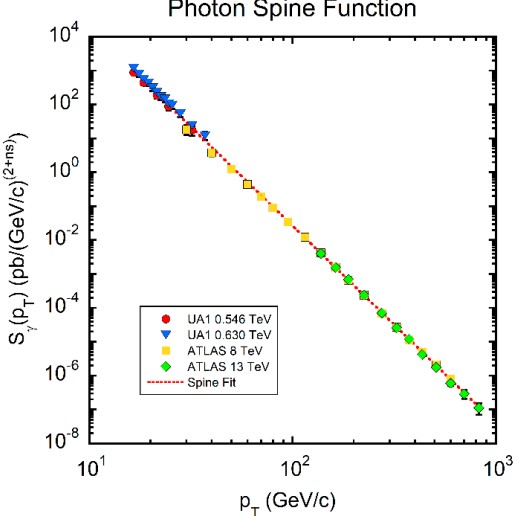

**Figure 27.** The spine function for direct photons as measured by the ATLAS and UA1 Collaboration. Data from 8 [36] and 13 TeV [37], yellow squares, green diamonds, respectively, are plotted. The UA1 data [70] are represented by the red circles, blue triangles for $\sqrt{s}$ = 546 and 630 GeV, respectively. Error bars are plotted but are generally smaller than the symbols of the plotted points. The red dotted line represents the global fit to the ATLAS data sets. The $\chi^2/$d.f. of the red dotted line through all the plotted points is 48/45 ($p = 0.35$). Note that the UA1 $\sqrt{s}$ = 0.546 TeV data have been normalized with respect to the ATLAS 13 TeV data by a computed factor of $(13/0.546)^{1.56} \sim 141$.

**Table 13.** Spine function parameters for inclusive photons. Data are compared with Toy MC simulation through the parameters $n_s$, $n_{pT}$ and $n_r$. The MC simulation has the relative errors of invariant differential cross section fixed to 2%. The simulation was from $2.76 \leq \sqrt{s} \leq 13$ TeV, whereas the data were $8 \leq \sqrt{s} \leq 13$ TeV. Since the MC had no absolute normalization, the values of $\kappa_0$ for them are arbitrary and therefore not tabulated. The $p_T$ and y binning of the MC simulation were forced to be the same as data.

| Inclusive Photons | Parameter | Value |
|---|---|---|
| Data | $\kappa_0$ | $(5 \pm 2) \times 10^3$ (pb $(\mathrm{GeV}/c)^{(\mathrm{npT}\text{-}\mathrm{ns}-2)}$) |
| Data | $n_s$ | $1.56 \pm 0.04$ |
| Toy MC | $n_s$ | $1.08 \pm 0.01$ |
| Data | $n_{pT}$ | $5.83 \pm 0.02$ |
| Toy MC | $n_{pT}$ | $5.30 \pm 0.01$ |
| Data | $n_r$ | $0.27 \pm 0.05$ |
| Toy MC | $n_r$ | $0.22 \pm 0.01$ |
| Data | $\chi^2/\mathrm{d.f.}$ | $44/27$ ($p = 0.02$) |
| Toy MC | $\chi^2/\mathrm{d.f.}$ | $277/95$ ($p = 1 \times 10^{-19}$) |

The residual power for direct photons is expected to be smaller than that of inclusive jets because the underlying hard parton scattering cross sections are dependent on the product $\alpha_e \, \alpha_s$ rather than $\alpha_s{}^2$. Because $\alpha_e(Q)$ has a negative $\beta$ function forcing the coupling to grow stronger as the Q scale increases, whereas $\alpha_s(Q)$ has a positive $\beta$ function making the coupling weaker with increasing Q scale, resulting in the residual power for inclusive photons to be smaller. Unfortunately, the data are not good enough to determine this difference by this method. By the fitted parameters above, we find the residual power for photons $n_r = 0.27 \pm 0.05$ (5.4$\sigma$).

Referring to Table 13 for the spine function for direct photon production, we find the photon $n_{pT}$ and $n_s$ values significantly smaller than the inclusive jet values. The residual dimensions, $n_r$, are, however, consistent within errors.

The dimensional constraint fixed by the underlying parton–parton hard scattering is further demonstrated by performing similar Toy MC analyses for various parton distribution assumptions at various $\sqrt{s}$ as tabulated in Tables 2 and 8. We find that these different (toy) processes have different $n_{pT}$ and $n_s$ values allowing us to explore the $n_{pT} - n_s$ relation. In Figure 27, we plot the resultant $n_{pT}(n_{ET})$ vs. $n_s$ values as well as the measured $n_{pT} - n_s$ pair values for inclusive jets and inclusive photons described above. It is obvious that the dimension of the data cross section as well as that of the MC is $\sim 1/(\mathrm{GeV}/c)^4$ in the factorized form for whatever process. A linear fit to the power indices correlation (jet data and MC and photon data and its MC) shown in Figure 28 yields the custodial relations for inclusive jets and photons that are:

$$\begin{aligned} n_{pT}(jets) &= (1.00 \pm 0.02)n_s + 4.28 \pm 0.03 \\ n_{pT}(\gamma) &= (1.05 \pm 0.03)n_s + 4.16 \pm 0.04, \end{aligned} \tag{49}$$

where $n_{pT}$ is the $p_T$ power index of the a function and $n_s$ is the power index of the $\sqrt{s}$ dependence of the magnitude parameter $\kappa(\sqrt{s}) = \kappa_0(\sqrt{s})^{n_s}$ of the a functions. The slopes of both custodial relations are consistent with 1.0 within errors, whereas the intercepts differ by $0.12 \pm 0.5$, or about 2.4$\sigma$, or in ratio $n_r(jets)/ \, n_r(\gamma) = 1.7 \pm 0.7$ consistent with the evolution of $\alpha_s(Q^2)^2$ versus $\alpha_s(Q^2)$.

Thus, both the $p_T$ power law index, $n_{pT}$, and the s dependence, $\kappa_0(\sqrt{s})^{ns}$ with a power index $n_s$ are coupled. The dimensional custodial is the result of the constraint of the dimensions of the invariant cross section being $1/(\mathrm{GeV}/c)^4$ that is the same dimension of the a function and the same dimension of the underlying parton–parton cross section. The residual power for inclusive jets, determined by combining data with MC, is $n_r = (4.28 \pm 0.03) - 4.0 = 0.28 \pm 0.04$ (9.2$\sigma$) and arises from the $p_T$ evolution of the PDFs and the strong coupling constant analyzed as an approximate power law in the low $p_T$

region of the data. For inclusive photons, the residual power by combining data and MC, $n_r = 4.16 \pm 0.04 - 4 = 0.16 \pm 0.04\ (3.7\sigma)$—both consistent with the spine function analysis.

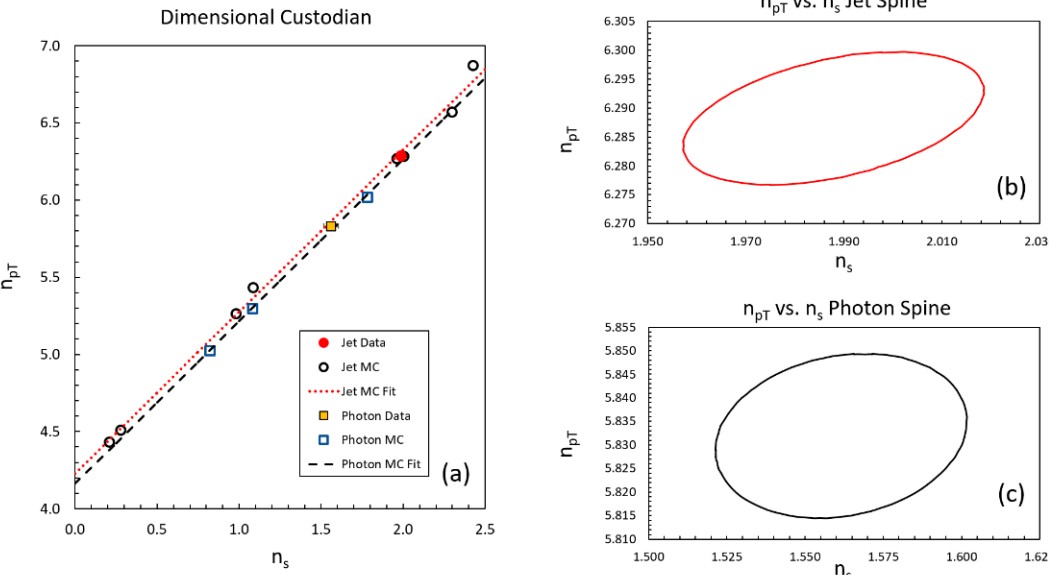

**Figure 28.** The power of $p_T$ of the amplitude of the $A(\sqrt{s}, p_T)$ function for inclusive cross sections is plotted against the power of the $\sqrt{s}$ dependence of the amplitude for data (jets red circle, photons yellow square) and simulations (jets open black circles, photons open blue squares) in (**a**). Note that all values of data and MC fall on the same line indicating that the dimensions $\sim 1/(\text{GeV}/\text{c})^4$ of the inclusive cross sections are always preserved but must include the residual $p_T$ dependence from QCD evolution of the PDFs and $\alpha s(Q^2)$ at the $n_s = 0$ intercept for jets (red dotted line) and dashed black line for photons. The MC points include toy simulations of dijets through the $g\,g \to g\,g$ and $q\,q \to q\,q$ channels with CT10 PDF [23] weighting as well as Pythia 8.1 inclusive jet simulations with R = 0.4 [27]. The Toy MC considered $1.96\,\text{TeV} \le \sqrt{s} \le 13\,\text{TeV}$ for the $n_s$ calculation and the Pythia 8.1 $n_s$ value was determined from $2.76\,\text{TeV} \le \sqrt{s} \le 100\,\text{TeV}$. The photon residual power is expected to be $\sim\frac{1}{2}$ that of jets since the QCD term is $\alpha_s(Q^2)$. In (**b**) is plotted the 1 $\sigma$ contour of the $n_{pT}$ vs. $n_s$ fit for jets and in (**c**) the corresponding contour for photons.

It is interesting to observe that the hard scattering of the partons, which occurs at the causal beginning of jet production, weighted by the participating PDFs, actually controls the s dependence of the jet and direct photon a functions by the dimensional custodian. This is another example of the utility of the a function—its dependences on $p_T$ and $\sqrt{s}$ is independent of the 'soft physics' of fragmentation and hadronization.

That the residual $p_T$ power is due to the $Q^2$ evolution of the PDFs through the DGLAP [71] equations and the explicit running of the value of coupling $\alpha_s{}^2(p_T)$ governing the size of the various parton–parton scattering cross sections, is easily demonstrated by our Toy MC with the $\alpha_s{}^2$ strength term of the hard-scattering cross sections set to a constant and assuming that the 13 TeV PDFs are operative at all $\sqrt{s}$ values. With these conditions, we find that the residual $p_T$ power for the simulated spine function for inclusive jets in the energy range $2.76 \le \sqrt{s} \le 13$ TeV is $n_r = 6.123 \pm 0.002 - 2.120 \pm 0.004 - 4 = 0.003 \pm 0.004$, namely zero.

We can cross check the residual power by analyzing $\alpha_s{}^2$ as a power law in $p_T$ at the minimum $p_{T\min}$ by the expression:

$$n_{pT}(residual) \approx \left(\frac{2p_{T\min}}{\alpha_s(p_{T\min})}\right)\frac{d\alpha_s}{dp_T} \sim 2.42\,\alpha_s(p_{T\min})\,, \tag{50}$$

where $\alpha_s(p_T)$ is evaluated at $p_{T\min}$ where the QCD scale evolution is the largest. For the ATLAS inclusive jet data ($p_{T\min} = 25$ to $100\,\text{GeV}/\text{c}$) we estimate that the residual power index ranges from $0.29 \le n_r \le 0.36$, which agrees with inclusive jet data $n_r = 0.30 \pm 0.04$.

Hence, all dimensional factors are accounted for in terms of the power indices of energy-momentum variables. In our earlier publication [9], we noted that $2 \to 3$ scattering processes have a natural $1/p_T^6$ behavior. We remarked that the existence of diquarks in the nucleon is consistent with a putative $2 \to 3$ scattering processes. Contrary to those earlier speculations [9], we show here that the $p_T$ behavior for jets, photons and heavy meson inclusive cross sections can be adequately explained by $2 \to 2$ processes controlled by hard-scattering cross sections $d\hat{\sigma}/d\hat{t}$, in the range $p_{T\min} \leq p_T \leq \sqrt{\hat{s}}/2$, where the cross sections are finite at each limit, weighted by the $\hat{s}$ distribution generated by the parton distributions. The important factors in drawing this conclusion are: (1) the determination of $A(\sqrt{s}, p_T)$ by extrapolating to $x_R \to 0$, which forces the shape of the $A(\sqrt{s}, p_T)$- distribution to be independent of soft physics, and (2) noticing that the magnitude of the $A(\sqrt{s}, p_T)$ function is s dependent by just the right amount, when corrected by the residual power, to make the dimension of the invariant cross sections for a given $1/p_T$ power to be fixed to the dimension of the underlying hard-scattering $\left[d\hat{\sigma}/d\hat{t}\right]_d \sim 1/(\text{GeV}/\text{c})^4$. That the measured cross sections follow this behavior is an experimental verification of the well-known factorization hypothesis [72].

## 9. Applications to HI Collisions at the LHC

Many analyses of heavy ion (HI) collisions are performed using the so-called nuclear modification factors, RpA and RAA, defined by the ratio of heavy ion data divided by p–p collision data of the same kinematic range corrected by a collision overlap factor [73,74]. Generally, these ratio measures do not attempt to separate the $p_T$- dependence from the $y$ dependence of the cross sections—or equivalently separate kinematic boundary effects, controlled by $x_R$, from the $p_T$ dependence. Our formulation, which we have used to study inclusive reactions in p–p collisions, can be applied easily to single particle and jet production in heavy ion collisions. In particular, we can separately compare the $A(\sqrt{s}, p_T, \Lambda_m)$ functions in p–p collisions with those of p–A or A–A collisions and contrast the corresponding $x_R$ behaviors as described by $n_{xR}$, the power of $(1 - x_R)$.

Since the a function is constructed by taking the limit $x_R \to 0$, it should only be dependent on the parton distributions and on the parton hard-scattering cross sections and not on the subsequent formation of QGP, fragmentation and hadronization. Hence, we expect that the a functions for p–A and A–A collisions should be quite similar to those of p–p collisions. This was indeed noted in [9].

### 9.1. Jets in HI Collisions

For a first look at the utility of our variables when applied to HI collisions, we have studied the $p_T$ dependence of the a function for high transverse momentum jets in p–Pb collisions and find that it is consistent with the a function of p–p collisions, whereas the $n_{xR}$ behaviors of the two types of collisions are quite different as noted in an analysis of ATLAS jets in p–Pb data at $\sqrt{s}$ = 5.02 TeV in our earlier publication [9]. We caution, however, that this behavior observed at high energies may not appertain to low energy. In fact, there may be differences between these dissimilar colliding beams at lower transverse momenta, where there could be sensitivity to the lower x shape of the colliding nucleon structure functions.

Following the pA case, we note that there is a congruency of the a functions of HI collisions with p–p collisions. Here, we compare p–p inclusive jets for $p_T \geq 35$ GeV/c at $\sqrt{s}$ = 2.76 TeV with those of Pb–Pb collisions at the same nucleon–nucleon energy for $p_T \geq 56.5$ GeV/c as measured by the ATLAS collaboration [75] in the centrality bin 0–10%. The corresponding $A(p_T)$ functions are shown in Figure 29, where we have normalized the Pb–Pb data to p–p data by an overall factor. From the figure, we see that the $A(p_T)$ functions for the two types of collisions (p–p and Pb–Pb) are the same within an overall scale factor (the Pb–Pb data had been normalized to the number of events, whereas the p–p data were full cross section measurements). The power law fit of the p–p data yields $n_{pT} = 6.36 \pm 0.01$ and of the Pb–Pb data $n_{pT} = 6.32 \pm 0.05$, agreeing within errors. The average ratio of

$A(p_T)_{PbPb}/A(p_T)_{pp} = 1.0 \pm 0.2$ and is flat in the interval $56.5 \leq p_T \leq 357 \, \text{GeV/c}$ with a $\chi^2/\text{d.f.} = 2.1/8$ ($p = 0.98$). Hence, we reinforce our previous conclusion from [9] that the $A(p_T)$ power laws for inclusive jets in these transverse momentum intervals for p–p, p–Pb (ATLAS $\sqrt{s} = 5.02$ TeV) and in p–p and Pb–Pb collisions at $\sqrt{s} = 2.76$ TeV have the same power indices, $n_{pT}$.

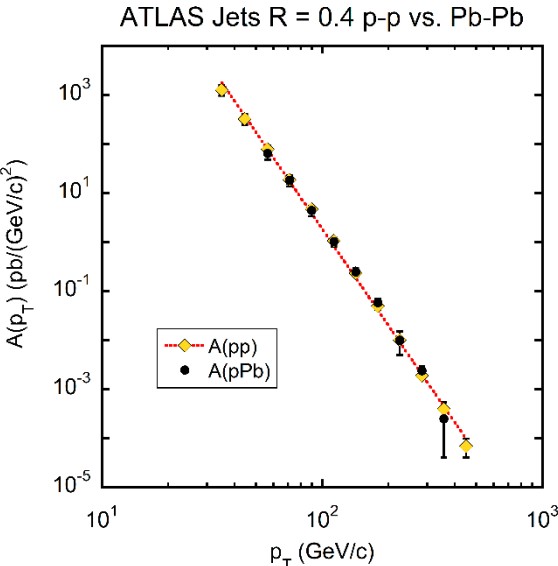

**Figure 29.** A comparison of $A(p_T)$ functions of jets measured by ATLAS at $\sqrt{s_{NN}} = 2.76$ TeV with anti-$k_t$ jet definition for R = 0.4 in both p–p and Pb–Pb collisions. The p–p data are represented by yellow diamonds and the Pb–Pb data with black circles. The Pb–Pb data were normalized to the p–p data by minimizing the $\chi^2$ of a fit to the p–p data applied to the Pb–Pb data. The red dotted line represents a power law fit to the p–p data with the average value $n_{pT} = 6.28 \pm 0.12$ (Appendix A).

The congruency of the $A(p_T)$ functions for p–p, p–Pb and Pb–Pb jet data suggests that heavy ion effects are best studied by comparing the respective $x_R$ behaviors, rather than their $p_T$ behaviors through the a functions. Remember that the a function is determined by taking the limit $x_R \rightarrow 0$ and is therefore insensitive to fragmentation and hadronization—the sector where most heavy ion effects are presumably operative. (Of course, the hard scattering of partons is controlled by the parton distributions in heavy ion collisions, which are different from those of the p–p collisions).

### 9.2. Heavy Flavors in Heavy Ion Collisions

The LHCb collaboration has studied J/ψ prompt and non-prompt production in p–p collisions [46] and in p–Pb (Pb–p) collisions [58] at $\sqrt{s} \sim 8$ TeV at very low $p_T$ where nuclear effects are expected to be quite strong. The collaboration has measured the nuclear modification factor by the ratio p–Pb data to p–p data defined by:

$$R_{pPb}(p_T, y*) = \frac{1}{A} \frac{d^2\sigma_{pPb}(p_T, y*)/dp_T dy*}{d^2\sigma_{pp}(p_T, y*)/dp_T dy*}, \tag{51}$$

where $A = 208$ for Pb and y* is the rapidity of the J/ψ in the nucleon–nucleon COM with respect to the proton direction. Data of both prompt and non-prompt J/ψ production in the two fragmentation regions (y* > 0 the p-fragmentation region and y* < 0 the Pb-fragmentation region) were analyzed. By integrating over the y* ($\sim 1.5 \leq |y*| \leq 4$), the collaboration finds a suppression of the ratio Equation (51) of ~50% at the lowest $p_T$ for direct J/ψ production in p–Pb collisions. The ratio approaches unity at higher $p_T$ (~14 GeV/c). The collaboration reports smaller modification factors for direct Pb–p (~25%) J/ψ production and smaller suppression for beauty decay (indirect) J/ψ production.

The LHCb ratio analysis is a convolution of the unseparated $p_T$ and the y* dependencies of the cross sections resulting in the $p_T$ dependence being influenced by the kinematic boundary for large |y*|. We assert that a clearer picture of the heavy ion effects can be obtained by our $[p_T - x_R]$ variables, which isolates kinematic boundary effects from dynamic effects. In general, it is natural to relate p–A and A–A collisions with p–p collisions, where A is the atomic number of the colliding nuclei, by separately studying the $p_T$ sector through ratios of the respective $A(\sqrt{s}, p_T, \Lambda_m)$ functions and by examining the ratios of the respective $[p_T - x_R]$ sectors controlled by $n_{xR}(p_T)$. Because the LHCb data are at low $p_T$, we find it unnecessary to include the $D_Q$ and $n_{xRQ0}$ terms of Equation (10).

In order to analyze these data by our method, we begin by first determining the $\Lambda_m$ and $n_{pT}$ terms of the $A(\sqrt{s}, p_T, \Lambda_m)$ functions for each of the six LHCb data sets [46,76]. The results are tabulated below. We have included all statistical and non-correlated systematic errors added in quadrature, but we have neglected the correlated systematic errors.

Note that the power indices in p–Pb collisions for direct production are larger ($< n_{pT} > = 7.4 \pm 0.1$ weighted average) than for decay production ($< n_{pT} > = 5.7 \pm 0.1$ weighted average). We checked the agreement of the LHCb $n_{pT}$ values for p–p production at 8 TeV direct vs. decay ($n_{pT}$(direct) = 6.9 ± 0.3, $n_{pT}$(decay) = 5.6 ± 0.1, respectively) versus those determined by ATLAS at 8 TeV [47] ($n_{pT}$(direct) = 6.5 ± 0.3, $n_{pT}$(decay) = 5.6 ± 0.3, respectively). Further, we observed that the decay $< n_{pT} >$ value as determined in these LHCb data is consistent with the value for b-jets, $n_{pT} = 5.6 \pm 0.2$ measured by the ATLAS collaboration. We also note that the values of $\Lambda_m$ for the four cases of p–Pb and Pb–p collisions are the consistent within errors, and that the values of $\Lambda_m$ for p–p collisions ($\Lambda_m = (4.1 \pm 0.1)$ GeV/c) are smaller than that of p–Pb collisions ($\Lambda_m = (4.7 \pm 0.1)$ GeV/c).

Remember that the $A(\sqrt{s}, p_T, \Lambda_m)$ function is determined by the extrapolation of the function $(1 - x_R)^{nxR}$ to the limit $x_R \to 0$. Thus, the a function is weakly sensitive to the primordial parton x distributions as demonstrated above. Further, we assert that the a function is not influenced by any dilution or quenching arising from heavy ion effects after the collision by assuming that there are none in the limit $x_R \to 0$. Figure 30 that shows the $A(\sqrt{s}, p_T, \Lambda_m)$ functions for p–p collisions and the two combination of p–Pb scattering for both direct and decay J/ψ production, supports this conclusion. The $p_T$ dependences of the a functions for each scattering process are remarkably similar and contain little information about heavy ion effects as expected, except that the values of $\Lambda_m$ are different for the direct and decay processes and the observation that $n_{pT}$ values for p–p collisions are somewhat smaller than the values for p–Pb and Pb–p collisions.

While we found that the $p_T$ dependences of these reactions are quite similar and insensitive to putative heavy ion effects (except for the $\Lambda_m$ parameter), the $n_{xR}$ dependences are strikingly different. In Figure 31, we show the $n_{xR}$ behaviors of the p–p and p–Pb productions of J/ψ as measured by the LHCb collaboration as a function of 1/ $P_T$, the modified transverse momentum.

It is interesting to observe that $n_{xR}(P_T)$ values for the decay channel are larger than for the direct channel but in both cases the values converge among themselves for large $P_T$ (small 1/ $P_T$). This is not unexpected. Several effects can perturb the $x_R$ distributions. One could arise from the lead nucleon PDFs being different from the proton. Another is that the J/ψ could lose energy, but remains intact as it moves through cold nuclear matter (CNM)[6]. A third possibility is that the J/ψ disintegrates as would be the case when the putative CNM acts as an opaque medium. All three mechanisms may be at play.

We consider the transparency possibility first. The different $n_{xR}$ values for the various cases shown above can be used to determine the transparency of the medium. We plot the power index differences for the four cases [$\Delta n_{xR}$(direct, decay) = $n_{xR}$(p–Pb, Pb–p) − $n_{xR}$(pp)] as a function of $P_T$ for the two hemisphere cases in Figure 32.

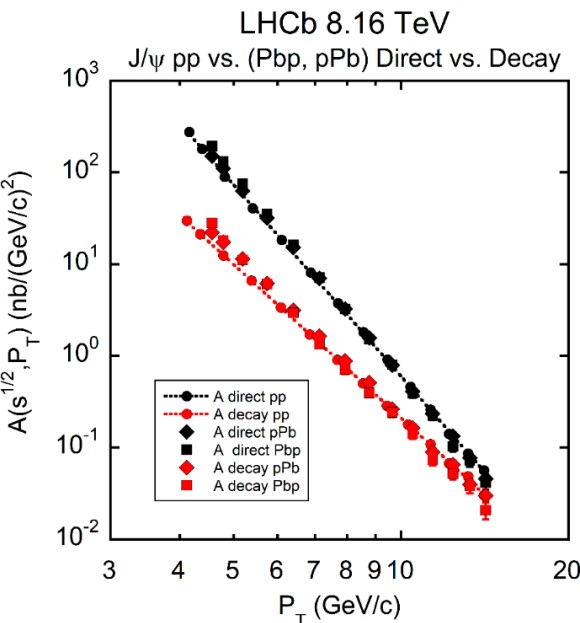

**Figure 30.** Shown are the a functions plotted versus the modified transverse momentum, $P_T$, for direct J/$\psi$ production (black points upper curve) and for decay production (red points lower curve) in p–p and p–Pb collisions measured by the LHCb collaboration. The data are consistent with pure power laws—the direct production has a larger power index than decay production. The 8 TeV p–p data have been scaled up to 8.16 TeV by adjusting $\kappa = \kappa_0 \, (\sqrt{s})^{ns}$ with the computed $n_s$ value computed from $n_{pT}$ by the dimensional custodial as shown in Figure 28 and the pPb data have been divided by the nucleus A number = 208. The p–p data have a slightly smaller power index $n_{pT}$ than the corresponding pPb, Pbp data and are in good agreement with ATLAS p–p data. The error bars were computed by adding all tabulated errors in quadrature (statistical, uncorrelated systematic and correlated systematic errors).

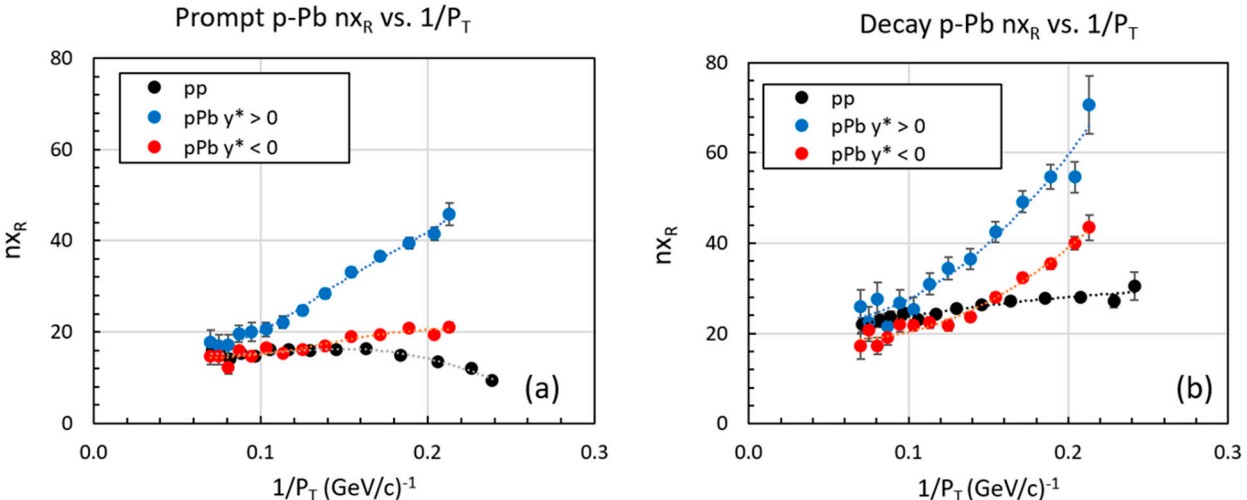

**Figure 31.** Shown is the $n_{xR}$ vs. 1/ $P_T$ behavior of direct (**a**) and decay (**b**) production of J/$\psi$ in p–Pb collisions compared with that of p–p collisions as measured by the LHCb collaboration. The black diamonds correspond to measurements of $n_{xR}$ for p–p collisions. Shown in blue circles are the values for y* > 0 where the J/$\psi$ is detected in the proton fragmentation hemisphere and in red triangles for y* < 0 corresponding to the Pb fragmentation region. The dotted lines indicate quadratic parameterizations that are used in the calculation of the transparency. Since each point in the figures above were computed for constant $P_T = (p_T^2 + \Lambda_m^2)^{1/2}$, we consider only the uncorrelated experimental errors added in quadrature.

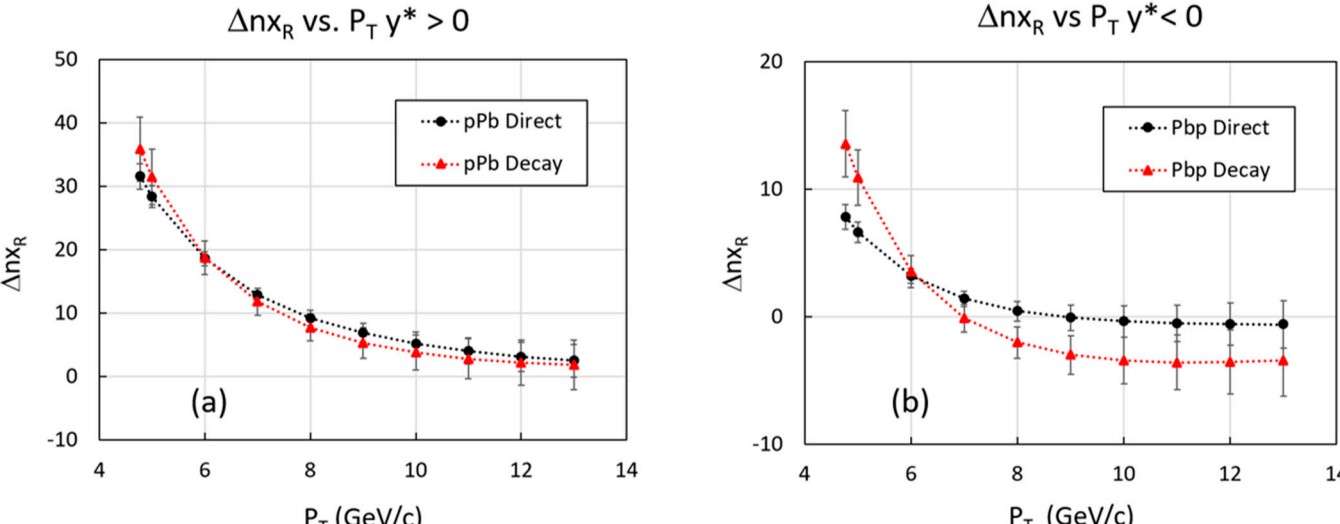

**Figure 32.** The $\Delta n_{xR}$ values are plotted against the modified transverse momentum $P_T$ for the LHCb p–p data $\sqrt{s}$ = 8 TeV and for the p–Pb data for $\sqrt{s}$ = 8.16 TeV for y* > 0 (**a**) and y* < 0 (**b**). The black circles are for J/ψ direct production and the red triangles are for decay production. (**a**) corresponds to the proton fragmentation region y* > 0, where the J/ψ has to penetrate the Pb nucleus. (**b**) is for the Pb fragmentation region y* < 0, where the J/ψ is co-traveling with the Pb debris. The dotted lines are quartic fits to $\Delta n_{xR}(P_T)$. Since there is little difference in the two cases (direct and decay) for y* > 0 (left plot), we show a simultaneous fit to both direct and decay data sets.

We posit that the transparency can be estimated by the ratio of the $x_R$ functions of the p–Pb data divided by the p–p data given by $\Delta n_{xR}$ shown in Figure 32. Here, we neglect the small differences in a functions (direct, or decay) between p–p and p–Pb collisions. Thus:

$$T(P_T, y*) = (1 - x_R)^{\Delta nxR(P_T)} = \left(1 - \frac{2P_T \cosh(y*)}{\sqrt{s}}\right)^{\Delta nxR(P_T)} \approx \exp(-\Delta nx_R(P_T)x_R), \tag{52}$$

where $T(P_T, y*)$ is the transparency at a given $P_T$ and y*, $\Delta nxR(P_T) = nx_{Rp-Pb}(P_T) - nx_{Rp-p}(P_T)$ is the difference between the measured $n_{xR}$ values for each p–Pb case (direct, decay: y* > 0, y* < 0) minus the corresponding value for p–p collisions. The modified transverse momentum $P_T = \sqrt{p_T^2 + \Lambda_m^2}$ parameterizes the power indices. The exponential expression is only approximate in the limit of small $x_R$, where the binomial function is approximately an exponential function, but is nevertheless suggestive of a transparency effect.

Examining the y* > 0 case in Figure 32a we see that the $\Delta n_{xR}(P_T)$ values, which determine the transparency of the medium through Equation (52), are the same for both direct and decay J/ψ productions, whereas in Figure 32b for the y* < 0 case, the direct and decay productions are different. Referring to Figure 32b y* < 0, we note that $\Delta n_{xR}$ for decay data becomes negative, but with large errors. This negative value makes the calculated transparency > 1, which may represent quark–antiquark recombination, but is consistent with unity within errors. The direct $\Delta n_{xR}$ values for y * < 0 are essentially 0 for $P_T$ > 8 GeV/c meaning that the transparency becomes ~ 1 above this momentum.

The y* dependence of the transparency is computed through the definition of $x_R$ for fixed $P_T$ and is plotted in Figure 33 for $P_T$ = 5 to 11 GeV/c as a function of y*. The transparencies for direct and decay production (Figure 33, black and red points) are the same within errors in the y* > 0 hemisphere and are smaller than those in the y* < 0 hemisphere. The y* > 0 is the hemisphere where the J/ψ has to penetrated the Pb nucleus. Thus, the J/ψ (direct) and its progenitor (decay) experience the same attenuations traveling through the Pb nucleus. For y* < 0, when the J/ψ is comoving with the Pb fragmentation debris, the direct production shows no recombination (black points), whereas decay (red points) does. Note that since the transparency given in Equation (52) is approximately exponential in $x_R$, the transparency is also approximately exponential in

cosh(y*) as $T(P_T, \cosh(y*)) = \exp(-\lambda \cosh(y*))$ with the coefficient $\lambda = 2P_T \Delta n / \sqrt{s}$. When $P_T = 5$ GeV/c, $\lambda = (3.5 \pm 0.2) \times 10^{-2}$ for direct J/$\psi$ production in the y* > 0 case.

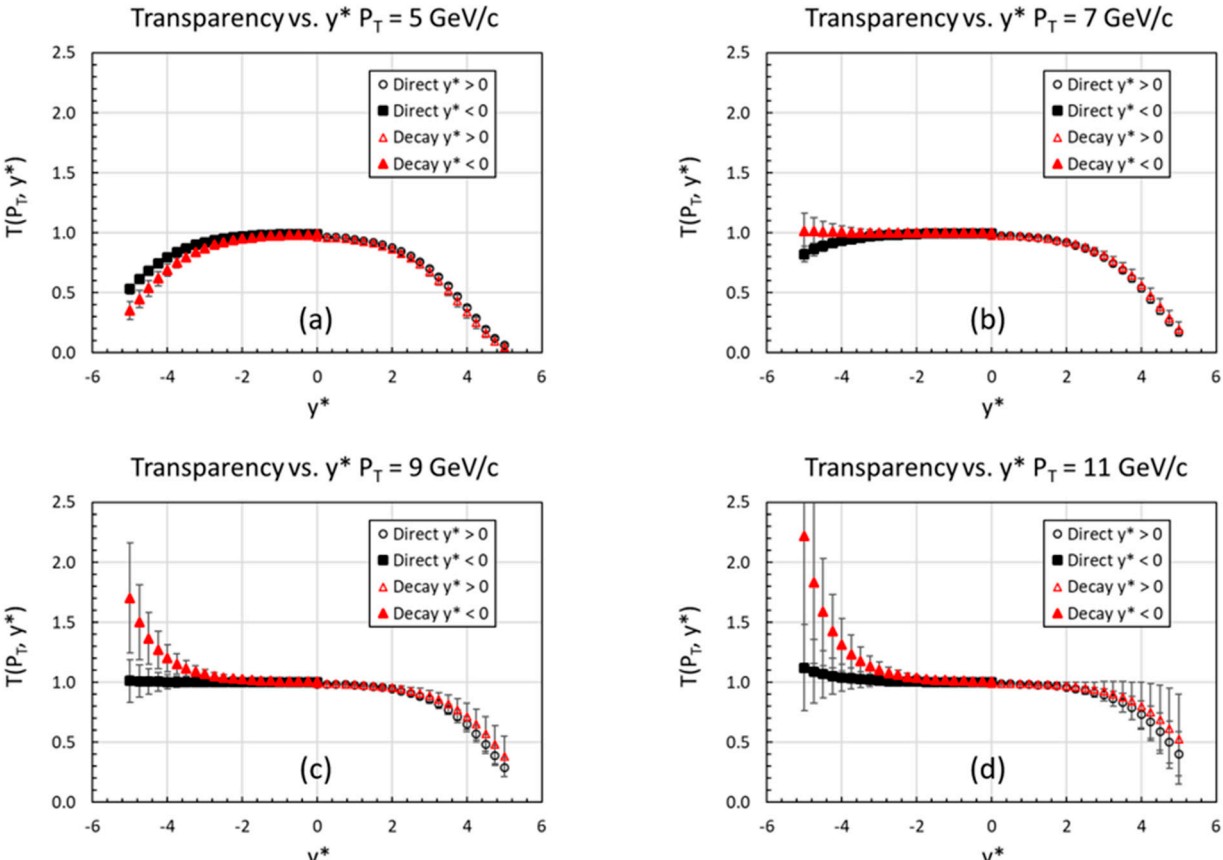

**Figure 33.** The transparency at $P_T = 5, 7, 9, 11$ GeV/c are plotted in (**a**–**d**), respectively, as a function of rapidity y* for the LHCb heavy ion p–Pb data at $\sqrt{s} = 8.16$ TeV. Direct J/$\psi$ production is represented as black circles and decay production of J/$\psi$ by red triangles. The points below |y*| < 1.5 are extrapolation beyond the range of the LHCb data. The transparency decreases with increasing $P_T$ for y* > 0. The transparency becomes almost 1 for direct J/$\psi$ production y* < 0 for $P_T > 7$ GeV/c (black circles). The decay data for y* < 0, $P_T \geq 9$ GeV/c (red triangles) are consistent with a recombination process. The error bars are determined from fits to the statistical errors.

The power indices shown in Figure 31 can be used to calculate the transparency contour. This is shown for direct J/$\psi$ production in the y* > 0 hemisphere in Figure 34. It is obvious that the transparency decreases with increasing y* and increases for increasing $P_T$. Since the transparency is a function of $x_R$, the band structure of the plot follows the contours of constant $x_R$.

Another explanation for the heavy ion effect could be that the J/$\psi$ loses momentum as it moves through the Pb nucleus (CNM), but in so doing remains intact. However, we show that this possibility is strongly disfavored. The effect of this momentum loss is to make the $n_{xR}$ values for p–Pb collisions shown above significantly larger than the corresponding ones for p–p collisions. We exploit this interpretation of the larger values of $n_{xR}$ for p–Pb collisions by equating the value of $(1 - x_R)^{n_{xR}}$ of p–p collisions with that of p–Pb collisions to solve for the value of $P_{T0}$ in p–p collisions, which we assume to be the primordial value before momentum loss following the parton–parton scattering in p–Pb collisions. Thus:

$$nx_{Rp}(P_{T0}) \ln(1 - 2P_{T0} \cosh(y*)/\sqrt{s}) = nx_{RPb}(P_T) \ln(1 - 2P_T \cosh(y*)/\sqrt{s}), \qquad (53)$$

where the right-hand side of the equation is fixed by the measured values of p–Pb J/$\psi$ production at the experimental point $P_T$ and y*, and the parameters $n_{xRp}$ and $n_{xRPb}$ are

measured separately in each data set. We solve this equation numerically to find the value of $P_{T0}$ on the left-hand side of the equation to find the p–p scattering $P_{T0}$ value which equals the p–Pb right-hand side value. The modified transverse momentum loss is then $\Delta P_{T0} = P_{T0} - P_T$. Choosing $P_T = 5$ GeV/c and $y^* = 2$ we find that for direct J/$\psi$ production in the $y^* > 0$ hemisphere, this computes to a very large modified transverse momentum loss, $\Delta P_T = 14.6 - 5.0 = 9.6$ GeV/c, or equivalently $\Delta p_T = 13.8 - 2.9 = 10.9$ GeV/c using the average values of $\Lambda_m$ given in Table 14. Such a large momentum loss would imply that the p–Pb data at the observed low momentum came from higher momentum, well above the influence of the $\Lambda_m$ term in the a functions. Since the influence of the $\Lambda_m$ term is quite evident in the $p_T$ dependence of the a function, this momentum-loss explanation of the heavy ion effect in J/$\psi$ production is ruled out. A much more consistent picture is that the differences in the $n_{xR}$ values arises from a transparency effect where the J/$\psi$ disintegrates as it moves through nuclear matter.

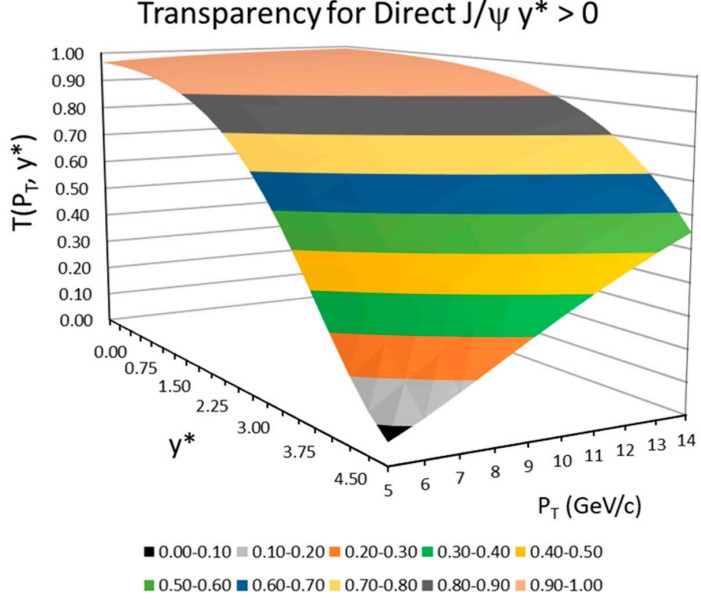

**Figure 34.** The transparency contour for direct J/$\psi$ production as measured by the LHCb collaboration at $\sqrt{s} = 8.16$ TeV is shown for the $y^* > 0$ hemisphere. The color bands correspond to contours of constant $x_R$ by which the transparency is calculated. The contour for $y^* < 1.5$ is an extrapolation beyond the acceptance of the experimental data. High momentum and small $y^*$ have the largest transparency, whereas low momentum and large $y^*$ has the smallest transparency.

**Table 14.** The parameters of the $p_T$ dependence of J/$\psi$ production in p–p and p–Pb collisions. The p–p data were taken at $\sqrt{s} = 8.0$ TeV and the p–Pb and Pb–p data were taken at a nucleon–nucleon COM energy $\sqrt{s} = 8.16$ TeV. The values of $\kappa$ for p–Pb collisions have been normalized to per nucleon by dividing by A = 208. The numbers tabulated were computed with uncorrelated systematic errors added in quadrature with statistical errors. The correlated systematic errors were not included. The values of $\kappa$ were computed with the weighted averages of $\Lambda_m = (4.1 \pm 0.1)$ GeV/c for p–p data and $\Lambda_m = (4.7 \pm 0.1)$ GeV/c for p–Pb data. The p–p $\Lambda_m$ values tend to be smaller than those of p–Pb and Pb–p collisions by 1.7$\sigma$ for direct production and 1.3$\sigma$ for decay production.

| Process | y Range | $\Lambda_m$ (GeV/c) | $n_{pT}$ | $\kappa$ (nb/(GeV/c)$^2$) |
|---|---|---|---|---|
| Direct p–p | $2.0 < y^* < 4.5$ | $4.1 \pm 0.2$ | $6.9 \pm 0.3$ | $(4.7 \pm 0.2) \times 10^6$ |
| Direct p–Pb | $1.5 < y^* < 4.0$ | $4.8 \pm 0.2$ | $7.5 \pm 0.2$ | $(1.2 \pm 0.1) \times 10^7$ |
| Direct Pb–p | $-5.0 < y^* < -2.5$ | $4.6 \pm 0.1$ | $7.5 \pm 0.2$ | $(2.3 \pm 0.2) \times 10^7$ |
| Decay p–p | $2.0 < y^* < 4.5$ | $4.1 \pm 0.1$ | $5.6 \pm 0.1$ | $(8.0 \pm 0.3) \times 10^4$ |
| Decay p–Pb | $1.5 < y^* < 4.0$ | $4.6 \pm 0.3$ | $6.0 \pm 0.3$ | $(2.7 \pm 0.2) \times 10^5$ |
| Decay Pb–p | $-5.0 < y^* < -2.5$ | $4.3 \pm 0.3$ | $5.9 \pm 0.3$ | $(4.2 \pm 0.6) \times 10^5$ |

The a functions and $x_R$ functions shown in the figures above can be used to study the Cronin effect [77]. The ratio of the a functions, each determined by the extrapolation of $x_R \to 0$, is especially interesting in that it sheds light on the $p_T$ dependence of the Cronin effect independent of the complicating influence of the kinematic boundary, thus of soft processes, and, as we have seen in the discussion above, also independent of transparency. Since the a functions are independent of y*, evidence of a Cronin effect in the a function ratio would be evidence that the effect arises from the hard-scattering domain rather than softer processes such as the subsequent fragmentation and hadronization following hard scattering. As a demonstration, we plot the ratio R(pA/pp) = A(pA)/A(pp), the ratio of a functions for the respective pA and p–p collisions, for direct J/ψ production measured by the LHCb collaboration [46,76] in Figure 35.

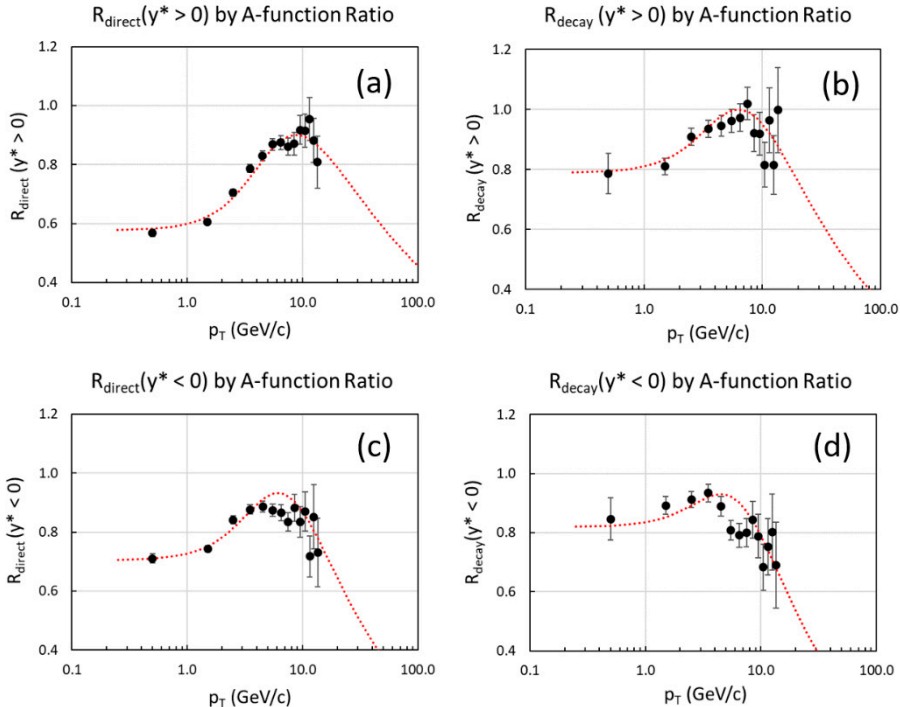

**Figure 35.** The ratio of a functions for LHCb data (back circles) as a function of $p_T$ vs. calculated ratio using the a function parameters (red dotted lines) given by Equation (54) extrapolated to $p_T = 100$ GeV/c for each of the four combinations of direct, decay, y* > 0 (direct y* > 0 (**a**), decay y* > 0 (**b**), direct y* < 0 (**c**), decay y* < 0 (**d**)) and y* < 0 J/ψ production at $\sqrt{s} = 8.16$ TeV. The data in all combinations show broad peaks at $p_T \sim 6$ to 9 GeV/c. Small adjustments upward in the magnitude of the red dotted curves have been made in the direct data ratios of 10% for y* > 0 and 5% for y* < 0, both well within the error caused by our using an average $\Lambda_m$ value for all p–Pb data and similarly for the p–p data, rather than using each $\Lambda_m$ value for each a function.

Our analysis of the a functions for inclusive single particle production is parameterized by three terms, $\kappa$, $\Lambda_m$ and $n_{pT}$, where the amplitude factor, $\kappa$, has been normalized to describe the cross section per nucleon. The suppression at low $p_T$ follows from the smaller values of $n_{pT}$ and $\Lambda_m$ for p–p than for the p–Pb data. Thus, the Cronin effect in the a function is controlled by these terms when different for p–A, A–A and p–p collisions. The Cronin effect by the a function ratio is therefore given by:

$$R_{i/j}(p_T) = \frac{A_i\left(\sqrt{s}, p_T, \Lambda_{mi}\right)}{A_j\left(\sqrt{s}, p_T, \Lambda_{mj}\right)} = \frac{\kappa_i}{\kappa_j} \frac{\left(p_T^2 + \Lambda_j^2\right)^{n_j/2}}{\left(p_T^2 + \Lambda_i^2\right)^{n_i/2}}, \tag{54}$$

where $p_T$ is the standard transverse 3-momentum, and $\kappa_{i,j}$; $\Lambda_{mi,j}$ and $n_{i,j}$ are the parameters of the respective a functions. (We have simplified our notation by call $n_i \equiv n_{pTi}$.) The ratio so expressed has a maximum at a $p_T$ value given by:

$$p_{Tmax} = \sqrt{\frac{\Lambda_j^2 n_i - \Lambda_i^2 n_j}{n_j - n_i}}. \tag{55}$$

Additionally, $R_{ij}$ has a maximum value at $p_{Tmax}$ given by:

$$R_{i/j}(p_{Tmax}) = \frac{\kappa_i}{\kappa_j} \left( \frac{\Lambda_j^2 - \Lambda_i^2}{n_j - n_i} \right)^{(n_j - n_i)/2} \frac{n_j^{n_j/2}}{n_i^{n_i/2}}. \tag{56}$$

Therefore, the Cronin effect in the a function ratio at finite momentum requires three conditions to be met: (1) the indices of the modified transverse momentum power law have to be unequal, $n_j \neq n_i$, (2) the $\Lambda_\mu$ terms must be different $\Lambda_{mj} \neq \Lambda_{mi}$ and (3) the factor underneath the radical of Equation (55) must be positive. Note that if the a function ratio were plotted vs. the modified transverse momentum $P_T$ the ratio would be monotonic and there would be no peak. All four combinations of these LHCb data, such as direct, decay, y* > 0 and y* < 0 show peaks as shown in Figure 35. Referring to the correlation of $\Lambda_m$ with m displayed in Figure 20a, we expect that the 'Cronin' peak in the a function ratio will be at smaller $p_T$ values for lighter particles so long as the three conditions listed above are met.

Thus, there is little physical significance in the a function ratio peak—it is simply the result of the power law parameters in Equation (54) above. The theory challenge then is to determine why the $p_T$ and $\Lambda_m$ parameters for p–Pb collisions are different from p–p collisions. The model of Krelina and Nemchik [78,79], for example, is based on initial state interactions with the nuclear broadening calculated by a color dipole formalism.

Our analysis has one caveat, however. Our procedure assumes that the parton distributions of the nucleons in the Pb nucleus are not so different from that of the proton as to significantly affect the a function. A measure of a possible operative difference is to compare the integrated cross section in each fragmentation region—namely, p forward vs. Pb forward in the p–Pb collisions. A difference in the parton distributions in the nucleon–nucleon COM would skew the parton–parton COM and make the average value of $\beta_{cm}$ different for the two fragmentation regions. This putative asymmetry would change the number of events in each hemisphere. Integrating the cross sections over their respective y* and $p_T$ ranges, we find that for direct production, $\sigma(Pb–p)/\sigma(p–Pb) = 1.0 \pm 0.1$; and for decay production, $\sigma(Pb–p)/\sigma(p–Pb) = 0.8 \pm 0.1$. Note that the $\Delta y^*$ intervals are the same for y* > 0 and y* < 0 but the y* ranges are different: 1.5 < y* < 4.0 and −5.0 < y* < −2.5, respectively. The errors in the ratios were computed by adding the statistical and systematic errors in quadrature, but the systematic errors dominate. These cross section ratios indicate that the putative parton differences in the relevant kinematic regions are small. However, a more complete analysis would correct for any differences in the parton distribution of the Pb nucleus and the proton.

The $x_R$ variable is a natural probe of transparency and/or energy loss since it is linearly proportional to the energy of the inclusively detected particle in the COM unlike the rapidity, y, which is mostly a measure of the COM angle. The techniques for computing the transparency and/or the energy loss through CNM (or through the Quark Gluon Plasma (QGP)) can be deployed for other heavy quark production studies as well as for jet quenching. The only requirement is to measure the inclusive cross sections in two dimensions, namely to measure both the $p_T$ as well as the y distributions in a double differential cross section. Furthermore, binning the double differential cross section data in centrality would allow quite interesting probes of the underlying physics to be performed. High statistical measurements would enable the transparency as well as the energy loss to be determined as a function of centrality and nucleus A number. The ratio of the a

functions of heavy ion collisions over that of p–p collisions should yield information about the underlying hard scattering independent of the final-state soft physics.

In summary, studying the ratio R(AA/pp) integrated over y, as traditionally performed, is a rather blunt tool for probing nuclear effects. Much more incisive is to compare the $A(p_T)$ functions and the $x_R$ dependencies separately, as shown here.

## 10. Conclusions

This paper demonstrates the utility of a formulation of inclusive invariant hadronic cross sections in p–p scattering at various values of $\sqrt{s}$ in terms of the transverse momentum $p_T$ and $x_R$, the ratio of the energy of the detected particle to the maximum energy possible in the collision COM. This paper attempts to relate observables to the underlying parton physics—especially to their primordial $2 \to 2$ hard scattering [80]. Several novel functions/concepts are introduced—the $A(\sqrt{s}, p_T, \Lambda_m)$ function, the $F(\sqrt{s}, x_R)$ function, the spine function $S_{j,\gamma}(p_T)$ and the dimensional custodian.

The invariant cross sections can be factorized in terms of two separable $p_T$ dependences, a $[p_T - \sqrt{s}]$ sector and an $[x_R - p_T - \sqrt{s}]$ sector, for many different inclusive reactions. Expressing invariant inclusive cross sections in terms of these variables allows different reactions to be compared without the particular distortions of the kinematic boundary that depend on the experimental $p_T$ and rapidity acceptance. The $[p_T - \sqrt{s}]$ sector is used to construct the $A(p_T)$ function while the byproducts of its determination define the $[x_R - p_T - \sqrt{s}]$ sector, which leads to the $n_{xR}$ vs. $p_T$ relation and, in the case of inclusive jets, $B^\pm$ and b-jets, to the corresponding $F(x_R)$ functions.

An alternate description of inclusive cross sections is referenced in [80] and discussed in the Supplemantary Materials.

Inclusive cross section data gathered at the LHC followed the high energy imperative. Data were taken mostly at the highest value of $\sqrt{s}$ = 13 TeV for searches for exotic heavy objects that would indicate physics beyond the standard model. Data at lower energies were accumulated only during the commissioning phase of the collider and at a few selected energies for the heavy ion program. Here, we have shown that measuring the s dependence uncovers systematic effects beyond what can be observed through just the $p_T$ and y dependences at the highest value of $\sqrt{s}$. Given the long period over which the data were taken and the natural maturation process in equipment calibrations and alignments, analysis and simulations, the errors of the inclusive jet and photon measurements are probably underestimated.

The a function, defined as the limit of the invariant inclusive cross section as $x_R \to 0$ evaluated at constant $p_T$, is a determination of the $p_T$ dependence of the cross sections at a unique (virtual) kinematic point and are useful in comparing different processes. By virtue of its definition, the $A(p_T)$ function is free of final-state soft processes, such as fragmentation and hadronization. Inclusive jets, inclusive direct photons, inclusive heavy quark mesons and the Z-boson have this common behavior—they all, with the exception of the Z-boson, have an $A(p_T)$ function that, to a good approximation, follows a $p_T$ power law with a signature power exponent, $n_{pT}$. The power index of the a function is closely correlated with the low-x behavior of the colliding partons. Even the heavy meson and baryon inclusive cross sections follow a power law, not in the transverse momentum $p_T$, but in the modified transverse momentum $P_T \equiv \sqrt{p_T^2 + \Lambda_m^2}$. Additionally, for these heavy particle production cross sections, determining the power law behavior in $P_T$ enables the mass of the heavy quark/meson/baryon to be cross checked through the $\Lambda_m$− m relation. We demonstrated that the a function expressed in the modified transverse momentum, $P_T$ for $B^0$ and $B^\pm$ measured at low $p_T$ by the LHCb collaboration, has the same $P_T$ power of the a function for inclusive b-jets measured by ATLAS.

An explanation of the power $n_{pT} \sim 6$, being strikingly different from the naïve expectation of $\sim 4$ is understood for inclusive jet production to involve the "shingled roof" $\hat{s}$ —weighted $p_T$ segment distributions controlled by the primordial parton–parton scattering cross section $d\hat{\sigma}/d\hat{t}$ between the experimental $p_{T\min}$ and the high- $p_T$ kinematic bound-

ary $p_T \leq \sqrt{\hat{s}}/2$. Deviations from this power law are related to the detailed shape of the colliding PDFs at low x. It was demonstrated that the power index, $n_{pT}$, for inclusive jets is strongly influenced by the power index $\mu(x, Q)$ that characterizes the $(1/x)^{\mu(x,Q)}$ low-x behavior of the gluon PDF. This enables the effective $\mu$ value of the colliding partons to be determined through the $\mu$– $n_{pT}$ relation.

The $\sqrt{s}$ dependence of the magnitude of the $A(p_T)$ function can be expressed as a power law in $\sqrt{s}$ of the form $\kappa(\sqrt{s}) = \kappa_0(\sqrt{s})^{ns}$ where the $n_s$ power index is controlled by the $p_T$ power law index, $n_{pT}$. This so-called dimensional custodial, which relates the s dependence of the a function with its $p_T$ dependence, arises from the constraint that the dimensions of the invariant cross section has to be $1/(GeV/c)^4$. By the factorization hypothesis, this dimension is that of the underlying parton–parton hard scattering. The spine function for inclusive jets and photons demonstrates the s independence of the $A(\sqrt{s}, p_T, \Lambda_m)$ power law, except for its magnitude which is controlled by $\kappa(s)$.

The residual power of the a function, $[\, n_r = n_{pT} - n_s - 4]$, arises from the $Q^2$ dependence of the strong coupling constant, $\alpha_s(Q^2)$—estimated by its slope at the lowest $p_T$ of the data set and evolution of PDFs. Its experimental evaluations in this paper for inclusive jets and inclusive direct photons are consistent with theory.

Every aspect of the hard scattering and subsequent fragmentation and hadronization goes into the $x_R$ distribution. Consequently, the $x_R$ behaviors of the invariant cross sections are much more diverse among different inclusive reactions than the corresponding $p_T$ distributions characterized by the a functions. For example, inclusive jets have a positive $D$ term, whereas inclusive photon and heavy mesons do not follow such a simple form. Comparisons of the corresponding $x_R$ distributions with QCD simulations are therefore much more stringent tests of theory than comparisons of the $p_T$ distributions. Even though $n_{xR}(p_T)$ and $n_{xRQ}(p_T)$ are always well defined, as is the a function, the $F$-function exists only for those cross sections where $D(p_T)$ and $D_Q(p_T)$ can be fit by Equation (10).

A Toy MC was developed and used to explore the dependence of the invariant cross section on the underlying parton distributions and the parton–parton scattering cross sections. It provided insights into the $p_T$ dependences of inclusive jets and photons and their $x_R$ distributions. The model is successful in simulating the main power law features of the $A(p_T)$ functions for inclusive jets and photons, but is only qualitative in emulating the $x_R$ dependences of these reactions, as it is in simulating both the $p_T$ and $x_R$ distributions of heavy meson production.

An application to heavy ion collisions using LHCb p–Pb data was given, where the transparency of nuclear matter probed by the $J/\psi$ as it moves through the Pb nucleus/proton debris is determined. The attenuation of the $J/\psi$ is more severe when the meson penetrates the Pb nucleus than when co-traveling with the Pb fragmentation debris for both direct and decay production. The transparencies for both direct and decay $J/\psi$ productions are the same within errors when the $J/\psi$ has to penetrate the Pb nucleus.

A method using the $x_R$ variable to determine particle or jet momentum/energy loss in a QGP was described. We show that the momentum loss can be estimated by the numerical solution of a log-cosh non-linear equation (Equation (53)). The a functions can be deployed to determine the Cronin effect. We give a formula for determining its peak value and position in terms of the parameters $\kappa$, $n_{pT}$ and $\Lambda_m$ that characterize the a functions. Additionally, we have shown that the a function for inclusive jet production at $\sqrt{s} = 2.76$ TeV Pb–Pb collisions has the same $p_T$ power index as the corresponding a function for p–p collisions, suggesting that HI effects are mostly in the $[p_T - x_R]$ sector.

Other applications of our formulation are envisioned, such as a study of inclusive production of selected topologies of jets. For example, gluon-initiated jets should have larger values of $D$ and $n_{xR0}$ than quark-initiated jets. The correlation of the event topology, such as jet multiplicity or N-jettiness [81] with the corresponding a functions and $x_R$ distributions, using our formulation could provide interesting tests of QCD. Remaining to do are detailed comparisons of MC simulations, such as Pythia and JETPHOX, with

data through our variables. More sophisticated fitting procedures of global data could be followed where all the correlated errors are accommodated.

This analysis suggests a rather inexpensive physics program for the LHC to operate at various values of $\sqrt{s}$ with controlled pileup to gather inclusive data, such as jets, heavy mesons and baryons and gauge bosons. These systematic data would take advantage of reduced errors as well as the refinements of tracking and calorimeter algorithms that were developed as the LHC physics program matured. One anomaly in this study is that the inclusive jet data at $\sqrt{s} = 5.02$ TeV that seem more like those of 7 TeV. Another is the s dependence of the $n_{xR}(p_T)$ functions of inclusive Z-boson production.

After analyzing such a broad list of inclusive cross sections, it has become apparent to this author that that there is an imperative that diverse collaborations present their data in a coherent manner, using the same definitions of signals, background estimations, and $p_T$ and $y$ finite binning corrections.

**Supplementary Materials:** The following are available online at https://www.mdpi.com/article/10.3390/universe7060196/s1, A discussion of the Tsallis description [80] can be found in the Supplementary Materials to this paper.

**Funding:** This research received no external funding.

**Institutional Review Board Statement:** Not applicable.

**Informed Consent Statement:** Not Applicable.

**Data Availability Statement:** All data analyzed in this paper are in the public domain and can be found throughout the references quoted. Especially helpful during the early phase of this work was the repository http://durpdg.dur.ac.uk/HEPDATA (accessed on 16 April 2021). This data compilation has been superceded by https://www.hepdata.net/ (accessed on 16 April 2021). At both sites are found data from the ATLAS, CMS and LHCb Collaborations.

**Acknowledgments:** The author is grateful to Sergei Chekanov for help in running the HepSim simulations of high energy jets and to Wit Busza, Dennis Duke, Peter Fisher and Lawrence Rosenson for clarifying discussions. Special thanks go to the late Ulrich Becker for discussion of the $\Lambda_m$−m relation. The supports of the MIT Physics Department and the MIT Laboratory of Nuclear Science are gratefully acknowledged.

**Conflicts of Interest:** The author declares no conflict of interest.

## Appendix A

The results of Minuit fits to the ATLAS inclusive jet data (R = 0.4) are tabulated as a function of $\sqrt{s}$.

**Table A1.** The kinematic regions of inclusive jet data.

| $\sqrt{s}$ (TeV) | $p_T$ (GeV/c) | $|y|$ |
|---|---|---|
| 2.76 | $25 \leq p_T \leq 285$ | $0 \leq |y| \leq 4.4$ |
| 5.02 | $45 \leq p_T \leq 716$ | $0 \leq |y| \leq 2.8$ |
| 7.00 | $25 \leq p_T \leq 1100$ | $0 \leq |y| \leq 4.4$ |
| 8.00 | $77.5 \leq p_T \leq 1040$ | $0 \leq |y| \leq 3.0$ |
| 13.00 | $108 \leq p_T \leq 1420$ | $0 \leq |y| \leq 3.0$ |

**Table A2.** The $\kappa$-$n_{pT}$ parameters are tabulated for a two-parameter fit where both $\kappa$ and $n_{pT}$ are computed and for a one parameter fit where $\kappa(s)$ is determined for a fixed $n_{pT} = 6.28$, the unweighted average of the ATLAS jet data ($2.76 \leq \sqrt{s} \leq 13$ TeV).

| $\sqrt{s}$ (TeV) | $\kappa$ [pb (GeV/c)$^{n_{pT}-2}$] | $n_{pT}$ | $\chi^2$/d.f. $p$-Value | $\kappa(n_{pT} = 6.28)$ [pb (GeV/c)$^{n_{pT}-2}$] | $\chi^2$/d.f. $p$-Value |
|---|---|---|---|---|---|
| 2.76 | $(4.9 \pm 1.8) \times 10^{12}$ | $6.18 \pm 0.09$ | $6.7/7$ $p = 0.46$ | $(2.8 \pm 0.3) \times 10^{12}$ | $208/8$ $p = 0$ |

**Table A2.** *Cont.*

| $\sqrt{s}$ (TeV) | $\kappa$ [pb (GeV/c)$^{n_{pT}-2}$] | $n_{pT}$ | $\chi^2$/d.f. *p*-Value | $\kappa(n_{pT} = 6.28)$ [pb (GeV/c)$^{n_{pT}-2}$] | $\chi^2$/d.f. *p*-Value |
|---|---|---|---|---|---|
| 5.02 | $(4.7 \pm 1.5) \times 10^{13}$ | $6.41 \pm 0.06$ | 9/11 $p = 0.62$ | $(2.4 \pm 0.1) \times 10^{13}$ | 13/12 $p = 0.37$ |
| 7.00 | $(1.6 \pm 2.6) \times 10^{13}$ | $6.10 \pm 0.03$ | 41/13 $p = 9.5 \times 10^{-5}$ | $(3.6 \pm 0.1) \times 10^{13}$ | 67/14 $p = 7 \times 10^{-9}$ |
| 8.00 | $(9.0 \pm 1.0) \times 10^{13}$ | $6.37 \pm 0.02$ | 43/27 $p = 2.6 \times 10^{-2}$ | $(5.45 \pm 0.07) \times 10^{13}$ | 61/28 $p = 3 \times 10^{-4}$ |
| 13.00 | $(2.1 \pm 0.3) \times 10^{14}$ | $6.35 \pm 0.02$ | 33/29 $p = 0.28$ | $(1.44 \pm 0.02) \times 10^{14}$ | 42/30 $p = 7 \times 10^{-2}$ |

The unweighted average value $n_{pT} = 6.28 \pm 0.12$.

**Table A3.** The $D$-$n_{xR0}$ parameters are tabulated. The $D$ value for 5.02 TeV is larger than the general trend.

| $\sqrt{s}$ (TeV) | D (GeV/c) | $n_{xR0}$ | $\chi^2$/d.f. | *p*-Value |
|---|---|---|---|---|
| 2.76 | $113 \pm 75$ | $3.4 \pm 1.0$ | 1.9/7 | 0.97 |
| 5.02 | $473 \pm 169$ | $3.1 \pm 0.7$ | 2.6/11 | 1.00 |
| 7.00 | $183 \pm 73$ | $4.0 \pm 0.5$ | 2.5/13 | 1.00 |
| 8.00 | $228 \pm 66$ | $4.1 \pm 0.2$ | 16.9/27 | 0.93 |
| 13.00 | $700 \pm 112$ | $3.6 \pm 0.2$ | 14.3/29 | 0.99 |

**Table A4.** The $D_Q$-$n_{xRQ0}$ parameters are tabulated.

| $\sqrt{s}$ (TeV) | $D_Q$ (GeV/c)$^2$ | $n_{xRQ0}$ | $\chi^2$/d.f. | *p*-Value |
|---|---|---|---|---|
| 2.76 | $(2.0 \pm 2.1) \times 10^3$ | $0.6 \pm 0.4$ | 2.2/7 | 0.95 |
| 5.02 | $(5.5 \pm 3.5) \times 10^4$ | $0.1 \pm 0.4$ | 3.3/11 | 0.99 |
| 7.00 | $(9.2 \pm 1.5) \times 10^3$ | $0.7 \pm 0.2$ | 2.9/13 | 1.00 |
| 8.00 | $(2.0 \pm 1.5) \times 10^4$ | $0.3 \pm 0.1$ | 22.9/27 | 0.69 |
| 13.00 | $(1.5 \pm 0.4) \times 10^5$ | $0.1 \pm 0.1$ | 23.7/29 | 0.74 |

**Appendix B**

Shown are the complete set of parameter dependences resulting from our simulation of 13 TeV jets with the simplified gluon distributions $xG(x) = 1/x^\mu$ and $xG(x) = (1 - x)^\nu$, with $\alpha_s(Q^2)$ evolved by $Q = p_T$. The fits to the Toy MC data were performed by MINUIT in ROOT. Each cross section 'data' point was assigned a 2% error. As discussed in the text, the two strongest behaviors are that $n_{pT}$ of the $A(p_T)$ function depends on the low x shape of the gluon distribution characterized by $\mu$, whereas the parameters $n_{xR0}$ and $n_{xRQ0}$ of the $F(x_R)$ function have a roughly linear dependence on the large x shape of the gluon PDF parameterized by $\nu$. Thus, measuring both the a function and the $F$-function gives information about the shape of the colliding parton distribution functions in the low-x and high-x regions, respectively.

We find that $\mu$ PDF case produces a purer power law than $\nu$ as determined by comparisons of log–log linear vs. log–log quadratic fits of the a function. In the case of the Pomeron, the deviation from the $1/p_T^{n_{pT}}$ power law is $\sim \pm 5\%$, whereas for the $\nu$ case with $\nu = 3$, the deviation from a pure power law is $\sim \pm 10\%$.

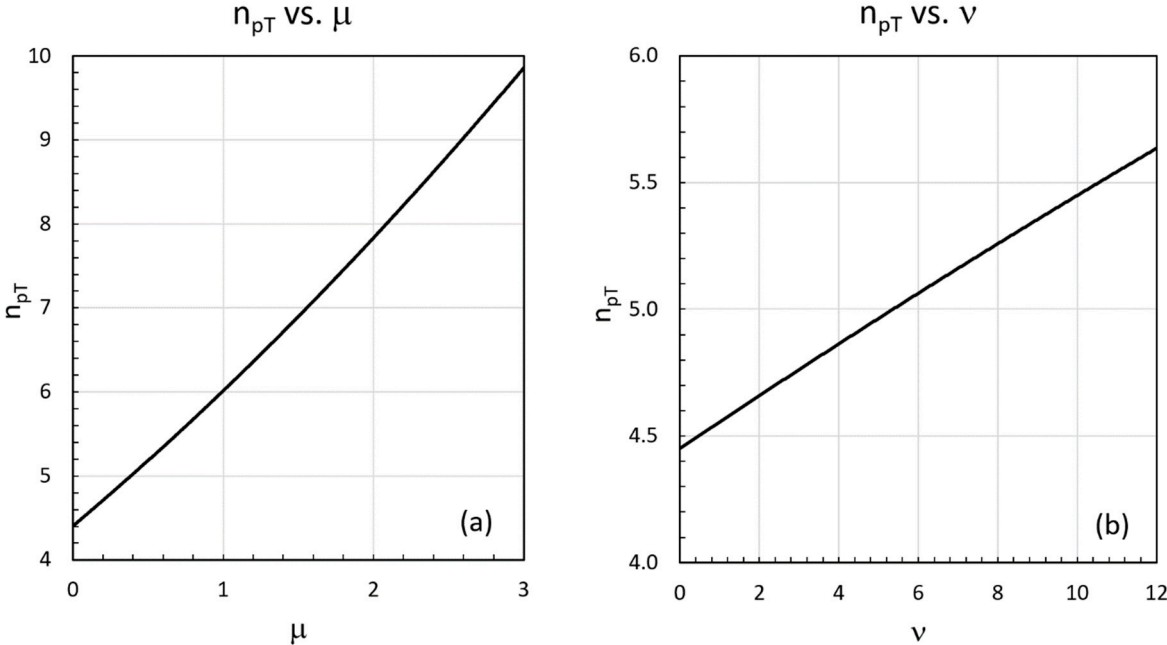

**Figure A1.** Shown is the behavior of $n_{pT}$, the $p_T$ power of the a function as a function of $\mu$ (**a**) and $\nu$ (**b**). We note that $n_{pT}$ is most strongly dependent on the low-x behavior characterized by $\mu$. In both cases, where the simulated gluon PDF was flat ($\mu = \nu = 0$), the $p_T$ power $n_{pT} \sim 4.4$ differing from the dimensional limit of 4 by the residual power $\sim 0.4$ determined by $\alpha_s^2(n_{pT})$.

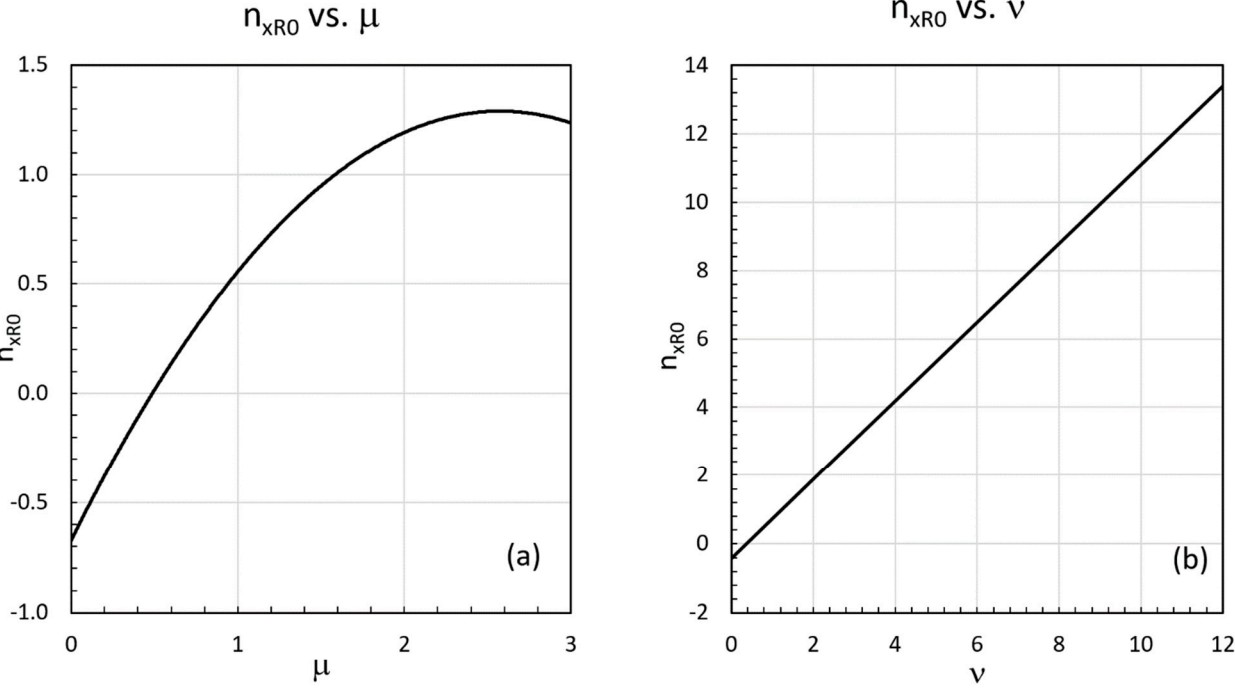

**Figure A2.** Shown are the dependences of $n_{xR0}$ on $\mu$ and $\nu$ ((**a**,**b**), respectively). It is apparent that $n_{xR0}$ is approximately linearly dependent on $\nu$ (**b**).

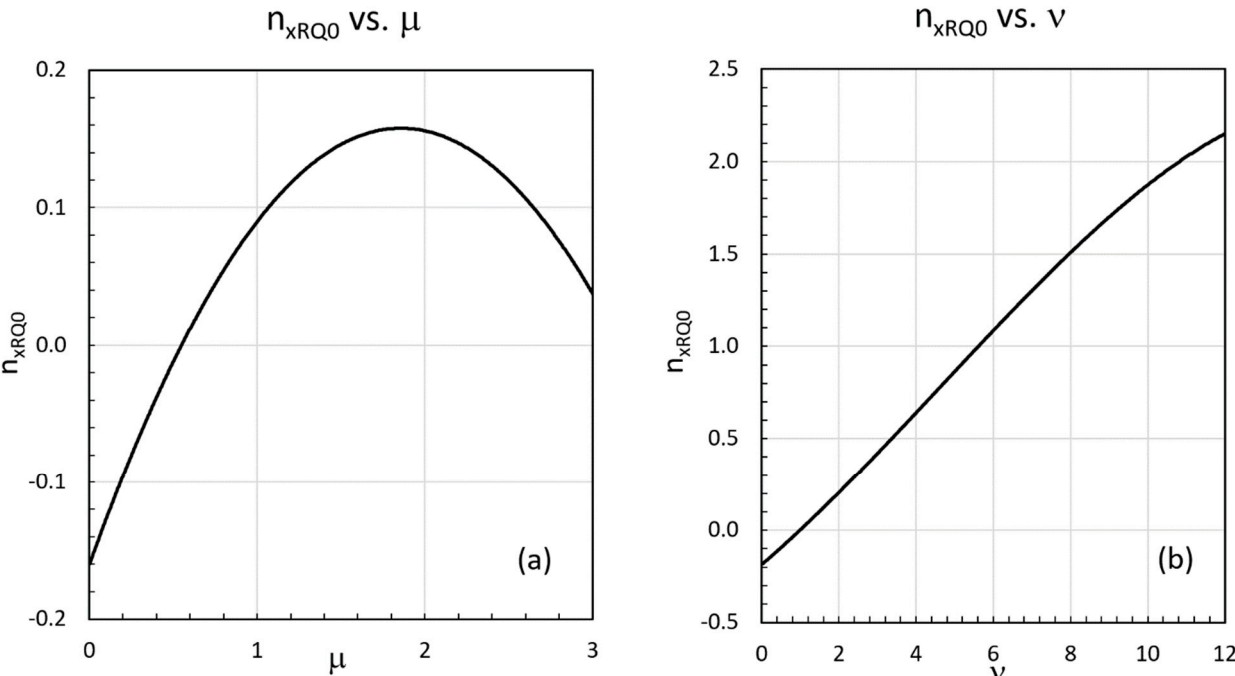

**Figure A3.** The parameter $n_{xRQ0}$ is almost independent of $\mu$ (**a**) and close to 0, but is strongly dependent on $\nu$ (**b**) and takes on a much larger value.

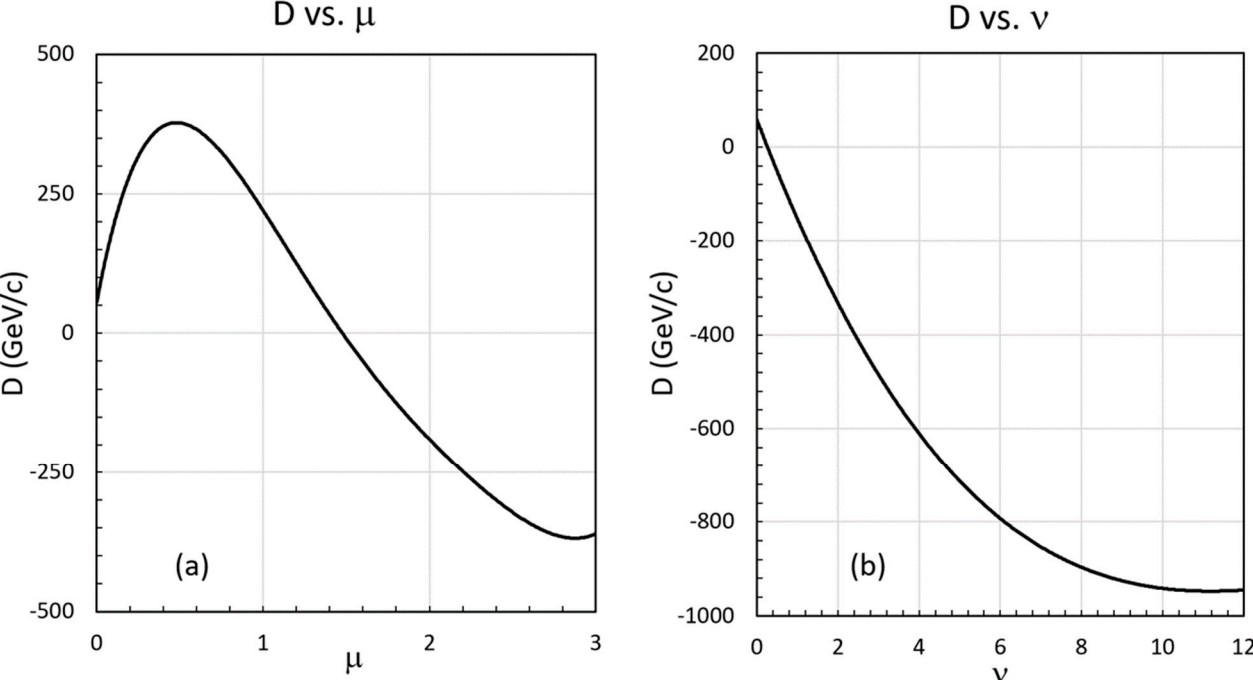

**Figure A4.** The distortion parameter $D$ has complex behaviors as functions of $\mu$ and $\nu$ ((**a,b**), respectively). As a function of $\mu$, the $D$ parameter peaks at approximately $\mu \sim 0.5$ (Pomeron limit at low x) and decreases thereafter with increasing $\mu$. On the other hand, $D$ is monotonically decreasing and negative as a function $\nu$. At $\mu \sim 1.4$, $D$ and $D_Q \sim 0$, resulting in pure radial scaling for gg $\rightarrow$ gg scattering and is also obtained at the trivial point $\mu = \nu = 0$.

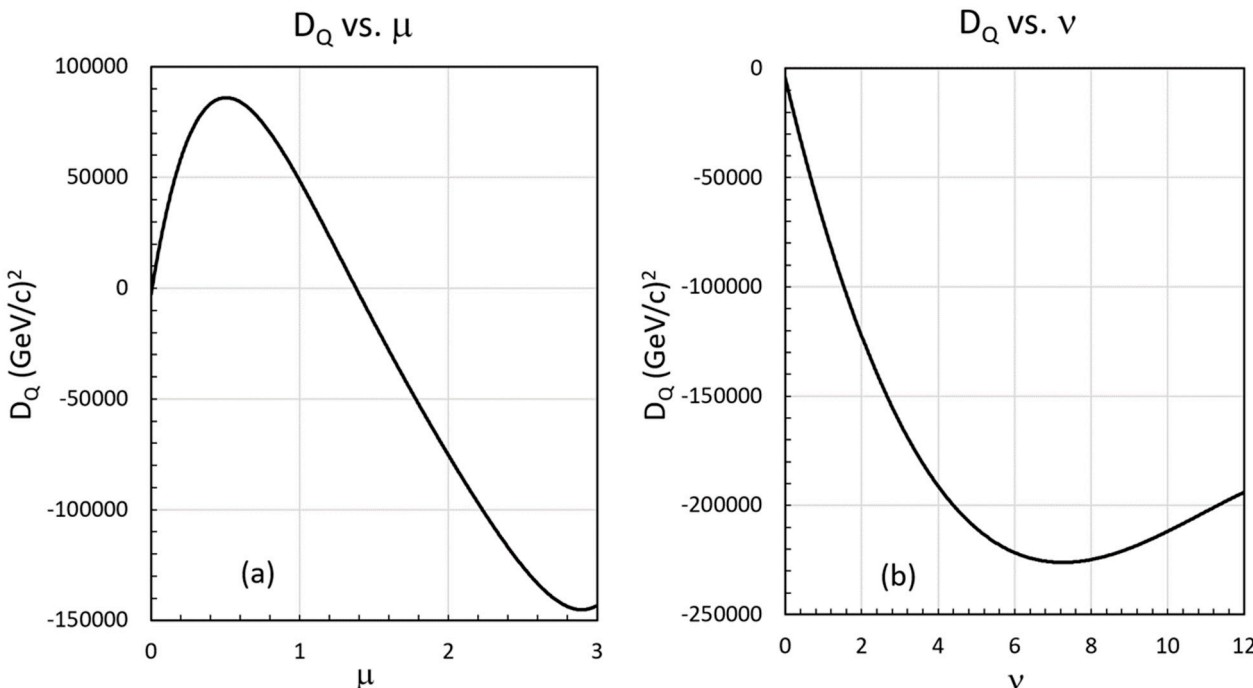

**Figure A5.** The distortion parameter $D_Q$ is shown as functions of $\mu$ and $\nu$ ((**a,b**), respectively). Both dependences follow the same general behaviors as shown in Figure A4 for the $D$ parameter.

## Notes

1     Here we note that $1 \times 10^{-13}$ cm = $(0.1975 \text{ GeV/c})^{-1}$.

2     This variable is usually called the 'transverse mass', $m_T$, but we prefer to call it the '*modified transverse momentum*' since at large $p_T$ >> m, $P_T \approx p_T$, becoming the usual transverse momentum, essentially independent of mass.

3     The power law fits to $A(p_T)$ were performed by MINUIT through calculating the full correlation matrixes. For subtle details of fitting to power laws, see: Goldstein, M.L.; Morris, S.A.; Yen, G.G., Problems with fitting to the power-law distribution. *Eur. Phys. J. B* 2004, 41, 255; https://doi.org/10.1140/epjb/e2004-00316-5.

4     In a special study, we simulated $g\,g \to g\,g$ scattering but allowed the final-state gluons to fragment by PGG(z) ~ z/(1 − z) + (1 − z)/z + z(1 − z) in the range $0.85 \leq z \leq 0.95$ with the gluon fragment lost. We find $n_{pT} = 6.60 \pm 0.03$ in comparison with no fragmentation $n_{pT} = 6.56 \pm 0.03$—hence there is little sensitivity of the A-function power law to fragmentation.

5     It is interesting to note that, although the two differential cross section definitions are well defined without singularities over the entire kinematic phase space, the Jacobian of the coordinate transformation between the two schemes has a singularity of the form ~$1/\sinh(\eta)$ for $\eta \sim 0$. This is because at this kinematic point, $\eta$ and $x_R$ are orthogonal implying that a variation in $\eta$ does not change $x_R$.

6     pA collisions, even at the LHC, are understood to be not hot enough to produce a Quark Gluon Plasma. This is discussed in [73].

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
