# Peer review of "Applications of pT-xR Variables in Describing Inclusive Cross Sections at the LHC"

_universe, doi:10.3390/universe7060196_

Round 1

Reviewer 1 Report

The manuscript proposes a different way to measure and present experimental cross sections of jets, photons, and single particles produced via the strong interaction. The idea is original, well motivated, and supported by adequate studies. However, there are some issues that need to be clarified before publication.

==== Major Comments ====

  1. The analysis of the experimental data is not presented adequately. In particular, given that the ATLAS and CMS Collaborations publish the measured cross sections in bins of transverse momentum (and rapidity), what has been used in the expression (1/pT)d^2sigma/dpTdy as pT in the denominator? Is is the average of each pT bin, or something else? Also, if an actual measurement is performed according to the proposal of the manuscript, what should the experimentalists do? Divide by pT on a jet-by-jet basis, use the average of each bin, or something else?
  2. What is the impact of the leading experimental uncertainties (e.g. jet energy scale) on the proposed observable? How much is it affected by the presence of the pT in the denominator? How will the unfolding be done (from reco to parton/particle level)?
  3. How are the results obscured by parton showering and fragmentation at the so called particle level?
  4. Although the suggestion to describe a variety of processes with the same functional form (that is strongly motivated by the underlying physics) sounds appealing, what do we actually learn about the fundamental parameters of QCD? In other words, how will the PDF fits or the measurements of the strong coupling constant will be improved? Or, how will this alternative help the MC generators to tune better their parameters?
  5. What is the sensitivity of the proposed observable on the choice of scale for the NLO calculations?How do the NLO predictions compare with data for the new observable?

==== Minor Comments ====

  1. Add legends to all figures (like Fig. 14, for example) so that they are self-explanatory.
  2. Add subfigure labels (i.e. (a), (b), etc) wherever appropriate.
  3. How was Eq. 5 derived (no reference or details given at the text)?
  4. While you mention inclusive jet production, the MC toy studies were performed assuming dijet topology. Does the presence of additional jets change the conclusions?
  5. For the assessment of the fit quality, please also report the p-values.
  6. By construction, both in ATLAS and CMS, the jets are massive and especially at low jet pT their masses cannot always be neglected. How does this affect the presented results?

Reviewer 2 Report

The multiparticle production process has been a field of intense interest, and lots of data are being analyzed evidencing new and unsuspected aspects of the QCD interaction. To understand the multitude of information, different phenomenological approaches are used in the analyses, and the present work offers a new attempt to simplify the description of the existent data in terms of new parameters that can unveil hidden patterns in the experimental data.

The approach presented here uses a factorization of the differential cross sections in terms of a new function of the variables sqrt(s) and pT, with the introduction of a new adimensional quantity, x_R, which gives the fraction of the maximum energy allowed that is carried by the parton. The author claim that with these definitions, the analysis of the HEP data leads to simple patterns that can be clearly identified. The analysis of data and a numerical approach using the Monte Carlo method indicates that, at least partially, the objective is attained. It is worth mentioning the regards on the student view of the present status of the QCD, and the difficulties the youngster researcher is faced with when trying to find her/his way in the jangle of concepts and phenomenological approaches. However, the long manuscript turns out to suffer from the same lack of clarity in some points. The method presented in this work would be easier to understand if the author was able to provide links between his concept and those already used in the area. With the simple patterns that one can be interested in appearing in the analyzes of HEP data in several ways, most often using the Tsallis distributions, the z-scaling, or even in self-similarities that were investigated rather directly in experimental data, there is a plenty of possibilities to turn the method introduced here and in previous works of the author clearer. For instance, the Tsallis Statistics provides q-exponential distributions that, at the high momentum region, turns out to be well-described by power laws. The results obtained here can be, probably, connected to the more usual Tsallis-distribution approach. The q-exponential distribution is usually associated with non-additive entropy and non-extensive thermodynamics. The emergence of the non-additivity has been investigated in several works, and it is usually attributed to temperature fluctuations, the finite size of the system or a fractal structure of QCD fields. All these features can be found in the experimental data. The z-scaling is a factorization very similar to the one used in the present work (Eq 2 - 4) and provides a universal function very similar to the one observed in Figs 5, 13,18,19, among others. The thermofractal approach, which assumes the existence of fractal structures in QCD (or in Yang-Mills fields), allows obtaining the entropic parameter, q, from the QCD field parameters, without fit. The result seems to be in excellent agreement with the experimental data analyses and gives exponents to the q-exponential function around 7, not much different from the one obtained here. Maybe the function obtained by the author can be related to some derivative of the functions obtained in those works. The logarithmic oscillations shown in Fig 25 are studied in many works using Tsallis distributions, which one could understand as another evidence that the method been developed by the author is not dissimilar from that one using Tsallis distributions. The physical origin of these oscillations is yet unknown. The q-exponential functions are not exactly power-law functions but are quasi power-law. This small difference can be the reason the exponents of the power-law functions found in this work present dependence on momentum and energy. In summary, the work is interesting and important, but those aspects mentioned above must be improved before the manuscript can be considered for publication.

Round 2

Reviewer 1 Report

I am satisfied with the author's replies.

Author Response

All suggested corrections were done.